# Data assimilation for volcanic ash plumes using a Satellite Observational Operator: a case study on the 2010 Eyjafjallajökull volcanic eruption

Guangliang Fu[1], Fred Prata[2], Hai Xiang Lin[1], Arnold Heemink[1], Arjo Segers[3], and Sha Lu[1]

[1]Delft University of Technology, Delft Institute of Applied Mathematics, Mekelweg 4, 2628 CD Delft, The Netherlands.
[2]Nicarnica Aviation AS, Gunnar Randers vei 24, NO-2007 Kjeller, Norway.
[3]TNO, Department of Climate, Air and Sustainability, P.O. Box 80015, 3508 TA Utrecht, The Netherlands.

*Correspondence to:* Guangliang Fu (G.Fu@tudelft.nl)

**Abstract.** Using data assimilation (DA) to improve model forecast accuracy is a powerful approach that requires available observations. Infrared satellite measurements of volcanic ash mass loadings are often used as input observations for the assimilation scheme. However, because these primary satellite-retrieved data are often two-dimensional (2D) and the ash plume is usually vertically located in a narrow band, thus directly assimilating the 2D ash mass loadings in a three-dimensional (3D) volcanic ash model (with an integral observational operator) can usually introduce large artificial/spurious vertical correlations.

In this study, we look at an approach to avoid the artificial vertical correlations by not involving the integral operator. By integrating available data of ash mass loadings and cloud top heights, and data-based assumptions on thickness, we propose a Satellite Observational Operator (SOO) that translates satellite-retrieved 2D volcanic ash mass loadings to 3D concentrations. The 3D SOO makes the analysis step of assimilation comparable in the 3D model space.

Ensemble-based data assimilation is used to assimilate the extracted measurements of ash concentrations. The results show that satellite data assimilation with SOO can improve the estimate of volcanic ash state and the forecast. Comparison with both satellite retrieved data and aircraft in situ measurements shows that the effective duration of the improved volcanic ash forecasts for the distal part of the Eyjafjallajökull volcano is about 6 hours.

## 1 Introduction

It has been known for many years that volcanic ash is dangerous to commercial jet aircraft (Casadevall, 1994). Little is known about the exact level of ash concentrations that becomes dangerous to the jet turbine, and the current recommendation states that the highest concentration an aircraft can endure is 4.0 mg m$^{-3}$ (EASA, 2015). Until carefully designed engine performance tests are conducted in realistic volcanic ash cloud conditions, a cautious approach to advising commercial jet operations in airspace is recommended. As a consequence, the eruption of the Eyjafjallajökull volcano in Iceland from 14 April to 25 May 2010, caused an unprecedented closure of the European and North Atlantic airspace resulting in a huge global economic loss of up to 5 billion US dollars (Oxford-Economics, 2010). Due to the major impacts on the aviation community, a lot of research

has been initiated on how to efficiently reduce these aviation impacts, starting with improving the accuracy of volcanic ash forecasts after eruption onset (Eliasson et al., 2011; Schumann et al., 2011).

For forecasting volcanic ash plumes, many Volcanic Ash Transport and Dispersion Models (VATDM) are worldwide available, e.g., PUFF (Searcy et al., 1998), HYSPLIT (Draxler and Hess, 1998), ATHAM (Oberhuber et al., 1998), NAME (Jones et al., 2007) and LOTOS-EUROS (Fu et al., 2015). Literatures have reported in-depth comparisons between volcanic ash real-time advisories and volcanic ash transport models (Witham et al., 2007; Webley et al., 2012). The meteorological wind fields and estimates of Eruption Source Parameters (ESPs) such as Plume Height (PH), Mass Eruption Rate (MER), Particle Size Distribution (PSD) and Vertical Mass Distribution (VMD) are necessary as inputs to the VATDM (Mastin et al., 2009). A VATDM uses physical parameterizations of particle sources and removal processes (including sedimentation and deposition) that affect the concentrations in a dispersing volcanic plume. Without accurate knowledge of the ash removal rate in atmospheres and the temporal variation of MER at the volcano, it is impossible to provide quantitatively accurate concentration forecasts for the ash plume arriving in an airspace over hundreds of kilometers (Prata and Prata, 2012; Fu et al., 2016).

For the purpose of improving the forecast accuracy of volcanic ash concentrations, efficient solutions must be employed to compensate the ESPs' inaccuracies. Data assimilation (DA) can be used to create accurate initial conditions for model runs by using available measurements, which is one of the most commonly used approaches for real-time forecasting problems (Evensen, 2003; Bocquet et al., 2015; Fu et al., 2015). In each assimilation step, a forecast from the previous model simulation is used as a first guess, then available observations are used to modify this forecast in better agreement with these observations. An important aspect of the assimilation approach is that it reduces the dependency on accurate knowledge of the ESPs – which are generally unknown at the time of an eruption. This is an effective approach where valid real-time volcanic ash measurements are required to guarantee the forecast accuracy (Fu et al., 2015). For the 2010 Eyjafjallajökull volcanic eruption, during volcanic ash transport, different types of scientific measurement campaigns were performed to collect information of the ash plume. The measurements contained e.g., ground-based lidar and ceilometer measurements (Pappalardo et al., 2010; Wiegner et al., 2012), satellite observations (Stohl et al., 2011; Prata and Prata, 2012), aircraft-based measurements (Schumann et al., 2011; Weber et al., 2012; Schäfer et al., 2011), ground-based in situ measurements (Emeis et al., 2011), balloon measurements (Flentje et al., 2010) and ground-based remote sensing Sun photometer observations (Ansmann et al., 2010). However, it should be noted that such measurements usually are not available globally and for remote volcanoes it is usually hard to perform measurement campaigns, especially as consequence of sudden eruptions.

Geostationary satellite measurements are of special interest, because the detection domain is large and the output data is is at high temporal frequency (typically 15 - 30 minutes). For example, the Spin Enhanced Visible and Infrared Imager (SEVIRI), on board the Meteosat Second Generation (MSG) platform provides a large view coverage of the atmosphere and Earth's surface (Schmetz et al., 2002). Images can be acquired every 15 minutes. These satellite data have been used for many years to retrieve ash mass loadings in a dispersing volcanic plume (Prata and Prata, 2012). Nowadays, ash mass loadings (Prata and Prata, 2012), the effective particle size (Kylling et al., 2015) as well as the ash cloud top height (Francis et al., 2012), are available in near real-time as satellite products during volcanic plume transport. The availability of satellite-based data provides us with

an opportunity to employ ensemble-based data assimilation with a VATDM to continuously correct the volcanic ash state, and then improve the forecast accuracy of volcanic ash concentrations.

There still exist difficulties on how to efficiently use volcanic ash mass loadings, because a VATDM is in most cases a 3D model, while the satellite-retrieved ash mass loadings are 2D data. One 2D mass loading can be considered as an integral of ash concentrations along a retrieval path (Prata and Prata, 2012). Thus, the 2D measurements are not directly suited in a 3D ensemble-based DA system. One way to ameliorate this difficulty in the analysis step of DA, is to compare the measurements and the model results in the 2D measurement space. This simply requires an observational operator to take the vertical integral of the modeled ash profile. After the analysis in the measurement space, the corrected 2D mass loadings are distributed to each vertical layer based on the prior modeled ash profile. However, this approach adds artificial vertical correlations to all the vertical ash layers when the prior modeled ash vertical profile is not accurate (Lu et al., 2016b). This is a common problem with respect to passive data assimilation due to the lack of vertical resolution in data (Blayo et al., 2014; Bocquet et al., 2015). For applications where the vertical profile is not an issue (i.e., the prior profile can be modeled well), the added artificial correlations are of minor importance (Blayo et al., 2014). However, this is not the case for volcanic ash application where the ash plume usually has significant vertical variation and is located in a narrow vertical band (e.g., see Fig. 7 in (Prata and Prata, 2012) and Fig. 15(c) in (Lu et al., 2016a)). In general, the used model-based vertical profile is very inaccurate, thus the integral approach cannot accurately reconstruct states for all/most of the vertical ash layers. The influences of the artificial/spurious vertical correlations (introduced by the integral approach with the standard DA) on the assimilation performance has been extensively studied by Lu et al. (2016b) for the volcanic ash application in a concept of variational DA. (In this paper, we also have Section 5 to discuss on this issue with respect to ensemble-based DA).

In this study we look at an approach to avoid the problem of the artificial vertical correlations. Where the satellite provides 2D ash mass loadings, 3D information is available from the model and from additional observations. A 3D Satellite Observational Operator can be derived to make both types of information directly comparable in the 3D model space. This approach does not involve the integral operator, and thus avoids the artificial vertical correlations. For this purpose, vertical information of the ash cloud, such as the ash cloud top height (de Laat and van der A, 2012), the cloud thickness and the corresponding uncertainties, should be included. Cloud-Aerosol Lidar with Orthogonal Polarization (CALIOP) (Winker et al., 2012) lidar measurements can provide detailed vertical information on plumes, but the measurements have low temporal resolution (polar-orbit) and the data processing and delivery are not designed for near real-time applications. Thus CALIOP data is not suitable to provide the near real-time thickness information for the overall volcanic ash plume.

For the vertical thickness information of volcanic ash clouds, Schumann et al. (2011) found for the 2010 Eyjafjallajökull eruption using airborne data that the volcanic ash clouds spread over large parts of Central Europe, mostly from hundreds of meters to 3 km depth. This is consistent with the results of (Marenco et al., 2011) who observed layer depths between 0.5 and 3.0 km. Dacre et al. (2015) also examined the ground-based lidar data for the Eyjafjallajökull eruption and found a mean layer depth of 1.2±0.9 km and compared this with model based estimates of 1.1±0.8 km. Prata and Prata (2012) found variable thicknesses ranging from 0.2 up to 3 km. The vast majority of data suggest thickness in the range 0.2–3 km for the 2010 Eyjafjallajökull eruption. Based on these investigations, it is not realistic to use a deterministic value to represent the overall

ash cloud thickness, but we can reasonably assume that the thickness has a range of 0.2–3.0 km at the corresponding horizontal location of the SEVIRI retrieved measurements. Although this thickness information is not deterministic, its uncertainty spread is suitable in an observational operator for satellite data assimilation. Note that the thickness range can be different for other volcanic eruptions. For example, Prata et al. (2015) reported low cloud thickness with 80% of cases for the 2006 Chaiten eruption less than 400 m, thus a thickness range of 0.1–0.4 km is recommended for that eruption. Another note is that we are only considering the distal plume, at least the part >100 km's from source, which is because close to the emission source the layering of volcanic ash did not necessarily take place. Layering is a typical property for distal volcanic ash cloud, thus the ash cloud is often called an ash plume. This property results in that usually there are clear edges around the plume, but it could indeed happen that the ash concentration decays smoothly over some vertical range, resulting in unclear ash cloud edges. In these cases, there would be long tails of very low concentrations, but the reported thickness ranges from literature can also fit, because (1) the reported thickness ranges are actually based on the visible/detectable ranges of the ash clouds, which means the observed plume edges do not exactly represent the "zero" concentration edges, but the "detectable" edges; (2) very low concentration is not of interest with respect to air-safety in volcanic ash application.

In this paper we focus on the case study of the Eyjafjallajökull volcanic ash plume in May 2010. In order to integrate data and information about volcanic ash clouds, the first goal in this study is to develop a Satellite Observational Operator to translate satellite-retrieved 2D ash mass loadings to 3D concentrations. Secondly, using the extracted 3D concentrations, we investigate whether ensemble-based data assimilation can significantly improve the volcanic ash state. Finally, the effective duration of the improved volcanic ash forecasts after satellite data assimilation is quantified.

## 2 Available data for data assimilation

In this study, geostationary SEVIRI observations for the 2010 Eyjafjallajökull volcanic eruption plume (Prata and Prata, 2012) are used as the study case to design a suitable Satellite Observational Operator (SOO) for DA. SEVIRI is a 12-channel spin-stabilized imaging radiometer. Measurements are made with a spatial resolution from 3 km × 3 km at the sub-satellite point to 10 km × 10 km at the edges of the scan. A region covering 30° W to 15° E and 45° N to 70° N is selected here for analysis which includes parts of the geographic areas affected by the Eyjafjallajökull volcanic ash (see Fig. 1). Actually the ash affected a larger area than shown, e.g, Spain (Navas-Guzmán et al., 2013), Greece (Kokkalis et al., 2013), Rominia (Nemuc et al., 2014). There is also an 'European overview' given by Pappalardo et al. (2013).

The main retrieval products from SEVIRI are ash mass loadings (Prata and Prata, 2012; Kylling et al., 2015) (see Fig. 1**a**, value at 0 means no data) where 01:00 UTC 16 May 2010 is chosen for the illustration, without loss of generality. The mass loading at each 2D pixel gives information on the ash cloud from the top view (Prata and Prata, 2012), which can be taken as an integration of ash concentrations along the retrieval path. Besides ash mass loadings, other products including the ash cloud top height (Fig. 1**b**), and the error of ash mass loadings (Fig. 1**c**, which indicates the uncertainty and accuracy of the retrieved mass loadings ) are also available in a near real-time sense (Francis et al., 2012; Prata and Prata, 2012). The ash cloud top height is adopted with the SEVIRI-KNMI product of ash height, which has been evaluated with a reasonable accuracy, as reported by

de Laat and van der A (2012). Although there is indeed an error in the ash cloud top height, the product of the errors in the SEVIRI-KNMI ash height has not been available as the product of mass loading errors. Thus, for the current study, we use the data of ash cloud top height as deterministic.

We acquire the data (described above) from the European Space Agency (ESA) funded project – Volcanic Ash Strategic Initiative Team (VAST). The data are illustrated in Fig. 1. The VAST retrieval utilizes two techniques: 1) A rudimentary cloud detection scheme implemented in the Eumetsat operational scheme called "VOLE" (http://navigator.eumetsat.int/discovery/ Start/DirectSearch/DetailResult.do?f%28r0%29=EO:EUM:DAT:MSG:VOLE), and 2) A more complex scheme called CID (Cloud Identification). This scheme is described in an Algorithm Theoretical Basis Document (ATBD) (unpublished but available here: (http://vast.nilu.no/satellite-observations/)). We have used retrievals from the CID scheme. Additional processing on the retrieved measurements is needed to translate the data from the original SEVIRI resolution (i.e., $0.1°$ longitude $\times$ $0.1°$ latitude (Prata and Prata, 2012)) to a VATDM horizontal resolution (i.e., $0.25°$ longitude $\times$ $0.125°$ latitude, as used in (Fu et al., 2016)).

Limited validation has shown that the satellite ash retrievals are sufficiently accurate for use with dispersion models to correct ash concentration forecasts (Prata and Prata, 2012; Kylling et al., 2015). However, the correction on the 3D state cannot be directly implemented by DA in the 3D state space due to the insufficient vertical resolution in satellite data (Bocquet et al., 2015). Note that this statement only holds for passive remote sensing, e.g., SEVIRI retrievals. In case of active remote sensing the contrary is true. For example, the spaceborne CALIOP lidar certainly has good vertical resolution. In the lower atmosphere it is 30-60 m (below 20 km). The same is true for ground-based lidars but usually these are of limited value. Ceilometers cannot distinguish ash from other scatterers in the lower atmosphere and cloudiness is a big problem – worse than from passive satellite because, at least for the aviation hazard, the ash needs to be elevated (above 10,000 ft or more) and above clouds (a problem then for looking upwards but not for looking downwards).

## 3  Satellite Observational Operator (SOO)

### 3.1  Derivation

The derivation of the Satellite Observational Operator (SOO) is shown in Fig. 2. The retrieved values by SEVIRI for the ash mass loadings (M) can be taken as an integration of ash concentrations along the retrieval path. In principle, the satellite retrieval path could be complicated but generally it is assumed to be a straight line (along the line-of-sight, ignoring refraction) from the measuring apparatus. The angle between the local zenith and the line of sight to the satellite is called Viewing Zenith Angle (VZA). The VZA (represented with $\alpha$) for each pixel is computed according to the satellite VZA algorithms (Gieske et al., 2005) by using general parameters (such as longitude, latitude of each pixel). With the cosine of this angle and the retrieved ash mass loadings (M), the mass loadings in the vertical direction ($M_v$) can be calculated by Eq. (1),

$$M_v = M \times \cos(\alpha). \tag{1}$$

To extract ash concentrations from SEVIRI retrievals, $M_v$ only is not sufficient and knowledge about the vertical distribution of ash cloud must be included. The cloud vertical profile can be described with the height of the top and the thickness of the cloud. As introduced in Section 2, the cloud top height ($H_{top}$) is available from satellite remote sensing and the appropriate thickness range (from $T_{low}$ to $T_{high}$, i.e., 0.2 to 3 km) of the plume has been chosen for this case based on a literature review.

Fig. 2 illustrates how the 3D ash concentrations are extracted from the obtained mass loadings in the vertical direction ($M_v$). The blue and yellow layers in Fig. 2 are determined by the lowest and the highest considered thickness ($T_{low}$ and $T_{high}$) When the top height and the thickness range of ash cloud are known, the ash concentration can be calculated by using the ash mass loadings ($M_v$) at the corresponding horizontal location. The details are formulated as follows. First we define

$$N_s = \left\lceil \frac{T_{high} - T_{low}}{\Delta T} \right\rceil + 1 \quad , \tag{2}$$

$$T_i = T_{low} + (i-1) \times \Delta T, \quad C_i = \frac{M_v}{T_i}, \qquad i = 1, 2, \cdots, N_s \quad , \tag{3}$$

where $\Delta T$ is a step length and $N_s$ is the number of the possible thickness. $T_{low}$ represents the blue layer (see Fig. 2) with the fixed thickness of 0.2 km and $T_{high} - T_{low}$ (i.e., the substraction between $T_{high}$ and $T_{low}$) represents the yellow layer with the fixed thickness of 2.8 km. $\Delta T$ is chosen at a small value compared to $T_{low}$, which guarantees $N_s$ is not too small (e.g., less than 2) to have a sufficient number of sample thickness $T_1$, $T_2$, $\cdots$, $T_{N_s}$. (e.g., $\Delta T$ is chosen as 0.05 km in this case study,

thus $N_s$ is calculated as 57.)

Corresponding to the sampled thickness (i.e., $T_1$, $T_2$, $\cdots$), the ash concentration can be calculated also as a sample from $C_1$ to $C_{N_s}$ through Eq. (3). According to Eq. (2) and Eq. (3), $T_i$ ($i = 1, 2, \cdots, N_s$) is unchanged during the dispersion of the ash cloud, while $C_i$ is temporally changed but it does not depend on $H_{top}$. Therefore, at one measurement time, the mean (C) of the sampled ash concentrations can be calculated by Eq. (4)),

$$C = \frac{1}{N_s} \sum_{i=1}^{N_s} C_i = \frac{M_v}{\frac{N_s}{\sum_{i=1}^{N_s} \frac{1}{T_i}}} \quad . \tag{4}$$

Here, we note that $\frac{N_s}{\sum_{i=1}^{N_s} \frac{1}{T_i}}$ is the harmonic mean of $T_i$ ($i = 1, 2, \cdots, N_s$). Thus the representation of Eq. (4)) can be simplified as

$$C = \frac{M_v}{T_m} \quad , \tag{5}$$

where given a harmonic-mean thickness $T_m$ (which equals to the harmonic mean of $T_i$), C as calculated by Eq. (5) is used in

this study as the extracted concentration.

Note that, if $\Delta T$ is fixed (i.e., 0.05 km in this case study), the harmonic-mean thickness $T_m$ is a constant/static plume thickness (see Eq. (2)–(5)). Another note is that the extracted concentration C only represents the ash concentrations between the heights [$H_{top}$ - $T_m$] and $H_{top}$. This is very important to guarantee the vertical profile of extracted ash concentration to be consistent with the satellite mass loadings. Therefore, the target layers for the extraction is actually between the heights [$H_{top}$

- $T_m$] and $H_{top}$, see Fig. 2.

The outcome of SOO can be considered as preprocessing to the satellite data assimilation system. The extracted data represent the data at the target layers, which can be taken as the data within the assumed layer thickness ($T_m$).

## 3.2 SOO error

Fig. 2 and Eq. (2) to (4) describe the details of the SOO. The operator transforms the 2D ash mass loadings (M) to 3D ash concentrations (C). Fig. 3**a** shows the extracted ash concentrations (C) at the target layers. It can be seen that the extracted ash concentrations in the ash plume are mostly between 0.5 and 3.0 mg m$^{-3}$.

5    Now we quantify the standard uncertainty of C ($u_C$, i.e., the SOO error), which is important for a data assimilation system. Eq. (4) can be written as C = $\frac{\cos(\alpha)}{T_m} \times$ M. Thus, given the standard uncertainty in mass loadings ($u_M$, i.e., the data of retrieval error, as shown in Fig. 1**c**), $u_C$ can be calculated as

$$u_C = \frac{\cos(\alpha)}{T_m} \times u_M \quad , \tag{6}$$

which together with C describes the 3D measurements (mean, uncertainties) for ensemble-based data assimilation.

10    Note that in this study we discuss the situation that there is an ash plume in clear-sky atmosphere. However, in general there are water and ice clouds above/below/within the ash plume, and situations where clouds move above the ash layer. In such situations the ash plume cannot be detected by the satellite algorithms (Prata and Prata, 2012), leading to wrong assimilation input. For these reasons the SEVIRI retrieval takes a very conservative approach with lots of different thresholds and conditions to be met before a pixel is deemed to be ash-affected. Thus the error is more likely to be towards under-estimating the amount 15  of ash. There is an error associated with the ash retrieval that reflects the confidence of the detection algorithm, as well as the error estimates on the derived quantities. Therefore under these situations, the SOO errors are also more likely to be an under-estimation.

## 4    Assimilation of satellite-extracted ash concentrations

### 4.1    Satellite data assimilation system

20  An ensemble-based data assimilation technique is used in this study to assimilate the SEVIRI-based ash concentrations extracted by SOO. After the ensemble Kalman filter was proposed by Evensen (1994), many other algorithms were developed such as the reduced rank square root filter (Verlaan and Heemink, 1997), the ensemble Kalman smoother (Evensen and van Leeuwen, 2000), ensemble square root filter (Evensen, 2004). Ensemble-based data assimilation allows a very general statistical description of errors and is suitable for estimation of concentrations (Evensen, 2003). Based on the ensemble formulation, 25  the dynamical model is not restricted to linearity and the implementation of the algorithm is very simple (Bocquet et al., 2015). The ensemble square root filter (EnSR, see Appendix A), in most applications a more efficient method (Evensen, 2004) than the ensemble Kalman filter, is employed in this study to perform the ensemble-based data assimilation. Note that the observational operator (**H**, see Appendix A) used in EnSR is different from SOO. SOO is an operator designed as a preprocessing procedure before data assimilation, which doesn't depend on the model space and aims to transfer 2D satellite data into 3D measurements 30  for later usage in EnSR. While, **H** is an intrinsic operator in the EnSR algorithm, as specified in Appendix A.

To simulate volcanic ash transport, the LOTOS-EUROS model (Schaap et al., 2008) is used in this study. The configurations and evaluations of the LOTOS-EUROS as a proper volcanic ash transport model was reported by Fu et al. (2015). The model run starts at 00:00 UTC 15 May 2010 with an initial ash load obtained from previous LOTOS-EUROS model run. As the model state changes with time in the numerical simulation (the time step of the model run is 15 minutes used by Fu et al. (2015)), the

model result from the previous time step is taken as the initial state for the next time step. When the model run arrives at 01:00 UTC 16 May, the volcanic ash state gets continuously modified by the data assimilation process until 00:00 UTC 18 May, by combining the extracted measurements of ash concentrations.

## 4.2   Total measurement error

To assimilate measurements in a simulation model, the total measurement error must be first estimated, which not only contains

the SOO error (Section 3.2), but also includes an estimate of the model representation error (Fu et al., 2015). The model representation error is the discrepancy between the measurement location and where the model can represent the measurement. Concentration values are defined on discrete grids with a finite resolution at discrete time steps. The grid resolution of the model used in the study is $0.25°$ longitude $\times$ $0.125°$ latitude $\times$ 1 km altitude, while the SEVIRI pixel size here is $0.1°$ longitude $\times$ $0.1°$ latitude. After a careful check on the SEVIRI measurements, a measurement location does not coincide with the grid center

point where the concentration value is defined. In this study, a preprocessing procedure before data assimilation is employed to average all measurements in a model grid to generate a new measurement value for this model grid. With this approach, one new measurement thus almost corresponds to one model state point, which means the representation error of the model is probably small. For the moment we will therefore not explicitly specify a model representation error, but implicitly assume that it is zero. Therefore, the total measurement error used in data assimilation, is equal to the SOO error in this study.

After the measurements of concentrations are extracted and the total measurement error is quantified, EnSR can be used to combine them with the LOTOS-EUROS model running to reconstruct optimal estimates.

## 4.3   Creation of ensemble plumes

The specification of uncertainties is essential for a successful data assimilation. Here we use uncertainties in the Plume Height (PH) in the process of creating ensemble members.

The PH represents the eruption height above the vent of the volcano, which was monitored with the weather radar at Keflavík (155 km west of the volcano) by Icelandic Meteorological Office (IMO), sampling every 5 minutes (Gudmundsson et al., 2010). In this study PH is taken based on the detection data of this weather radar (see the Fig. 2a in (Gudmundsson et al., 2010)), and usually the uncertainty of PH is taken as 20% (Bonadonna and Costa, 2013). The stochastic Plume Height (PH) is assumed to be temporally correlated with exponential decay. The correlation parameter $\tau$ is set to be 1 hour (Fu et al., 2015). Thus, the PH

noise ($N_{ph}$) at two times ($t_1$ and $t_2$) has the relation (Evensen, 2004) of $\mathbb{E}[N_{ph}(t_1) \cdot N_{ph}(t_2)] = e^{\frac{-|t_1-t_2|}{\tau}}$, where $\mathbb{E}$ represents the mathematical expectation.

The Mass Eruption Rate (MER) is calculated based on each uncertain PH, by using an empirical relationship (Mastin et al., 2009) between PH(km) and MER (kg s$^{-1}$)

$$PH = 2.00V^{0.241}, \quad and \quad \frac{V}{MER} = \frac{1.5e^3}{4.0e^6}. \tag{7}$$

Thus, although we only add uncertainties in PH, MER is also not deterministic.

Therefore, the different ensemble members have different PH of the ash injection (which is based on the observed PH and its uncertainties) and the different MER (which is based on Eq. (7)).

## 4.4    Assimilation performance

In the following, we first examine how data assimilation actually works in the system (see Fig. 4). The first assimilation result with EnSR (Fig. 4**a**, **b**), at 01:00 UTC 16 May 2010, is shown against the SEVIRI extracted measurements (Fig. 4**c**). Ensemble-

based data assimilation includes two steps (forecast and analysis, see Appendix A). After one-day of model running started from 00:00 UTC 15 May 2010, the EnSR forecasted state at 01:00 UTC 16 May 2010 is shown in Fig. 4**a**. Comparing the state to the extracted measurements (Fig. 4**c**), the former shows a much larger estimation compared to the latter. After the EnSR analysis step (see Fig. 4**b**), the concentrations in large parts are now closer to the extracted measurements. In reality, a potential overestimation is usually elusive and hard to avoid, which is mainly due to the difficulty in getting an accurate estimation of the

particle size distribution and modeling the physical processes (Fu et al., 2016). The comparison between the state of analysis and forecast illustrates that the EnSR assimilation process can potentially solve the problem of overestimation. Note that, in this study only PM$_{10}$ ash component is considered in the assimilation system. This is consistent with that (during satellite retrievals) only the fine particles (mostly with sizes <10.0 $\mu$m) can be detected in the tropospheric volcanic plume based on the robust and reliable retrieval algorithms (Prata, 1989; Corradini et al., 2008). It is also the main mass fraction that is transported

at large distances from the source, since most of the large particles (and therefore mass) is removed quickly from the plume.

The results above were compared in terms of concentrations, not the original mass loadings. To guarantee the assimilation performance, the comparison in concentrations only is not sufficient, because the original data is not concentrations but mass loadings. If SOO is not accurate enough for extracting the concentrations at specified heights, the assimilation results still can approximate well the inaccurate extracted concentrations due to the intrinsic forcing of ensemble-based algorithms. Obviously,

the approximation in this case is incorrect. Based on this consideration, SEVIRI ash mass loadings need to be employed for a further validation. Note that the mass loadings used for this comparison are not the original mass loadings, but the mass loadings in the vertical direction, i.e., M$_v$ in Eq. (1) as plotted in Fig. 5**a**.

During two-days continuously assimilating SEVIRI measurements of the extracted PM$_{10}$ concentrations, without loss of generality, the analyzed volcanic ash state at 12:00 UTC 17 May/ 00:00 UTC 18 May 2010 is shown in Fig. 5**c**/ Fig. 6**c**.

The conventional simulation without assimilation is also presented (Fig. 5**b**/ Fig. 6**b**), which is currently the commonly used strategy for the simulation of volcanic ash transport (Webley et al., 2012; Fu et al., 2015). It is clear that in almost the entire plume, the mass loadings by EnSR with SOO are in a better agreement with the SEVIRI mass loadings. It can be seen that EnSR with SOO effectively decreases the estimation level compared to the conventional simulation. Note that using DA to

correct for model over-estimation is an unphysical "solution". It would be more satisfying to improve the physics of the model if in fact this is the source of the over-estimation, but for this case we don't know what is the exact source (which could be e.g., meteorology, ESPs, model processes). It should also be noted that while the assimilation does correct a rather large bias in the pure model output, it doesn't mean that the DA result is better in all locations. For example, in locations around Iceland, the

DA doesn't lead the pure model; At a location (-1°W, 58°N) around England (see Fig. 5**a**), the simulation without assimilation seems to match better.

## 5   Discussion and comparison between EnSR with SOO and without SOO

We have implemented satellite data assimilation combined with SOO. However, this does not mean in terms of methodology, ensemble-based data assimilation without SOO cannot work. Under that circumstance, applying data assimilation on 2D mea-

surements and 3D concentrations, the operator $H_k$ in Eq. (A5) does not select the grid cell in $x(k)$, but integrates all the vertical grid cells to calculate 2D mass loadings. This approach (denoted EnSR without SOO) would simplify the assimilation by only using 2D mass loadings. Although we focused SOO as a new way (as introduced in the introduction) to deal with satellite 3D data assimilation, the assimilation effects between with and without SOO should be better compared/studied in order to determine whether SOO is an effective approach.

Comparing to the EnSR implementations with SOO in Fig. 5**c**/ Fig. 6**c**, the cases of EnSR without SOO are shown in Fig. 5**d**/ Fig. 6**d**. Taking the SEVIRI retrieved mass loadings (Fig. 5**a**/ Fig. 6**a**) as references, it is revealed that EnSR without SOO performs better than the case without assimlation (Fig. 5**b**/ Fig. 6**b**), but worse than EnSR with SOO. One may also want to check the assimilation effect without SOO at the initial analysis time (i.e., 01:00 UTC, 16 May 2010), which is illustrated in Fig. 7. It can be seen that the assimilation difference between with and without SOO is small (by comparing Fig. 7**c** and

7**d**). While, at the second analysis time (i.e., 02:00 UTC, 16 May 2010) the results show much bigger differences (between Fig. 7**g** and Fig. 7**h**). These results verify and examine the influences of the artificial/spurious vertical correlations (caused by the integral-type of $H_k$, see introduction). These influences are examined to be accumulated step by step in our volcanic ash application, finally resulting in the assimilation result in Fig. 6**d** which barely differs in magnitude from the forecast (Fig. 6**b**). Therefore, these influences must be considered/avoided in order to obtain an acceptable assimilation result.

Now it is explained for our application why we chose SOO to compare measurements and model results in the model space rather than in the measurement space. Although the complications would be avoided if we do the analysis step in measurement space, however, the assimilation accuracy will then be worse than the case in the 3D model space by using SOO. Note that we perform EnSR without SOO in a standard/general way. We expect the problem of artificial vertical correlations (for volcanic ash plumes) may be partially compensated by employing some diagnostic or correction approaches, as discussed in (Blayo

et al., 2014; Houtekamer and Zhang, 2016) for other applications. However, this would add many complications/difficulties to standard EnSR assimilation and whether/how it would work remains unknown (at the moment, no literature has reported on this issue for our application), and that it would be another research.

We also note that EnSR without SOO only includes the mass loadings, not the cloud height, while EnSR with SOO includes both data. Furthermore, the inclusion of the cloud height would be a strong constraint in the DA (to constrain the concentrations above the cloud height) which the analysis without SOO would not have. Thus, the weighting of the observations in each case (3D extracted concentrations or 2D mass loadings) based on their estimated uncertainties might also be the reason for the apparent differences between the two cases. In this sense, EnSR without SOO would be improved if more data (e.g., the cloud height) can be properly included. However, how to include a proper constraint of the cloud height into the case of EnSR without SOO (combined with the integral-type of $H_k$), is a difficult issue (no solution has been reported) for automatically/directly assimilating the 2D ash mass loadings.

In this study, we focus on a way not dealing with the artificial vertical correlations, and we propose SOO by incorporating data and information available. The additional data (e.g., cloud top height and thickness information) are important for SOO, which describes the structure of a volcanic ash plume; we also provide an idea in the sense of incorporating many available measurements. In addition, we expect the SOO can be potentially improved by incorporating more data, but at the moment DA with SOO has shown its advantage than the standard way (without SOO) in dealing with passive satellite data assimilation.

## 6 Evaluation of the effective forecast duration after satellite data assimilation for the distal part of the Eyjafjallajökull volcano

According to Section 4.4, the accuracy of volcanic ash state is significantly improved by ensemble-based data assimilation after a continuous assimilation period (e.g., two days). Apparently, with the improved state as initialization, an improved forecast can be obtained (Fu et al., 2015). However, it remains unknown how long the improvement on forecasts will last.

To investigate the effective duration of the improved ash forecasts after assimilation, a one-day forecast is performed by initializing EnSR analyzed state (Fig. 6**c**) at 00:00 UTC 18 May 2010. Fig. 8 shows the forecast results after assimilation (first column in Fig. 8), the SEVIRI retrieved measurements (second column in Fig. 8) and the forecast results without assimilation (third column in Fig. 8). Without loss of generality, 06:00, 12:00 and 18:00 UTC are chosen to evaluate the comparisons over three divided distal regions of interest (denoted R1–R3). Note that we do not evaluate the near-volcano region (i.e., the left part of R1 in Fig. 8**a**), where the improved forecasts can be quickly influenced by the continuously noisy emissions. Thus it is safe to take the effective forecast duration for the near-volcano region as "zero hour".

For the region R1, the improved forecast after assimilation can last for 6 hours. This is viewed by that at 06:00 UTC, the forecast after assimilation (see R1 in Fig. 8**a**) is closer to the measurements (R1 in Fig. 8**b**) than the forecast without assimilation (R1 in Fig. 8**c**). However, the improvement diminishes after 6 hours (see the second and third rows in Fig. 8).

For the region R2, the forecast after DA at 06:00/12:00 UTC ( R2 in Fig. 8**a**/ 8**d**) has a good match with the retrieved mass loadings (R2 in Fig. 8**b**/ 8**e**). However, we are not sure whether the good match remains at the time 18:00 UTC, because there seems to be lack of data at that time (see R2 in Fig. 8**h**). Therefore, it is better to evaluate the effective forecast duration after DA with SOO for R2 is 12 hours.

For the region R3, at the three times 06:00/12:00/18:00 UTC, the forecasts after DA are generally better than forecasts without DA by comparing with the measurements. However it should be noted that in Fig. 8**e**, some higher values (over 3.5 g m$^{-2}$) are indeed retrieved in R3, but not in the forecasts after DA.

Our initial experimental tests show that, the effective time durations of the forecasts after satellite data assimilation with SOO can be considered as 6 hours for the region R1, 12 hours for the region R2, and 18 hours for the region R3. It also shows to us that the effective forecast duration tends to be longer as the region is farther away from the volcano. This is because longer time is needed for the influences of inaccurate ESPs to reach the areas farther from the volcano.

These results are evaluated using satellite retrieved data. For a further test, we employ another type of independent data. For this purpose, we use aircraft-based measurements (see details in Appendix B). Fig. 9**a** shows the flight route of the aircraft (in the region R3). Fig. 9**b** and 9**c** are the comparison of aircraft PM$_{10}$ measurements against the forecasted concentrations after assimilation and without assimilation. Note that the low magnitude of aircraft measurements may result in little importance to verify high ash concentrations, but comparisons using them can indicate how good the assimilation is at reconstructing the outskirts/boundaries of the ash plume, which is important to describe the plume's structure.

For the period from 09:30 to 11:00 UTC (Fig. 9**b**), although the forecasting time has been over 9 hours (i.e., the last assimilation is 9 hours ago), the forecasted concentrations still have a good match with the accurate aircraft measurements, while the conventional forecast (i.e., forecast without assimilation) doesn't. This result shows the forecast over 11 hours after assimilation has also kept a high accuracy compared to the measurements. The result can be extended to 15 hours comparing with the other period from 12:30 to 15:00 UTC (Fig. 9**c**). This test result using independent aircraft data confirms our evaluation result using satellite data that for the region R3, the effective forecast duration after DA is about 18 hours.

Based on all of the above evaluations and in the view of the whole distal regions of interest (R1+R2+R3), 6 hours can be taken as the a reasonable effective time duration (after assimilation with SOO) for the case study. This time duration can be taken as an indication about how long a valid regional aviation advice based on the forecast after assimilation can last. It should be noted that for other cases (e.g., another volcanic eruption), the effective duration is different because it depends on the specific weather condition and the specific model used for forecasting. Thus, the effective duration for other case studies should be re-evaluated, and what we presented in this section can be useful for the readers to test it.

## 7 Conclusions

In this paper, we choose the Eyjafjallajökull volcanic ash plume in May 2010 as the study case. In this study, a Satellite Observational Operator (SOO) was developed to translate 2D satellite ash mass loadings to 3D ash concentrations. To extract ash concentrations, not only the SEVIRI data of ash mass loadings, ash cloud top height are employed, but also a reasonable assumption of the ash cloud thickness range (0.2–3 km), at the corresponding horizontal location of the SEVIRI retrieved measurements, are combined. One advantage of SOO is that it can use rough thickness information to get uncertain concentrations, which are suitable for the data assimilation methodology.

The extracted ash concentration measurements enable us to perform ensemble-based data assimilation in a 3D volcanic ash transport model. By employing a preprocessing procedure before data assimilation to generate new measurement values by averaging all surrounding measurements, the model representation error is approximately zero. The SOO error is also calculated, and the total measurement error (defined as the sum of the SOO error and the model representation error) is therefore quantified, which together with the concentrations describe the 3D measurements (mean, error) for a data assimilation system. The results showed the assimilation with SOO significantly reduces the estimation level of the conventional simulation. The accuracy of the volcanic ash state was shown to be significantly improved by the assimilation of satellite mass loadings.

In this study, we proposed SOO as a new manner in dealing with passive satellite data assimilation. The assimilation with SOO was verified and examined to be more advantageous than the standard assimilation which introduces severe artificial vertical correlations in volcanic ash application. The development of SOO provides an idea of incorporating data and information available together to provide more accurate volcanic ash forecasts.

With the improved volcanic ash state as initialization, improved volcanic ash forecasts are obtained. Evaluations using both satellite retrieved data and aircraft in situ measurements showed that the effective forecast time duration after satellite data assimilation with SOO is about 6 hours for the distal part of the Eyjafjallajökull volcanic eruption. The results also showed that the effective forecast duration (after DA with SOO) tends to be longer as the region is farther away from the volcano, which is because inaccurate ESPs require longer time to impact on the regions farther from the volcano. Note that for other cases (e.g., other volcano), this duration should be different and should be re-evaluated due to its dependence on the specific weather condition and on the specific model.

In this study, we developed SOO by considering cases where one singular ash cloud is present. Actually, it could happen that there are several isolated volcanic ash clouds in the vertical direction. The methodology of SOO is also valid for these cases, where the top isolated ash cloud does not correspond to the full but to a fraction of SEVIRI ash mass loadings. In this study, two of the Eruption Source Parameters (ESPs)–Plume Height (PH) and Mass Eruption Rate (MER) are made uncertain to generate DA ensembles. Actually the uncertainties can also be added in other ESPs, e.g., Vertical Mass Distribution (VMD) and Particle Size Distribution (PSD). However, adding noise in VMD and PSD should be very careful to keep their empirical/realistic distribution, e.g., "umbrella" shaped VMD (Sparks et al., 1997) and PSD for different types of eruption (Durant and Rose, 2009). Otherwise, the noisy VMD or PSD could provide unphysically biased prior ensemble plumes, resulting in DA algorithm impossible to reconstruct physical plume estimates. In this paper, we applied an off-line approach for model running and simply used the deterministic meteorological input data. Actually these data also contain uncertainties which have an influence on ash cloud transport. In future work, for more accurate ash forecasting, uncertainties in the meteorological data like wind speed should also be taken into account.

Recently, a large number of national weather services has implemented ceilometer networks, mainly for monitoring the dispersion of volcanic ash clouds (Wiegner et al., 2014). These data set will be (and in part are already) available in near real time and will provide information about the (horizontal and) vertical distribution (with some restrictions due to cloud cover; but this is also true for space borne observations). Thus, they could be promising candidates for data assimilation as well, but

there are indeed some caveats. It will be interesting for future research to make use of and demonstrate the case for using ground-based lidars.

## 8 Data availability

All the satellite data shown in Fig. 1 are available and can be downloaded from http://vast.nilu.no/test-database/volcano/
Eyjafjallajokull/eruption/2010-04-14/main_data_type/Satellite/specific_data_type/seviri/ (Registration required). The aircraft
data used in this study are available from Fig. 9**b**, **c**. The model output data can be accessed by request (G.Fu@tudelft.nl).

## Appendix A: The ensemble square root filter

The ensemble square root filter (EnSR) is essentially a Monte Carlo sequential method (Evensen, 2003), based on the representation of the probability density of the state estimate by an ensemble of $N$ states, $\xi_1, \xi_2, \cdots, \xi_N$. Each ensemble member
is assumed as one sample of a true state distribution. The required ensemble size depends on the model's nonlinearity and the the involved uncertainties. For the application of the filter algorithm to a volcanic ash transport model, an ensemble size of 50 is considered acceptable for maintaining a balance between accuracy and computational cost (Fu et al., 2015, 2016). In the first step of this algorithm an ensemble of $N$ volcanic ash state $\xi^a(0)$ is generated to represent the uncertainty in the initial condition $\mathbf{x}(0)$. In the second step (the forecast step), the model propagates the ensemble members from time $t_{k-1}$ to $t_k$:

$$\xi_j^f(k) = M_{k-1}(\xi_j^a(k-1)). \tag{A1}$$

The state-space operator $M_{k-1}$ describes the time evolution from the time $t_{k-1}$ to $t_k$ of the state vector which contains the ash concentrations in all the model grid boxes. The filter state at time $t_k$ is a stochastic distribution with mean $\mathbf{x}^f$ and covariance $\mathbf{P}^f$ given by:

$$\mathbf{x}^f = \frac{1}{N}[\sum_{j=1}^{N} \xi_j^f] \quad, \tag{A2}$$

$$\mathbf{L}^f = [\xi_1^f - \mathbf{x}^f, \cdots, \xi_N^f - \mathbf{x}^f] \quad, \tag{A3}$$

$$\mathbf{P}^f = \frac{1}{N-1}[\mathbf{L}^f(\mathbf{L}^f)'] \quad, \tag{A4}$$

The observational network at time $t_k$ is defined by the observation operator $H$ that maps state vector $\mathbf{x}$ to observation space $\mathbf{y}$ by

$$\mathbf{y}(k) = H_k(\mathbf{x}(k)) + \mathbf{v}(k), \tag{A5}$$

where the observation error $\mathbf{v}$ is drawn from Gaussian distribution with zero mean and covariance matrix $\mathbf{R}$. Here, $\mathbf{y}$ contains the measurements of ash concentrations and $\mathbf{R}$ is assumed to be a diagonal matrix with the square of the standard deviation (measurement uncertainty) as diagonal entries. The operator $H$ selects the grid cell in $\mathbf{x}(k)$ that corresponds to the observation

location. When measurements become available, the ensemble members are updated in the analysis step using the Kalman gain and their ensemble covariance matrix following:

$$\mathbf{K} = \mathbf{P}^f \mathbf{H}' [\mathbf{H} \mathbf{P}^f \mathbf{H}' + \mathbf{R}]^{-1} \quad , \tag{A6}$$

$$\xi_j^a = \xi_j^f + \mathbf{K}[\mathbf{y} - \mathbf{H} \xi_j^f + \mathbf{v}_j] \quad , \tag{A7}$$

$$\mathbf{P}^a = (\mathbf{I} - \mathbf{K}\mathbf{H}) \mathbf{P}^f \quad , \tag{A8}$$

where $\mathbf{v}_j$ represents realizations of the observation error $v$. To reduce the sampling errors introduced by adding random numbers $\mathbf{v}_j$ to the observations, the analysis step can be written in a square root form (Evensen, 2004; Sakov and Oke, 2008a, b). Using the notations $\mathbf{Y} = \mathbf{H}\mathbf{L}^f$ and $\mathbf{S} = \mathbf{Y}\mathbf{Y}' + \mathbf{R}$, the updated covariance matrix becomes:

$$\mathbf{P}^a = \mathbf{L}^a (\mathbf{L}^a)' = \mathbf{L}^f (\mathbf{I} - \mathbf{Y}' \mathbf{S}^{-1} \mathbf{Y})(\mathbf{L}^f)' = \mathbf{L}^f \mathbf{T} \mathbf{T}' (\mathbf{L}^f)' \quad , \tag{A9}$$

thus $\mathbf{L}^a$ can be represented by

$$\mathbf{L}^a = \mathbf{L}^f \mathbf{T} \quad , \tag{A10}$$

where $\mathbf{T}$ is an $N \times N$ matrix which satisfies: $\mathbf{T}\mathbf{T}' = \mathbf{I} - \mathbf{Y}' \mathbf{S}^{-1} \mathbf{Y}$. It can be easily shown that there is a unique symmetric positive definite solution defined as the square root of the symmetric positive definite matrix: $\mathbf{T}^s = [\mathbf{I} - \mathbf{Y}' \mathbf{S}^{-1} \mathbf{Y}]^{\frac{1}{2}}$. By using the eigenvalue decomposition, the matrix $\mathbf{T}^s$ has the following form:

$$\mathbf{T}^s = \mathbf{C} \mathbf{\Lambda}^{\frac{1}{2}} \mathbf{C}', \tag{A11}$$

where $\mathbf{T}^s$ is referred as the symmetric factor. The symmetric algorithm defined above introduces the smallest analysis increments for an arbitrary compatible norm. The good performance of EnSR has been obtained on improving the forecast accuracies without introducing additional sampling errors (Evensen, 2004; Sakov and Oke, 2008a).

## Appendix B: Aircraft in situ measurements

Aircraft-based measurements allow sampling of the ash cloud with a high spatial and temporal resolution and by using optical particle counters (OPC) this type of measurement is estimated with a high accuracy of 10% (Weber et al., 2010).

In this study, the available aircraft measurements on 18 May 2010 from 09:30 to 15:30 UTC were used. The measurements were performed by the group Environmental Measurement Techniques at Düsseldorf university of Applied Sciences. The measurements took place in the North-West part of Germany including the border between the Netherlands and Germany, see Fig. 9**a**. The aircraft took off from the airfield "Schwarze Heide" in the Northern part of the Rhein-Ruhr area, headed along the Dutch-German border in the direction of the North Sea, continued towards Hamburg and then returned to the airfield. Along the route, concentrations of $PM_{10}$ and $PM_{2.5}$ were measured.

Note that the aircraft measurements represent $PM_{10}$ concentration at inlet position while the model gives values which are averages of the concentration for a $0.25° \times 0.125° \times 1$ km grid box. Assuming the uniformity of the fields, it is appropriate to

compare these two values given that the measurements are made at different resolutions and by different methods. However, when the heterogeneity is a serious issue, the model representation error should also be better considered.

*Author contributions.* All authors participated in the design of the experiment and the analysis of the assimilation results. Guangliang Fu, Arjo Segers and Sha Lu modeled the volcanic ash transport using the LOTOS-EUROS. Guangliang Fu and Fred Prata validated SEVIRI measurements and provided them for the development of SSO. Guangliang Fu, Hai Xiang Lin, Arnold Heemink, and Fred Prata carried out data assimilation experiment, analyzed the results and finalized the paper.

*Acknowledgements.* We are very grateful to the editor and reviewers for their reviews and insightful comments. We thank the European Space Agency (ESA) project – Volcanic Ash Strategic initiative Team (VAST) for providing the satellite data for the 2010 Eyjafjallajökull volcanic eruption. We thank Prof. Konradin Weber (University of Applied Sciences, Environmental Measurement Techniques, Düsseldorf, Germany) for providing the real-time aircraft in situ measurements. We thank Dr. Arve Kylling at NILU-Norwegian Institute for Air Research for his kind assistance with interpretations of volcanic ash cloud height. In this paper, we perform the ensemble-based data assimilation using OpenDA (open source software, www.openda.com). The first and the last author thank the China Scholarship Council for the financial support to this research. We would also like to thank the Netherlands Supercomputing Centre (SURFsara) for providing us the computing facility: the Cartesius cluster.

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

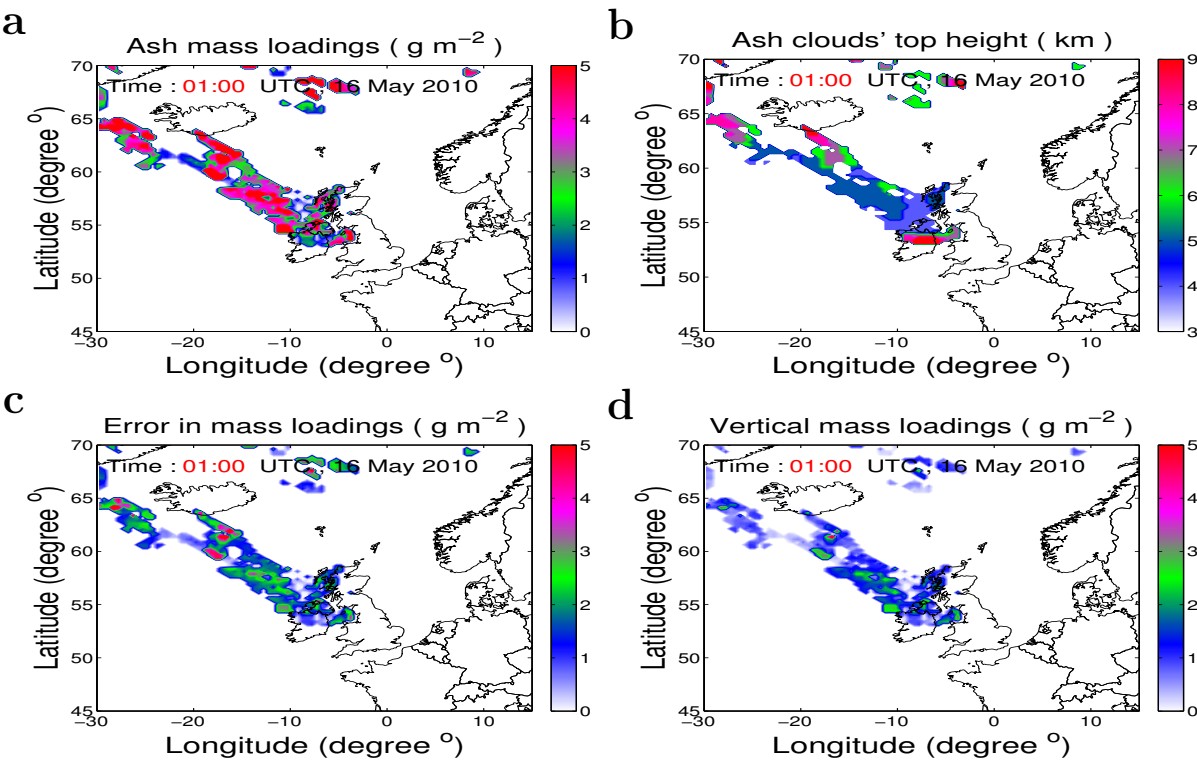

**Figure 1. Available volcanic ash data from SEVIRI on 16 May 2010 at 01:00 UTC.** Data are acquired from the European Space Agency (ESA) funded projects Volcanic Ash Strategic Initiative Team (VAST). **a**, Ash mass loadings. Values at 0 mean no data. **b**, Ash cloud top height. **c**, Error in the retrieved ash mass loadings **a**. **d**, Ash mass loadings in the vertical direction, i.e., $M_v$ in Eq. (1).

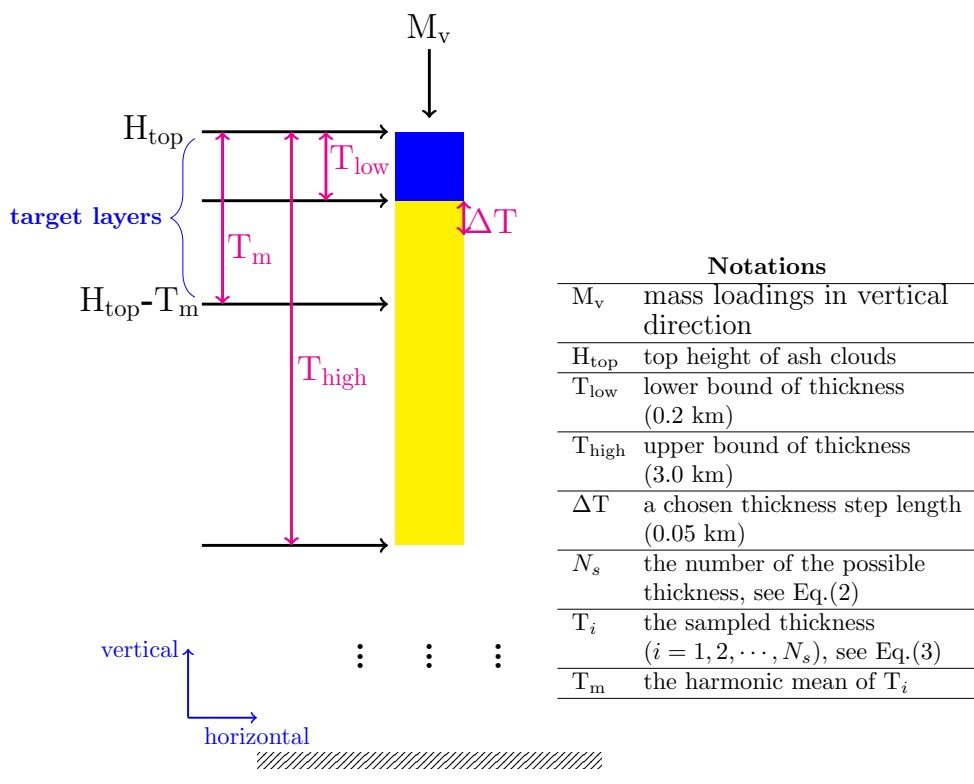

**Figure 2. Illustration of the Satellite Observational Operator (SOO).**

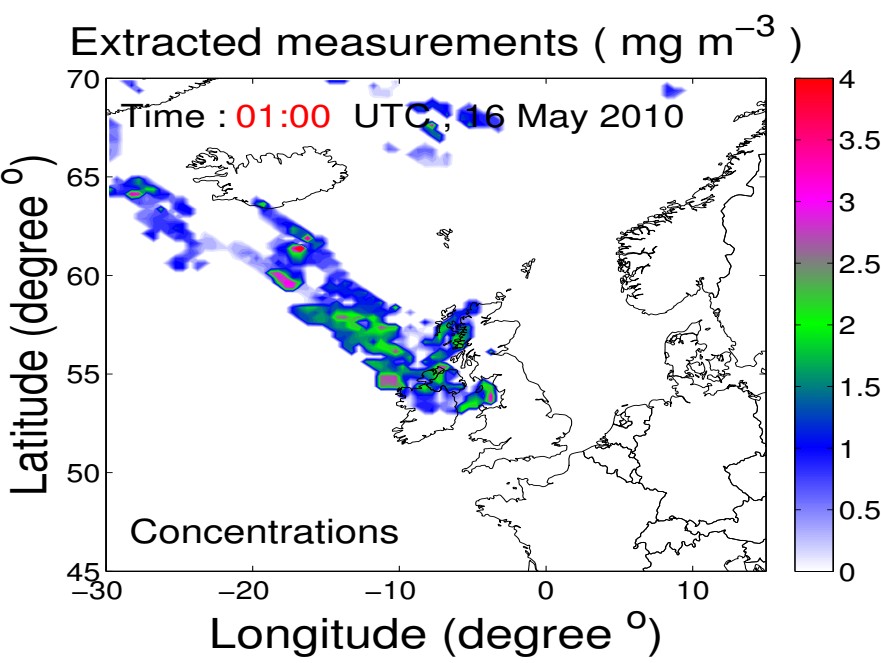

**Figure 3. Extracted ash concentrations by the Satellite Observational Operator (at 16 May 2010 at 01:00 UTC).**

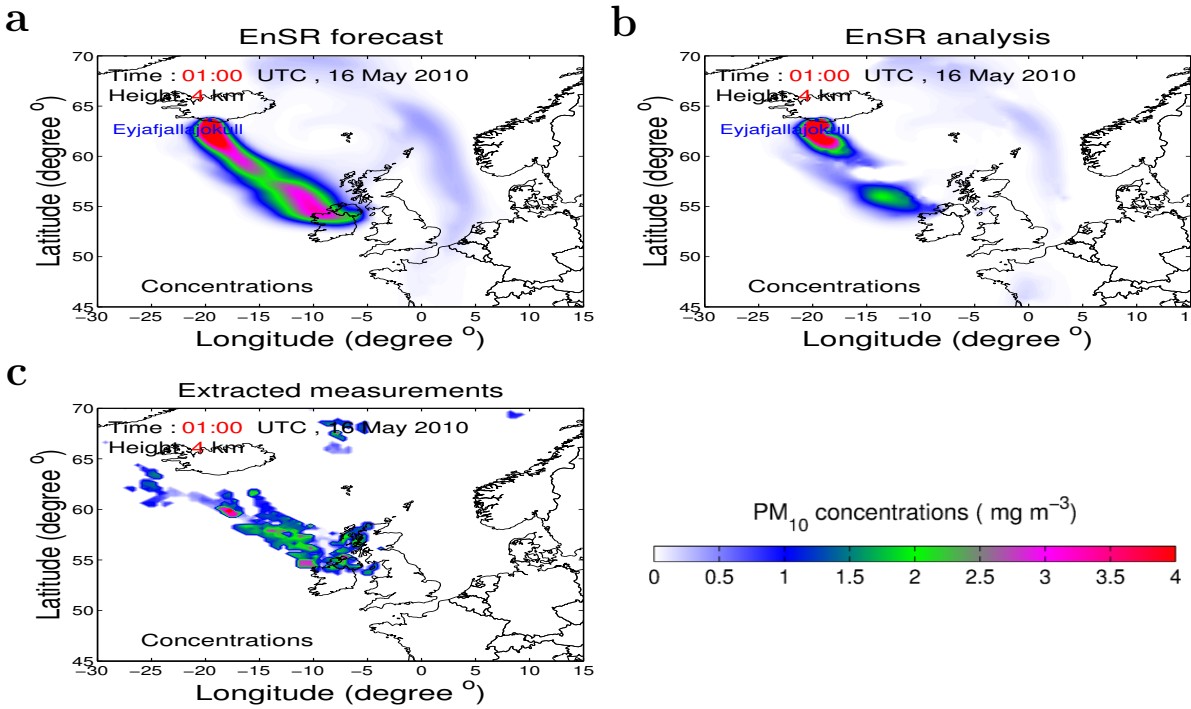

**Figure 4. Examination of EnSR effect when assimilating SEVIRI-extracted ash concentrations at 01:00 UTC 16 May 2010. a**, EnSR forecast (ensemble mean) of $PM_{10}$ concentrations. **b**, EnSR analysis (ensemble mean) of $PM_{10}$ concentrations. **c**, Extracted measurements of $PM_{10}$ concentrations from the SOO. **a**, **b**, **c** are illustrated at a height of 4 km.

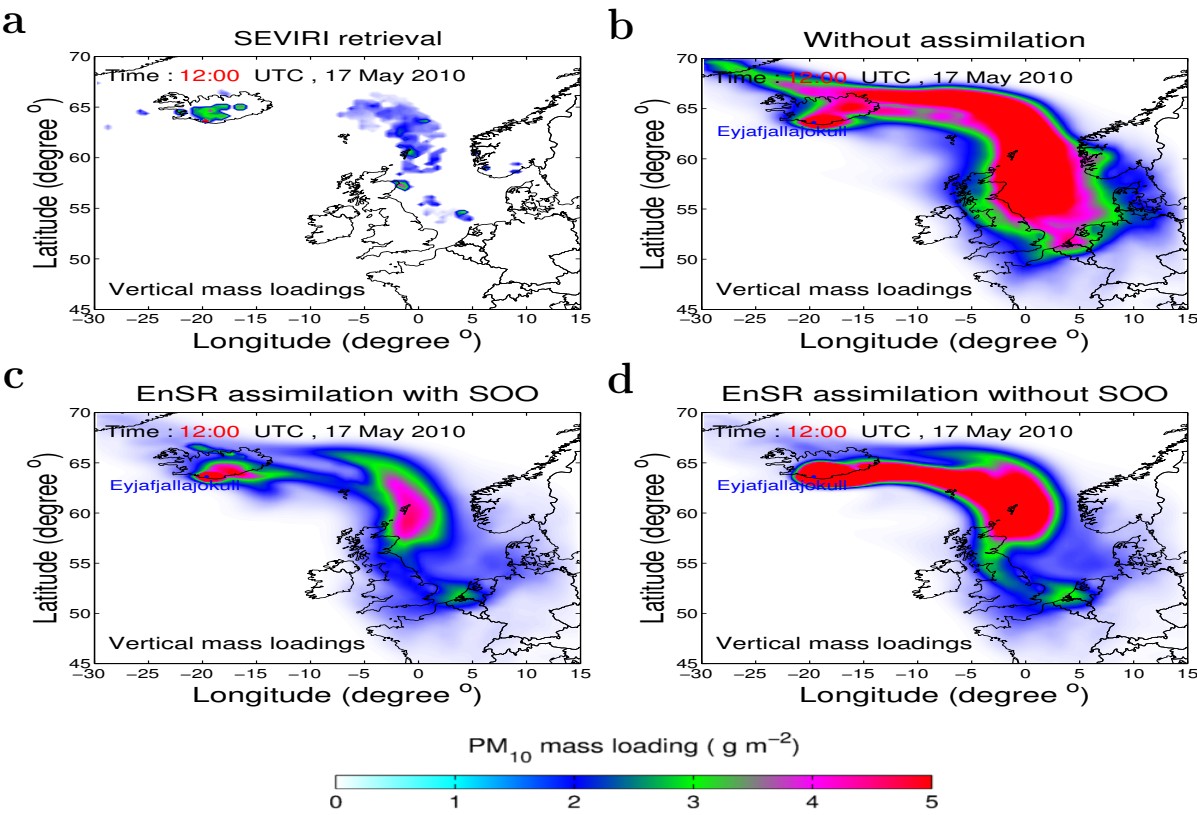

**Figure 5. PM$_{10}$ mass loadings with EnSR against the SEVIRI retrieval at 12:00 UTC 17 May 2010. a**, SEVIRI retrieved mass loadings in the vertical direction, i.e., M$_v$ in Eq. (1). **b**, Simulated mass loadings without assimilation. **c**, Mass loadings (ensemble mean) by EnSR assimilation with SOO. **d**, Mass loadings (ensemble mean) by EnSR assimilation without SOO, i.e., the operator $H_k$ in Eq. (A5) does not select the grid cell in $x(k)$, but integrates all the vertical grid cells to calculate 2D mass loadings.

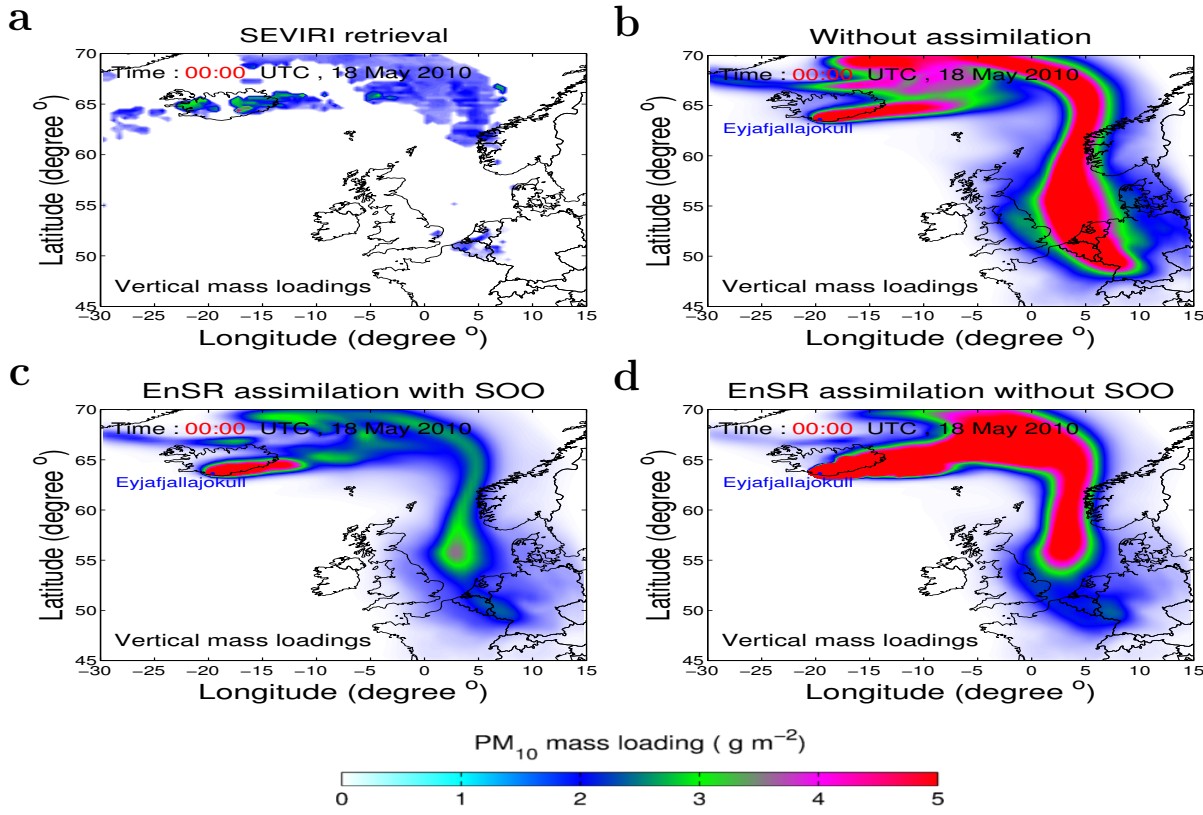

**Figure 6. PM$_{10}$ mass loadings with EnSR against the SEVIRI retrieval at 00:00 UTC 18 May 2010. a**, SEVIRI retrieved mass loadings in the vertical direction, i.e., M$_v$ in Eq. (1). **b**, Simulated mass loadings without assimilation. **c**, Mass loadings (ensemble mean) by EnSR assimilation with SOO. **d**, Mass loadings (ensemble mean) by EnSR assimilation without SOO.

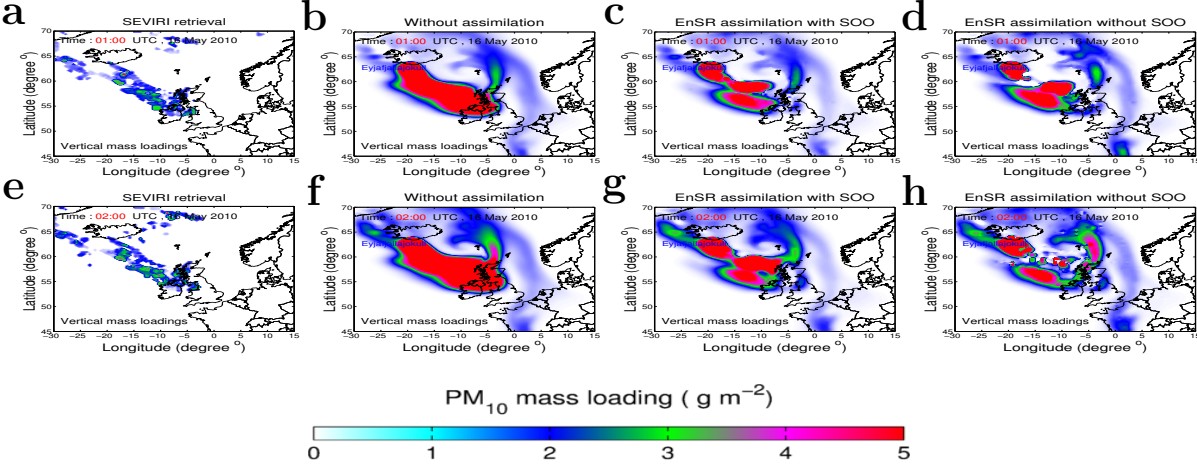

**Figure 7. Evaluation of the assimilation effects by EnSR with SOO and without SOO at 01:00/0200 UTC 16 May 2010.**

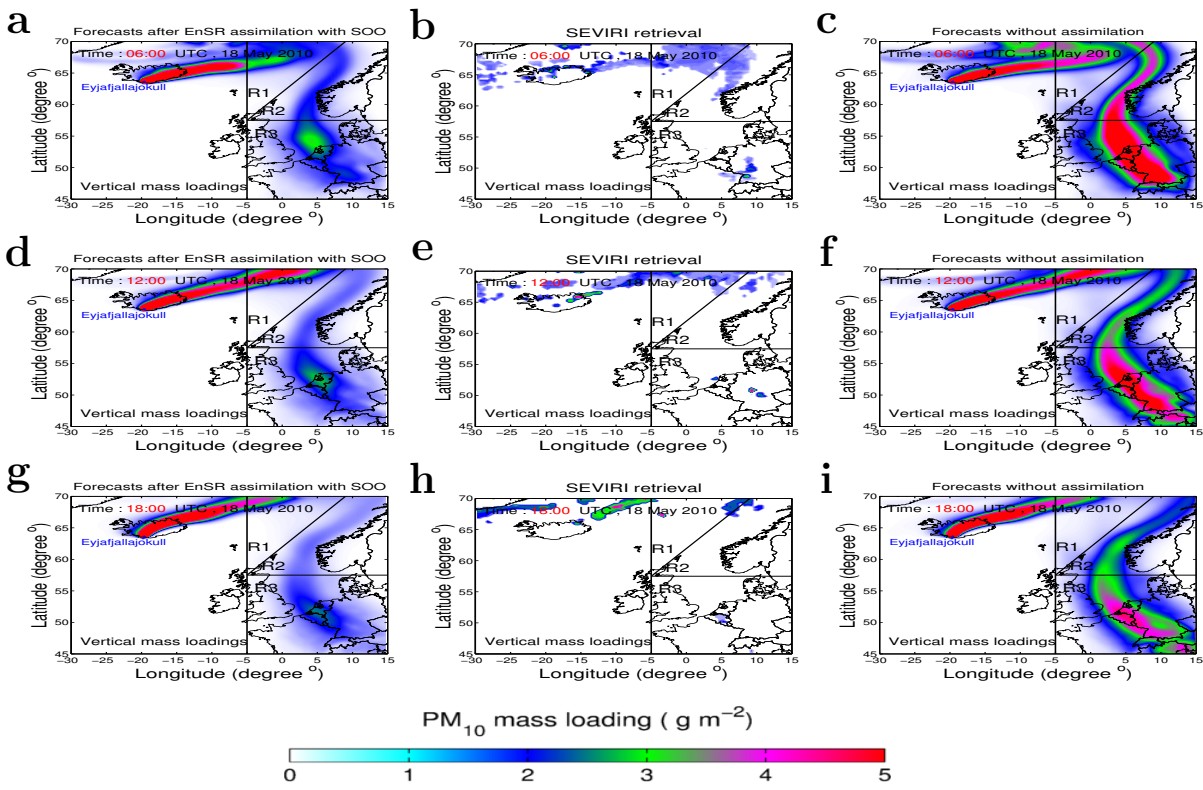

**Figure 8. Evaluation of effective assimilation forecasts using satellite retrieved mass loadings (Date: 18 May 2010). R1–R3 represent the three distal regions to be evaluated with respect to the forecast after data assimilation.**

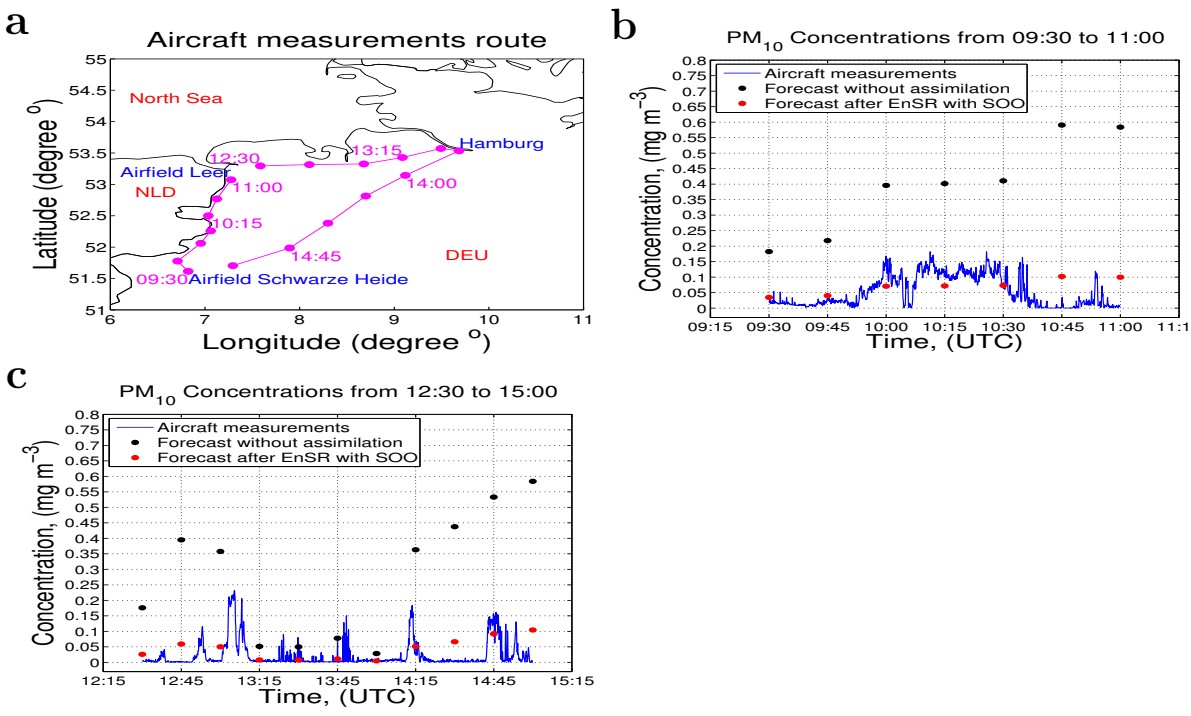

**Figure 9. Evaluation of effective assimilation forecasts using aircraft measurements (Date: 18 May 2010). a**, Aircraft measurements route. **b**, Comparisons of measurements, forecasts after assimilation (ensemble mean) or without assimilation from 09:30 to 11:00 UTC. **c**, Comparisons from 12:30 to 15:00 UTC.