# Peer review of "Data assimilation for volcanic ash plumes using a Satellite Observational Operator: a case study on the 2010 Eyjafjallajökull volcanic eruption"

_Atmospheric Chemistry and Physics, 2016_

## Referee Comment (RC2) · Anonymous Referee #2 · 12 Aug 2016

General comments:

This paper comprises nice work of the investigation of an ensemble square root filter to assimilate 2D volcanic ash mass loadings retrieved from SEVIRI satellite measurements. In detail a satellite observational operator is developed to enable the comparison between the 3D modelled ash concentrations and the 2D volcanic ash mass loading observations. Thereby, the authors focus on the measurement geometry and they define uncertainties and discuss errors in a proper and detailed way. By validation of the assimilation analysis with independent aircraft in-situ measurements the assimilation performance is evaluated and the improvements are assessed to be of benefit to aviation advice.

This study is of special interest to atmospheric science, since it applies a set of retrieved satellite remote sensing data in a new manner to a 3D assimilation system. Additionally, improvements of volcanic ash dispersion forecasts are meaningful to scientist, because better understanding of the system of the atmosphere can be achieved. To the reviewer's knowledge, the usage of an ensemble square root filter for volcanic ash assimilation is a new approach and the analysis concept of the paper is well structured and mainly clearly outlined.

But still, some analysis steps, especially the derivation of the SOO and the discussion of aviation advice need some additional revision (see specific comments below). Consequently, the conclusions should be rewritten, so that the conclusions are consistent and substantial. The choice of the figures should be reconsidered. Misspellings and minor grammatical issues do still appear in this paper and should be corrected implicitly.

Specific comments:

P 1, L 15: I query that Fu et al., 2015 is the right paper to cite at this point.

P 2, L 5: Aggregation is discussed, but isn't sedimentation the more important process. Is it included in the model calculations?

P 2, L 17-19: For remote volcanoes it is hard to perform measurement campaigns, especially as consequence of sudden eruptions. Treat such sentences carefully.

P 2, L 19-21: Don't forget about other ground based remote sensing techniques besides LIDAR.

Chap. 2: Please point out which retrieval techniques are included in the VAST-data set and which additional retrieval or processing was done by the authors of the paper. Unfortunately, data cannot be accessed, due to not working registration – but this is not the author's fault.

P 4, L 5-11: The discussion of the Marenco et al., 2011 paper seems a bit out of context here. Please specify the meaning of that publication to this paper. Maybe this

discussion could take place in Chap. 3.1.

P 4, L 10: Treat the expression of the "entire volcanic ash plume" carefully. Close to the emission source the layering of volcanic ash did not necessarily take place.

Chap. 3.1: To me it remains unclear, how the outcome of this SOO, as pre-processing to the assimilation, looks like. Is the extracted data only the data of 0.5 km layer thickness? Or does the extracted ash layer have the thickness of T_high? Or is the iterative layer thickness only used for the derivation of uncertainties?

P 4, L 27: "100 % certainty" is a dare statement. You are ignoring the retrieval errors of ash plume top height of the SEVIRI data and it might be possible to have ash layers smaller than 0.5 km thickness.

Chap. 3.2: The discussion of the ash effective particle radius seems a bit out of context. Is this even an observational parameter you are considering in your assimilation study? If not, then Fig. 1b might be of no interest to this study.

Chap. 4.2: I really like the detailed discussion of the measurement error. Nice chapter!

P 7, L 27-28: "... the overestimation has almost vanished." At this point it is not inferable that an overestimation appeared. Be careful with such assessments. Observations have large uncertainties, too. Independent data to compare would be necessary to do such judgements.

Chap. 5.2: Throughout this chapter it is important to precisely declare the regions where aviation advice can really be given. Talking about "entire Europe" while the study only takes place in the North-Western part of Europe is not acceptable. Aviation advice can only be given in areas where the assimilation took place and certainly only for the region which was additionally validated by independent observations. The note corresponding to near volcano regions and more certain advices at the end of this should be discussed already earlier. Moreover, it is discussed that the assimilation result is closer to the independent flight observations in North-West Germany, but

actually in this region the aviation advice for the forecast without assimilation would be similar, because the ash concentration in this area is far below 4 mg/m3 (see also Fig. 6d).

Chap. 6: Most parts of the conclusions chapter should be rewritten according to the changes discussed above.

P 10, L 11: "also measurements of the ash cloud thickness" -> Aren't you deriving the cloud thickness with the SOO?

Comments on figures:

Fig. 1: see comment to Chap. 3.2. Is Fig. 1b of interest to this study? If yes the choice of the colour table range should be revised.

Fig. 4: What height is the cloud top layer at 1:00 UTC, 16 May 2010? And is the cloud top layer height changing due to the EnSR? Model and observations heights might differ.

Fig. 6: To me there is no important information included in Fig. 6a. The region of interest to this study is shown also in Fig. 6b and the aircraft picture and the particle counter graphic are of no special meaning to this work. I suggest the removing of graphic 6a.

Technical corrections:

P 2, L 3: "VATDM model " -> the word model is part of the acronym

P 2, L 23: close bracket is missing

P 2, L 24: check spelling of the "atmosphere"

Chap. 2: check the spelling of "Eyjafjallajökull"

P 3, L 18: N for 70 degrees North is missing

P 3, L 21: check number of brackets

P 3, L 22 and L 24: "information of" -> information on

P 3, L 32: "The registration is needed." -> Suggestion: Registration required

P 6, L 15: "have been" -> were

P 6, L 27: "has been" -> was

P 8, L 12: "around the Netherlands" -> above / in the area of the Netherlands

P 8, L 13: "mass loadings from EnSR is" -> mass loadings from EnSR are

P 9, L 3: check comma within the date

P 9, L 4: "Düesseldorf" -> Düsseldorf

P 9, L 7: "along the Dutch border" -> be more precise: Dutch-German border
* * *

---

## Referee Comment (RC3) · Anonymous Referee #1 · 18 Aug 2016

[a4paper,10pt]article [utf8]inputenc hyperref

**1    Recommendation**

Due to major shortcomings I suggest to **reject the manuscript**.  I encourage the authors to resubmit their important and promising approach after a general revision of the study.

Here are my main concerns:

1. There is one critical assumption made in the manuscript.  The authors assume

that layers of volcanic ash are **always thicker** than 500 m (called T$_{low}$ through-out the manuscript). While this might be true for mean values, this assumption is not at all justified in general. There are papers reporting on layers which are **typically smaller than 400 m** (Prata et al, 2015), and Prata and Prata (2012) report on volcanic ash *"vertically localized in thin layers of 200 − 1000 m depth"* during the airspace closure, and to the reviewers point of view, there is no reason why layers with an extent of, e.g., 50 m should not be possible (cases with strong wind shear). Also keep in mind that in general airborne measurements are likely biased towards thicker layers (as the flight pattern is not selected randomly and thicker and more prominent layers might be preferred in flight planning). At first glance one might think that this is a minor shortcoming. Unfortunately the minimum vertical thickness plays a key role throughout the manuscript and in the concept of the Satellite Observational Operator (SOO). Therefore this issue can not be solved by a "simple" revision of the manuscript. It might require a rerun of the simulations and probably a general revision of the SOO conceptual setup.

2. The second main criticism is that the results shown in the paper are far from being sufficient to support the general conclusions. The performance of the SOO and the assimilation method is tested **only for one specific day** of the Eyjafjallajökull eruption phase. There is a plenty of other days available with volcanic ash in the European airspace, from the Eyjafjallajökull period as well as the Grimsvotn eruptions in 2011, or also the Etna, and for many of these days there are airborne measurements available (Weber et al.,2012; Marenco et al., 2011; Schumann et al., 2011). **A one day case study is not an appropriate basis for the very general conclusions** (*"... improves the forecast... ", "... significantly improves the quality of the advice ...".*)

3. The authors don't compare their results with **current** state-of-the-art VADTMs like the ones **currently implemented** in the VAACs. The only benchmark the authors take into account is their own model output without assimilation. This

is not sufficient for their general conclusions like (".. quality of advice given to aviation over continental Europe is improved").

**2  General comments**

In this section the questions listed in http://www.atmospheric-chemistry-and-physics.net/peer_review/review_criteria.html are addressed.

*Does the paper address relevant scientific questions within the scope of ACP? Does the paper present novel concepts, ideas, tools, or data?*

The manuscript describes a method to translate 2D fields of volcanic ash concentration, particle size distributions and top altitude of ash layers, as derived from satellite passive remote sensing, into three dimensional fields of the ash concentration (mass per area −> mass per volume). 3D information including vertical distribution is important, as VATDMs need 3D information of volcanic ash concentration for assimilation in order to allow accurate forecasts of volcanic ash. The SOO allows continuous assimilation of the model forecast. The authors claim that using this 3D information allows for significant improvements of the quality of volcanic ash forecasts. Therefore the paper deals with an important field of research, with high relevance to remote sensing specialists, VATDM modellers, and other key player like VAACs and NMS, and the scientific issues are within the scope of ACP. The concept is novel and worth to be investigated.

*Are substantial conclusions reached? Are the scientific methods and assumptions valid and clearly outlined? Are the results sufficient to support the interpretations and conclusions?*

[Figure]

Unfortunately the manuscript **fails to reach substantial conclusions**, as the scientific assumptions used by the authors are not appropriate; they are even wrong (namely the assumptions that volcanic ash layers are always thicker than $T_{low}$=500 m). Furthermore, the analysis only covers one specific day; therefore the results are not at all sufficient to support the very general conclusions ("..significantly improves the advice given to aviation..")

In addition, the authors only compare their own model runs (one run without and one run with assimilation). This experiment is not adequate in order to support the conclusion that their method improves the quality of advice given to aviation in Europe. For this conclusion, the authors have to compare their results with a benchmark representing the current state of the art, e.g. the VATDM standard products of the VAACs in London or Toulouse) and their assimilation scheme. Otherwise the author's general conclusion is not justified.

Furthermore, the authors only discuss the rather simple situation that there is one ash layer in the atmosphere. The state of the atmosphere is not always that easy. There are situations with water and ice clouds above/below/within the ash layer, and situations where high clouds move above the ash layer. In such situations the ash layer can't be detected by the satellite algorithms, leading to wrong assimilation input. I expect that in such a case the model run without assimilation will perform much better than the assimilated one. I completely miss the research on and the discussion of effects like these in the manuscript. Reducing the "effective time duration" from 15 h to 12 h is no adequate approach in order to account for atmospheric variability (page 9, line 16-19).

Also, the authors only discuss on cases where one singular layer of ash is present. However, there are many cases reported in the literature with several isolated volcanic ash layers in the vertical (e.g., Schumann et al., 2011). The authors do not discuss these cases and the implications of multi-layer volcanic ash for their SOO concept.

[Figure]

*Is the description of experiments and calculations sufficiently complete and precise to allow their reproduction by fellow scientists (traceability of results)? Do the authors give proper credit to related work and clearly indicate their own new/original contribution? Does the title clearly reflect the contents of the paper?*

The title is too general. I would suggest to either add ".. a case study" or to extend the analysis to much more cases if the authors want to keep the generality of their conclusions. The author should also give a better overview on the state of art regarding the assimilation procedures in other VATDMs.

*Does the abstract provide a concise and complete summary?*

Beside the criticism already mentioned, it is ok.

*Is the overall presentation well structured and clear? Is the language fluent and precise?*

In general the structure is ok, and the quality of the presentation is good. Some sentences which are not that clear are listed in the "specific comments" section. One shortcoming is that there are several typos which can easily be detected and corrected by any standard word processing software. This is not the job of a reviewer; it should be done by the authors before submission of a manuscript (e.g. "satallite","atmoshere").

*Are mathematical formulae, symbols, abbreviations, and units correctly defined and used?*

[Figure]

Yes.

*Should any parts of the paper (text, formulae, figures, tables) be clarified, reduced, combined, or eliminated?*

In general: ok. See specific comments below.

*Are the number and quality of references appropriate?*

Several references used by the authors are not appropriate and/or do not really fit the statement they should support in the manuscript. For example, the authors report on the economic damages due to closure of airspace, and they cite Bonadonna et al., 2012 in this context. However, Bonadonna et al., 2012 did not at all investigate the economic damage; they just cite a paper on the economic damage. See below for more examples.

*Is the amount and quality of supplementary material appropriate?*

N/A

**3 Specific comments:**

Page 1, L 15: Be more precise! EASA (2011) **is not the current regulation**. It is a **recommendation**. Also, the reference is outdated; please use the current version 2010-17-R7 instead of 2010-17-R4. Furthermore, the document does not state "... that the highest concentration an aircraft can endure is 4.0 mg m$^{-3}$" as claimed in

the manuscript. The reference to Fu et al. (2015) it not useful here in the context of regulation.

Page 1, L18: Prata and Prata (2012) states that eruption phase is **until May 25** 2010. Should be synchronized at least for papers with the same author.

Page 1 L 19: Bonadonna et al., 2012 is not appropriate here, cite original literature dealing with economic issues (I suggest "Oxford Economics, The economic impacts of air travel restrictions due to volcanic ash, report, 12 pp., Abbey House, Oxford, UK (2010)")

Page 2, L2: Reference (Mastin et al., 2009) is not linked to a statement; maybe it should be shifted to the end of the sentence?

Page 2, L 12: Zehner [Ed.](2010): it is not clear which statement should be supported by this reference. Zehner 2010 is a proceeding of a ESA-EUMETSAT workshop (with a plenty of participants. Please specifiy the work you refer in this context. Give either page numbers or (better) use original literature.

Page 2, L 18: I suggest to use the past: "can be performed" -> "were performed", as it better fits to the next sentence reporting on the past events.

Page 2, L18: "including occurrence of the ash plume and the nature": What does "..and the nature" mean? I thing this part of the sentence should be skipped or rephrased.

Page 2, L21: I suggest replacing "not always available" by "usually not available". At least from a global perspective. Maybe you can add the aspect of NRT availability of the data. General availability of airborne measurements, for example, does not necessarily mean that these data are available for NRT data assimilation.

Page 2, L 27: Again, Zehner (2010) is not a good reference here. Specify the algorithms you refer to, or use Prata and Prata, 2012

Page 2, L24: I strongly recommend not to use this sloppy coverage from 70 N – 70 S, 70 W – 70 E, as the visible latitude range depends on the longitude and vice versa. The field of view of a geostationary satellite is fully defined by the longitude of the sub-satellite point.

Page 3, L3: One might add that the maior "shortcoming" of CALIOP is the low "temporal resolution" (polar-orbit) and the data processing and delivery is not designed for NRT applications). It is not only the spatial coverage.

Fig 2(a): hard to interpret. What does *"where you are"* mean? Where is the volcanic ash layer? In the satellite, as indicated by the arrow? Why is the sun included in the figure?

Page 3, L7: please specify what you mean here with "entire volcanic ash plume". Specify the eruption(s) and the volcano you are talking about here, and the period of time. Marenco et al, (2011) is based on six flights between May 6th and May 22th 2010. Christopher et al. (2012) does not provide additional information on the thickness of ash layers (they just cite Marenco et al., 2011). Please also note that Marenco et al., 2011 reports on **typical ash layer depth**, one number for the whole flight (see description of table 3 in the paper)! This is something like an overall average but does not at all mean that ash layers are in general always thicker than this given value. Also look to the lidar pictures in Fig. 3 of Marenco et al. (2011) which should give you a good impression of the variability of the vertical extent of ash layers. Prata and Prata (2012) concludes that layers *"that ash was horizontally widespread, vertically localized in thin layers **of 200 – 1000 m depth"**. Prata et al.,2015 reports (1) not on the Eyjafjallajökull but the Chaiten eruption, and (2) found values of the vertical extent **clearly below 500 m**.

Page 4, L 14: "the observed values by SEVIRI": SEVIRI does not observe ash loadings, SEVIRI is a radiometer. Ash loadings are derived by algorithms applied to SEVIRI data. Please rephrase.

Page 4, L 27: T_low to T_high 0.5 – 3 km should be adapted to a realistic range. I suggest an lower limit of 50 – 100 m.

Page 4, L 5-11: One reference to Marenco et al., 2011 is sufficient; no need to cite it

three times in five lines.

Page 4, L 5-11: regarding the thickness estimation, see comments above (Page 3, L7)

Page 4, L 15: "The observed values ... ranged between 0.2 and 5.0 g/m$^2$": specify the period in time you are talking about.

Page 4, L 19: Vicente et al, 2002, is dealing with parallax corrections. You are talking about simple geometry. Did you made any parallax correction?

Page 4, L 25: I didn't find any statement in Prata and Prata (2012) that ash plumes can be considered as box-car distribution functions. In contrast, Prata and Prata (2012) states that *"because of the spatial heterogeneity of the ash, revealed in the SEVIRI retrievals, sole reliance on lidar measurements for monitoring ash clouds and for validating dispersion models may provide misleading conclusions on the concentration and homogeneity of ash in the atmosphere."* Please clearify.

Page 4, L 29: As discussed above, there is not at all a 100 % certainty that the blue layer is containing VA; T_low = 500 m is not the lowest possible thickness.

At this point I stopped evaluating the manuscript, as all the following steps are based on the assumption T$_{low}$ = 500 m.

**4 Technical comments**

Page 3, L 14: check spelling of Eyjafjalla (several times, check throughout the text)

Page 3, L 18: "N" (North) is missing

Page 3, L 30: funding information should be in the acknowledgement Page 3, L 32: "(The registration is needed)" sounds strange; I suggest (Registration required)

---

## Editor Comment (EC1) · T. von Clarmann (Editor) · 19 Aug 2016

This review replaces an earlier one which has been removed in agreement with the authors, the reviewer and the editor because it contained statements which could be understood as personal accusation.

---

## Author Comment (AC1) · 20 Sep 2016

Dear Anonymous Referee #2,

On behalf of all co-authors, first of all I would like to thank you for giving us very useful comments and suggestions. We really appreciate your detailed comments and suggestions, which certainly improve the quality of the paper. In the revision, we have carefully considered all the suggested changes ( specific comments, figures, conclusions, misspellings and grammatical issues) and included them to the best of our ability.

In the following we will give our response to every comment. To make the changes easier to identify, we have numbered them.

Best regards,
Guangliang Fu
on behalf all co-authors

(The revised manuscript is in the latter part of this pdf.)

**Reply to Specific comments**:

1. *P 1, L 15: I query that Fu et al., 2015 is the right paper to cite at this point.*

   Response: I agree. I have removed it and updated EASA2011 to the EASA2015, in line(s) 1.13–1.14:
   " and the current recommendation states that the highest concentration an aircraft can endure is 4.0 mg m$^{-3}$ (EASA, 2015). "

2. *P 2, L 5: Aggregation is discussed, but isn't sedimentation the more important process. Is it included in the model calculations?*

   Response: Yes, the sedimentation is more important than aggregation. The model includes sedimentation and deposition. I agree that the previous presentation was a bit strange. It has been re-written in line(s) 2.4–2.6:
   " A VATDM uses physical parameterizations of particle sources and removal processes (including sedimentation and deposition) that affect the concentrations in a dispersing volcanic plume. "

3. *P 2, L 17-19: For remote volcanoes it is hard to perform measurement campaigns, especially as consequence of sudden eruptions. Treat such sentences carefully.*

   Response: Yes, this is true. Thanks for the point. We have added a note in line(s) 2.21–2.23:
   " However, it should be noted that such measurements usually are not available globally and for remote volcanoes it is usually hard to perform measurement campaigns, especially as consequence of sudden eruptions. "

4. *P 2, L 19-21: Don't forget about other ground based remote sensing techniques besides LIDAR.*

Response: Yes. We include these important measurements now in line(s) 2.17–2.21:
" The measurements contained e.g., ground-based LIDAR measurements (Pappalardo et al., 2010; Flentje et al., 2010), satellite observations (Stohl et al., 2011; Prata and Prata, 2012), aircraft in situ measurements (Schumann et al., 2011; Weber et al., 2012), ground-based in situ measurements (Emeis et al., 2011), balloon measurements (Flentje et al., 2010) and ground-based remote sensing Sun photometer observations (Ansmann et al., 2010). "

5. *Chap. 2: Please point out which retrieval techniques are included in the VAST-data set and which additional retrieval or processing was done by the authors of the paper. Unfortunately, data cannot be accessed, due to not working registration – but this is not the author's fault.*

Response: Thanks for the suggestion. In the new version, this part is included in line(s) 4.15–4.21:
"

All the data shown in Fig. 1 are acquired from the European Space Agency (ESA) funded project – Volcanic Ash Strategic Initiative Team (VAST). The VAST retrieval utilizes two techniques: 1) A rudimentary cloud detection scheme implemented in the Eumetsat operational scheme caled "VOLE" (`http://navigator.eumetsat.int/discovery/Start/DirectSearch/DetailResult.do?f%28r0%29=EO:EUM:DAT:MSG:VOLE`), and 2) A more complex scheme called CID (Cloud Identification). This scheme is described in an Algorithm Theoretical Basis Document (ATBD) (unpublished but available here: (`http://vast.nilu.no/satellite-observations/`)). We have used retrievals from the CID scheme. In this study, additional processing on the retrieved data is needed to translate the data from the original SEVIRI resolution to the VATDM resolution.
"

6. *P 4, L 5-11: The discussion of the Marenco et al., 2011 paper seems a bit out of context here. Please specify the meaning of that publication to this paper. Maybe this discussion could take place in Chap. 3.1.*

Response: Yes. The whole issue of cloud thickness and independent measurements has been re-written, see: line(s) 3.9–3.26:
" For the vertical thickness information of volcanic ash clouds, Schumann et al. (2011) investigated on the 2010 Eyjafjallajökull eruption using airborne data that the volcanic ash clouds spread over large parts of Central Europe, mostly from hundreds to 3 km depth. This is consistent with the results of (Marenco et al., 2011) who observed layer depths between 0.5 and 3.0 km. Dacre et al. (2015) also examined the ground-based lidar data for the Eyjafjallajökull eruption and found a mean layer depth of 1.2±0.9 km and compared this with model based estimates of 1.1±0.8 km. Prata and Prata (2012) found variable thicknesses ranging from 0.2 up to 3 km. Recently, Clarisse and Prata (2016) reported 16 cases using ground-based lidar measurements during the Eyjafjallajökull

eruption and found 3 cases where the cloud thickness was less than 500 m. Cloud thicknesses for Kasatacho (1.01±0.43 km), Sarychev Peak (1.37±0.42 km) and Puyehue-Cordon Caulle (1.80±0.58 km) (private communication) all exceed 1 km, but Prata et al. (2015) reported lower cloud thickness with 80% of cases for the 2006 Chaiten eruption less than 400 m. The vast majority of data suggest thickness in the range 0.5–3 km, but it is entirely possible that thinner clouds (<400 m) do exist. Such clouds must have higher concentrations to be detectable by current infrared satellite techniques (Prata and Prata, 2012; Pavolonis, 2010)) that suggest a lower sensitivity in mass loading of 0.2 g m$^{-2}$. Thin ash clouds, by their nature are of less concern to aviation because such clouds would be traversed rapidly avoiding the possibility of particle build-up that might lead to engine failure. From a modeling perspective lack of vertical resolution in model wind data makes it not useful to make the cloud depth any less than 500 m.

Based on these investigations, it is not realistic to use a deterministic value to represent the overall ash cloud thickness, but we can reasonably assume that the thickness has a range of 0.5–3.0 km at the corresponding horizontal location of the SEVIRI retrieved measurements. "

7. *P 4, L 10: Treat the expression of the "entire volcanic ash plume" carefully. Close to the emission source the layering of volcanic ash did not necessarily take place.*

   Response: Yes and thank you for the reminder. We have clarified this in line(s) 3.27–3.29: " Note that we are only considering the distal plume, at least the part >100 km's from source, which is because close to the emission source the layering of volcanic ash did not necessarily take place. "

8. *Chap. 3.1: To me it remains unclear, how the outcome of this SOO, as pre-processing to the assimilation, looks like. Is the extracted data only the data of 0.5 km layer thickness? Or does the extracted ash layer have the thickness of T_high? Or is the iterative layer thickness only used for the derivation of uncertainties?*

   Response: Thanks for the question. In the new version, we have explicitly stated the outcome of SOO in line(s) 6.18–6.20:
   " The outcome of SOO can be considered as preprocessing to the satellite data assimilation system. The extracted data only represents the data at the cloud top height, which can be taken as the data within the 0.5 km layer thickness. The other layer thickness is also of high importance, which is used for the derivation of uncertainties. "

9. *P 4, L 27: "100 % certainty" is a dare statement. You are ignoring the retrieval errors of ash plume top height of the SEVIRI data and it might be possible to have ash layers smaller than 0.5 km thickness.*

   Response: Thanks for the correction. I agree that "100% certainty" is not appropriate and it has been removed in the new version.

10. *Chap. 3.2: The discussion of the ash effective particle radius seems a bit out of context. Is this even an observational parameter you are considering in your assimilation study? If*

*not, then Fig. 1b might be of no interest to this study.*

Response: I agree. The effective particle radius is not an observational parameter in assimilation. The discussion of it was indeed out of context. In the new version, the discussion and the related previous Fig. 1b have been removed.

11. *Chap. 4.2: I really like the detailed discussion of the measurement error. Nice chapter!*

Response: Thanks.

12. *P 7, L 27-28: "... the overestimation has almost vanished." At this point it is not inferable that an overestimation appeared. Be careful with such assessments. Observations have large uncertainties, too. Independent data to compare would be necessary to do such judgements.*

Response: I agree. It is not appropriate to state the "overestimation". We have re-written the related statements in line(s) 8.2-8.5:
" Comparing the state to the extracted measurements (Fig. 4c), the former (with concentrations higher than 2.0 mg m$^{-3}$ in the main plume) shows a much larger estimation compared to the latter (with concentrations mostly lower than 0.8 mg m$^{-3}$). After the EnSR analysis step (see Fig. 4b), the concentrations in large parts are now closer to the extracted measurements. "

13. *Chap. 5.2: Throughout this chapter it is important to precisely declare the regions where aviation advice can really be given. Talking about "entire Europe" while the study only takes place in the North-Western part of Europe is not acceptable. Aviation advice can only be given in areas where the assimilation took place and certainly only for the region which was additionally validated by independent observations. The note corresponding to near volcano regions and more certain advices at the end of this should be discussed already earlier. Moreover, it is discussed that the assimilation result is closer to the independent flight observations in North-West Germany, but actually in this region the aviation advice for the forecast without assimilation would be similar, because the ash concentration in this area is far below 4 mg/m3 (see also Fig. 6d).*

Response: Thanks for the comment. Yes and I agree the discussion on aviation advice in Chap. 5.2 was not sufficient or not appropriate. Recently we had also got feedback from other scientists that this part was claimed not very relevant to the paper's novel parts: SOO derivation and EnSR assimilation. I agree that the current presentation of the aviation advice was not convincing in aspect of real aviation advices, since our model LOTOS-EUROS is not the most authoritative VATDM and there are not so many independent data to validate where aviation advice can really be given.

Actually, the aviation advice in our study was not meant for real but a model-based aviation advice, which means the advice was given only depending on a chosen/fixed model. Based on this condition, the improvements of model-based aviation advice by EnSR assimilation was investigated. Certainly a better VATDM and more validation data can thus help gain

better model-based aviation advices (e.g., real aviation advice based on NAME model (in VAAC) can also be improved by EnSR).

However, the aviation advice usually is a serious issue in the sense of aviation safety. Since our study is not directly relevant on this issue but focusing on SOO and EnSR, therefore after careful discussions and considerations on the conflict between both real and model-based aviation advices, all the co-authors have agreed not to present this part (previous chapter 5.2) in the new version, but mention it in line(s) 9.19–9.22:

" Therefore, the validation test with aircraft in situ measurements shows that the regional forecasts (i.e., in the regions of North-West part of Germany) after satellite data assimilation remains valid and accurate for at least 15 hours. This is useful to provide guidance on how long a valid regional aviation advice based on the forecast after assimilation can last. "

14. *Chap. 6: Most parts of the conclusions chapter should be rewritten according to the changes discussed above.*

Response: The conclusion has been rewritten in line(s) 9.27–10.19:

" In this paper, we choose the Eyjafjallajökull volcanic ash plume in May 2010 as the study case. In this study, a satellite observational operator (SOO) was developed to translate 2D satellite ash mass loadings to 3D ash concentrations at the top layer of volcanic ash clouds. To extract ash concentrations, not only the SEVIRI data of ash mass loadings, ash cloud top height are employed, but also a reasonable assumption of the ash cloud thickness range (0.5–3 km), at the corresponding horizontal location of the SEVIRI retrieved measurements, are combined. The advantage of SOO is that it can use rough thickness information to get uncertain concentrations, which are suitable for the data assimilation methodology.

The extracted ash concentration measurements enable us to perform ensemble-based data assimilation in a 3D volcanic ash transport model. By employing a preprocessing procedure before data assimilation to generate new measurement values by averaging all surrounding measurements, the model representation error is approximately zero. The extraction error is also calculated, and the total measurement error (defined as the sum of the extraction error and the model representation error) is therefore quantified, which together with the concentrations describe the 3D measurements (mean, error) for a data assimilation system. The results showed the assimilation significantly reduces the estimation level of the conventional simulation. The accuracy of the volcanic ash state was shown to be significantly improved by the assimilation of satellite mass loadings. The good assimilation performance also verifies the suitability of the proposed SOO.

With the improved volcanic ash state as initialization, improved volcanic ash forecasts are obtained. Quantification using highly accurate aircraft in situ measurements showed that the regional forecasts after satellite data assimilation remain valid and accurate up to a half day. This effective time period probably lasts even longer and this should be further tested when more aircraft measurements are available.

In this study, we developed SOO by considering cases where one singular ash cloud is present. Actually, it could happen that there are several isolated volcanic ash clouds in the vertical direction. The methodology of SOO is also valid for these cases, where the top isolated ash cloud does not correspond to the full but to a fraction of SEVERI ash mass loadings. How to determine the reasonable proportions/percentages for multiple isolated vertical ash clouds will be investigated in future.

In this paper, we applied an off-line approach for model running and simply used the deterministic meteorological input data. Actually these data also contain uncertainties which have an influence on ash cloud transport. In future work, for more accurate ash forecasting, uncertainties in the meteorological data like wind speed should also be taken into account. "

15. *P 10, L 11: "also measurements of the ash cloud thickness" -> Aren't you deriving the cloud thickness with the SOO?*

    Response: Thanks for the question. It has been explicitly stated in line(s) 9.29–9.31:
    " To extract ash concentrations, not only the SEVIRI data of ash mass loadings, ash cloud top height are employed, but also a reasonable assumption of the ash cloud thickness range (0.5–3 km), at the corresponding horizontal location of the SEVIRI retrieved measurements, are combined. "

**Reply to Comments on figures**:

1. *Fig. 1: see comment to Chap. 3.2. Is Fig. 1b of interest to this study? If yes the choice of the colour table range should be revised.*

    Response: As replied to comment to Chap. 3.2, the previous Fig. 1b has been removed.

2. *Fig. 4: What height is the cloud top layer at 1:00 UTC, 16 May 2010? And is the cloud top layer height changing due to the EnSR? Model and observations heights might differ.*

    Response: The height of the cloud top layer is added in the new version as Fig. 4**d**. Comparing Fig. 4**d** to Fig. 2**b**, the height is changing at different time. The simulation results shown in Fig. 4**a** and Fig. 4**b** are not at a constant height but at the height of the cloud top layer (Fig. 4**d**). It is now stated in the caption of Fig. 4 :
    " Ash concentrations shown in **a**, **b**, **c** are at the ash cloud top layer height **d**. "

3. *Fig. 6: To me there is no important information included in Fig. 6a. The region of interest to this study is shown also in Fig. 6b and the aircraft picture and the particle counter graphic are of no special meaning to this work. I suggest the removing of graphic 6a.*

    Response: I agree. As suggested, the previous Fig. 6**a** has been removed.

**Reply to Technical corrections**:

1. *P 2, L 3: "VATDM model " -> the word model is part of the acronym*

    Response: It is corrected in line(s) 2.4:
    " are necessary as inputs to the VATDM (Mastin et al., 2009). "

2. *P 2, L 23: close bracket is missing*

   Response: It is corrected in line(s) 2.25:
   " the Spin Enhanced Visible and Infrared Imager  (SEVIRI),  "

3. *P 2, L 24: check spelling of the "atmosphere"*

   Response: It is corrected in line(s) 2.26:
   " the  atmosphere  and earth's surface "

4. *Chap. 2: check the spelling of "Eyjafjallajökull"*

   Response: Thanks for the correction. All the typo errors of "Eyjafjallajökull" are now
   corrected in the whole paper.

5. *P 3, L 18: N for 70 degrees North is missing*

   Response: in line(s) 4.5:
   " A region covering 30° W to 15° E and 45° N to 70°  N  is selected here for analysis "

6. *P 3, L 21: check number of brackets*

   Response: The extra bracket is now corrected in line(s) 4.7-4.8:
   " The main retrieval products from SEVIRI are ash mass loadings (Prata and Prata, 2012;
   Kylling et al., 2015)  (see Fig. 1a, value at 0 means no data)  where 03:15 UTC 16 May
   2010 is chosen for the illustration, without loss of generality. "

7. *P 3, L 22 and L 24: "information of" -> information on*

   Response: It is corrected in line(s) 4.9:
   " The mass loading at each 2D pixel gives information  on  the ash cloud from the top view
   (Prata and Prata, 2012), which can be taken as an integration of ash concentrations along
   the retrieval path. "

8. *P 3, L 32: "The registration is needed." -> Suggestion: Registration required*

   Response: As suggested, it is changed in line(s) 10.22:
   "  (Registration required).   "

9. *P 6, L 15: "have been" -> were*

Response: in line(s) 6.23:
" many other algorithms  were  developed "

10.  *P 6, L 27: "has been" -> was*

Response: in line(s) 7.6:
" the LOTOS-EUROS as a proper volcanic ash transport model  was  reported "

11.  *P 8, L 12: "around the Netherlands" -> above / in the area of the Netherlands*

12.  *P 8, L 13:"mass loadings from EnSR is" -> mass loadings from EnSR are*

Response: As suggested, both are changed in line(s) 8.22–8.23:
" For example,  in the area of the Netherlands,  the mass loadings from EnSR  are  accumulated to 2.9 – 3.2 g m$^{-2}$, "

13.  *P 9, L 3: check comma within the date*

14.  *P 9, L 4: "Düesseldorf" -> Düsseldorf*

Response: The comma is now deleted in line(s) 9.8–9.9:
" Fortunately, some aircraft measurements on  18 May 2010  from 09:30 to 15:30 UTC are available, which were performed by the group Environmental Measurement Techniques at  Düsseldorf  university of Applied Sciences. "

15.  *P 9, L 7: "along the Dutch border" -> be more precise: Dutch-German border*

Response: As suggested, it is changed in line(s) 9.11:

[revised manuscript text omitted]

$$\text{ML}_{\text{vert}} = \text{ML} \times \cos(\text{VZA}). \tag{1}$$

To extract ash concentrations from SEVIRI retrievals, $\text{ML}_{\text{vert}}$ only is not sufficient and knowledge about the vertical distri-
5   bution of ash cloud must be included. The cloud vertical profile can be described with the height of the top and the thickness of the cloud. As introduced in Section 2, the cloud top height ($\text{H}_{\text{top}}$) is available from satellite remote sensing and the thickness of the plume is investigated ($\text{T}_{\text{low}}$ to $\text{T}_{\text{high}}$, i.e., 0.5 to 3 km). Fig. 2 illustrates how the 3D ash concentrations are extracted from the obtained mass loadings in the vertical direction ($\text{ML}_{\text{vert}}$). The blue layer in Fig. 2 is determined by the lowest possible thickness ($\text{T}_{\text{low}}$) and the extraction layer used in this study only refers to the blue layer.

10   When the top height and the thickness range of ash cloud are known, the ash concentration (C) in the extraction layer can be calculated by using the ash mass loadings ($\text{ML}_{\text{vert}}$) at the corresponding horizontal location. The details are formulated as follows. First we define

$$N_s = \lceil \frac{\text{T}_{\text{high}} - \text{T}_{\text{low}}}{\text{T}} \rceil \quad , \tag{2}$$

$$\text{T}_i = \text{T}_{\text{low}} + (i-1) \times \text{T}, \quad \text{C}_i = \frac{\text{ML}_{\text{vert}}}{\text{T}_i}, \qquad i = 1, 2, \cdots, N_s \quad , \tag{3}$$

15   where T is a step length and $N_s$ is the number of the possible thickness. $\text{T}_{\text{low}}$ represents the blue layer (see Fig. 2) with the fixed thickness of 0.5 km and $\text{T}_{\text{high}} - \text{T}_{\text{low}}$ represents the yellow layer with the fixed thickness of 2.5 km. T is chosen at a small value compared to $\text{T}_{\text{low}}$, which guarantees $N_s$ is not too small (e.g., less than 2) to sample enough thickness $\text{T}_1, \text{T}_2, \cdots,$ $\text{T}_{N_s}$ with equal probability. (e.g., T is chosen as 0.05 km in this case study, thus $N_s$ is calculated as 50.)

Corresponding to the sampled thickness, the ash concentration can be calculated as also a sample from $\text{C}_1$ to $\text{C}_{N_s}$, as shown
20   in Eq. (3). Therefore, the mean ($\text{C}_{\text{mean}}$) and the standard deviation ($\text{C}_{\text{std}}$) of the sampled ash concentrations can be calculated by Eq. (4) and (5),

$$\text{C}_{\text{mean}} = \frac{1}{N_s}(\text{C}_1 + \text{C}_2 + \cdots + \text{C}_{N_s}) \quad , \tag{4}$$

$$\text{C}_{\text{std}} = \sqrt{\frac{1}{N_s - 1}[(\text{C}_1 - \text{C}_{\text{mean}})^2 + (\text{C}_2 - \text{C}_{\text{mean}})^2 + \cdots + (\text{C}_{N_s} - \text{C}_{\text{mean}})^2]} \quad . \tag{5}$$

$\text{C}_{\text{mean}}$ is therefore used in this study as the extracted concentration C between the heights [$\text{H}_{\text{top}}$ - $\text{T}_{\text{low}}$] and $\text{H}_{\text{top}}$ (i.e., the
25   blue layer in Fig. 2). How much of the mass is distributed to the blue layer ($\text{ML}_{\text{blue}}$) can be calculated by Eq. (6),

$$\text{ML}_{\text{blue}} = \text{C}_{\text{mean}} \times \text{T}_{\text{low}}. \tag{6}$$

[revised manuscript text omitted]

---

## Author Comment (AC2) · 26 Sep 2016

Dear Anonymous Reviewer2,

For the question "P 2, L 19-21: Don't forget about other ground based remote sensing techniques besides LIDAR", in our previous answer, ceilometers (automated one-wavelength backscatter lidars), as one important type of measurements were not mentioned, which are available as networks (the German weather service currently implemented almost 100 systems).

We include these important measurements in the paper now:
" The measurements contained e.g., ground-based LIDAR and ceilometer measurements (Pappalardo et al., 2010; Wiegner et al., 2012), satellite observations (Stohl et al., 2011; Prata and Prata, 2012), aircraft in situ measurements (Schumann et al.,

2011; Weber et al., 2012), ground-based in situ measurements (Emeis et al., 2011), balloon measurements (Flentje et al., 2010) and ground-based remote sensing Sun photometer observations (Ansmann et al., 2010). "

Best wishes, Guangliang on behalf of all co-authors

---

## Author Comment (AC3) · 5 Oct 2016

**Responses – part 2:**

**Main concerns**:

1. *There is one critical assumption made in the manuscript. The authors assume that layers of volcanic ash are always thicker than 500 m (called $T_{low}$ throughout the manuscript). While this might be true for mean values, this assumption is not at all justified in general. There are papers reporting on layers which are typically smaller than 400 m (Prata et al, 2015), and Prata and Prata (2012) report on volcanic ash "vertically localized in thin layers of 200 – 1000 m depth" during the airspace closure, and to the reviewers point of view, there is no reason why layers with an extent of, e.g., 50 m should not be possible (cases with strong wind shear). Also keep in mind that in general airborne measurements are likely biased towards thicker layers (as the flight pattern is not selected randomly and thicker and more prominent layers might be preferred in flight planning). At first glance one might think that this is a minor shortcoming. Unfortunately the minimum vertical thickness plays a key role throughout the manuscript and in the concept of the Satellite Observational Operator (SOO). Therefore this issue can not be solved by a "simple" revision of the manuscript. It might require a rerun of the simulations and probably a general revision of the SOO conceptual setup.*

   Response: We have reconsidered this issue thoroughly and carefully read our own papers and those of other reaserchers and find that the most reasonable estimate of ash layer thickness is 0.5–3 km. We included a reference to Prata et al. (2015). The response for this major question is in line(s) 3.9–3.29:
   " For the vertical thickness information of volcanic ash clouds, Schumann et al. (2011) investigated on the 2010 Eyjafjallajökull eruption using airborne data that the volcanic ash clouds spread over large parts of Central Europe, mostly from hundreds to 3 km depth. This is consistent with the results of (Marenco et al., 2011) who observed layer depths between 0.5 and 3.0 km. Dacre et al. (2015) also examined the ground-based lidar data for the Eyjafjallajökull eruption and found a mean layer depth of 1.2±0.9 km and compared this with model based estimates of 1.1±0.8 km. Prata and Prata (2012) found variable thicknesses ranging from 0.2 up to 3 km. Recently, Clarisse and Prata (2016) reported 16 cases using ground-based lidar measurements during the Eyjafjallajökull eruption and found 3 cases where the cloud thickness was less than 500 m. Cloud thicknesses for Kasatacho (1.01±0.43 km), Sarychev Peak (1.37±0.42 km) and Puyehue-Cordon Caulle (1.80±0.58 km) (private communication) all exceed 1 km, but Prata et al. (2015) reported lower cloud thickness with 80% of cases for the 2006 Chaiten eruption less than 400 m. The vast majority of data suggest thickness in the range 0.5–3 km, but it is entirely possible that thinner clouds (<400 m) do exist. Such clouds must have higher concentrations to be detectable by current infrared satellite techniques (Prata and Prata, 2012; Pavolonis, 2010)) that suggest a lower sensitivity in mass loading of 0.2 g m$^{-2}$. Thin ash clouds, by their nature are of less concern to aviation because such clouds would be traversed rapidly avoiding the possibility of particle build-up that might lead to engine failure. From a modeling perspective lack of vertical resolution in model wind data makes it not useful to make the cloud depth any less than 500 m.

   Based on these investigations, it is not realistic to use a deterministic value to represent the

overall ash cloud thickness, but we can reasonably assume that the thickness has a range of 0.5–3.0 km at the corresponding horizontal location of the SEVIRI retrieved measurements. Although this thickness information is not deterministic, its uncertainty spread is suitable in an observational operator for satellite data assimilation. Note that we are only considering the distal plume, at least the part >100 km's from source, which is because close to the emission source the layering of volcanic ash did not necessarily take place. "

2. *The second main criticism is that the results shown in the paper are far from being sufficient to support the general conclusions. The performance of the SOO and the assimilation method is tested only for one specific day of the Eyjafjallajökull eruption phase. There is a plenty of other days available with volcanic ash in the European airspace, from the Eyjafjallajökull period as well as the Grimsvotn eruptions in 2011, or also the Etna, and for many of these days there are airborne measurements available (Weber et al.,2012; Marenco et al., 2011; Schumann et al., 2011). A one day case study is not an appropriate basis for the very general conclusions ("... improves the forecast... ", "... significantly improves the quality of the advice ...".)*

Response: The question was asked why other events and other volcanoes are not considered, such as Grimsvotn eruptions in 2011, or the Etna. However, we have mentioned "for this case study" at many places in the discussion paper, which means we only focus on the 2010 Eyjafjallajökull volcanic eruption. To make it clear, the title has been changed to Satellite data assimilation to improve forecasts of volcanic ash concentrations: a case study on the 2010 Eyjafjallajökull volcanic ash plume . The conclusion is also made more clear by explicitly stating in line(s) 9.27:
" In this paper, we choose the Eyjafjallajökull volcanic ash plume in May 2010 as the study case. I believe that this clarifies that our conclusion is not that general as stated in the comment. "

3. *The authors don't compare their results with current state-of-the-art VADTMs like the ones currently implemented in the VAACs. The only benchmark the authors take into account is their own model output without assimilation. This is not sufficient for their general conclusions like (".. quality of advice given to aviation over continental Europe is improved").*

Response: The aviation advice in our study was not meant for real but a model-based aviation advice, which means the advice was given only depending on a chosen/fixed model. Based on this condition, the improvements of model-based aviation advice by EnSR assimilation was investigated. Certainly a better VATDM and more validation data can thus help gain better model-based aviation advices (e.g., real aviation advice based on NAME model (in VAAC) can also be improved by EnSR).

However, the aviation advice usually is a serious issue in the sense of aviation safety. Since our study is not directly focusing on this issue but on SOO and EnSR, therefore after careful considerations on the conflict between both real and model-based aviation advices, all the co-authors have agreed not to present this part (previous chapter 5.2) in the new version, but mention it in line(s) 9.19–9.22:
" Therefore, the validation test with aircraft in situ measurements shows that the regional forecasts (i.e., in the regions of North-West part of Germany) after satellite data assimilation remains valid and accurate for at least 15 hours. This is an important indication about

how long a valid regional aviation advice based on the forecast after assimilation can last. "

**Specific comments**:

1. *Page 1, L 15: Be more precise! EASA (2011) is not the current regulation. It is a recommendation. Also, the reference is outdated; please use the current version 2010-17-R7 instead of 2010-17-R4. Furthermore, the document does not state ?... that the highest concentration an aircraft can endure is 4.0 mg $m^{-3}$ as claimed in the manuscript. The reference to Fu et al. (2015) it not useful here in the context of regulation.*

   Response: It is changed in line(s) 1.13–1.14:
   " and the current recommendation states that the highest concentration an aircraft can endure is 4.0 mg m$^{-3}$ (EASA, 2015). "

2. *Page 1, L18: Prata and Prata (2012) states that eruption phase is until May 25 2010. Should be synchronized at least for papers with the same author.*

   Response: It is synchronized in line(s) 1.16–1.17:
   " from 14 April to 25 May 2010, "

3. *Page 1 L 19: Bonadonna et al., 2012 is not appropriate here, cite original literature dealing with economic issues (I suggest "Oxford Economics, The economic impacts of air travel restrictions due to volcanic ash, report, 12 pp., Abbey House, Oxford, UK (2010)")*

   Response: It is changed in line(s) 1.18:
   " 5 billion US dollars (Oxford-Economics, 2010). "

4. *Page 2, L2: Reference (Mastin et al., 2009) is not linked to a statement; maybe it should be shifted to the end of the sentence?*

   Response: It is changed in line(s) 2.2–2.4:
   " The meteorological wind fields and estimates of eruption source parameters (ESPs) such as plume height (PH), mass eruption rate (MER), particle size distribution (PSD) and vertical mass distribution (VMD) are necessary as inputs to the VATDM (Mastin et al., 2009). "

5. *Page 2, L 12: Zehner [Ed.](2010): it is not clear which statement should be supported by this reference. Zehner 2010 is a proceeding of a ESA-EUMETSAT workshop (with a plenty of participants. Please specifiy the work you refer in this context. Give either page numbers or (better) use original literature.*

Response: Zehner [Ed.](2010) is changed to (Evensen, 2003; Bocquet et al., 2015; Fu et al., 2015).. in line(s) 2.12:

" Data assimilation, which refers to the (quasi-) continuous use of the direct measurements to create accurate initial conditions for model runs, is one of the most commonly used approaches for real-time forecasting problems (Evensen, 2003; Bocquet et al., 2015; Fu et al., 2015). "

6. *Page 2, L 18: I suggest to use the past: "can be performed" -> "were performed", as it better fits to the next sentence reporting on the past events.*

   Response: Corrected.

7. *Page 2, L18: "including occurrence of the ash plume and the nature": What does "..and the nature" mean? I thing this part of the sentence should be skipped or rephrased.*

   Response: It is rephrased in line(s) 2.16–2.18:
   " Fortunately, during volcanic ash transport, different types of scientific measurement campaigns were performed to collect information of the ash plume. "

8. *Page 2, L21: I suggest replacing "not always available" by "usually not available". At least from a global perspective. Maybe you can add the aspect of NRT availability of the data. General availability of airborne measurements, for example, does not necessarily mean that these data are available for NRT data assimilation.*

   Response: As suggested, it is replaced in line(s) 2.22–2.23:
   " However, it should be noted that such measurements usually are not available globally and for remote volcanoes it is usually hard to perform measurement campaigns, especially as consequence of sudden eruptions. "

9. *Page 2, L 27: Again, Zehner (2010) is not a good reference here. Specify the algorithms you refer to, or use Prata and Prata, 2012.*

   Response: Zehner (2010) is changed to (Prata and Prata, 2012).

10. *Page 2, L24: I strongly recommend not to use this sloppy coverage from 70 N – 70 S, 70 W – 70 E, as the visible latitude range depends on the longitude and vice versa. The field of view of a geostationary satellite is fully defined by the longitude of the sub-satellite point.*

    Response: It is changed in line(s) 2.25–2.26:
    " on board the Meteosat Second Generation (MSG) platform provides a large view coverage of the atmosphere and earth's surface (Schmetz et al., 2002). "

11. *Page 3, L3: One might add that the major "shortcoming" of CALIOP is the low "temporal resolution" (polar-orbit) and the data processing and delivery is not designed for NRT*

*applications). It is not only the spatial coverage.*

Response: It is added in line(s) 3.6–3.7:
" but the measurements are spatially sparse and have low temporal resolution (polar-orbit) and the data processing and delivery is not designed for near real-time applications.  "

12. *Fig 2(a): hard to interpret. What does "where you are" mean? Where is the volcanic ash layer? In the satellite, as indicated by the arrow? Why is the sun included in the figure?*

Response: Yes, Fig 2(a) is hard to interpret. It is not very relevant to the paper and so the Fig 2(a) is deleted in the revision.

13. *Page 3, L7: please specify what you mean here with "entire volcanic ash plume". Specify the eruption(s) and the volcano you are talking about here, and the period of time. Marenco et al, (2011) is based on six flights between May 6th and May 22th 2010. Christopher et al. (2012) does not provide additional information on the thickness of ash layers (they just cite Marenco et al., 2011). Please also note that Marenco et al., 2011 reports on typical ash layer depth, one number for the whole flight (see description of table 3 in the paper)! This is something like an overall average but does not at all mean that ash layers are in general always thicker than this given value. Also look to the lidar pictures in Fig. 3 of Marenco et al. (2011) which should give you a good impression of the variability of the vertical extent of ash layers. Prata and Prata (2012) concludes that layers "that ash was horizontally widespread, vertically localized in thin layers of 200 – 1000 m depth". Prata et al.,2015 reports (1) not on the Eyjafjallajökull but the Chaiten eruption, and (2) found values of the vertical extent clearly below 500 m.*

Response: This issue is addressed and explained in the Response to Main Concern 1.

14. *Page 4, L 14: "the observed values by SEVIRI": SEVIRI does not observe ash loadings, SEVIRI is a radiometer. Ash loadings are derived by algorithms applied to SEVIRI data. Please rephrase.*

Response: It is rephrased in line(s) 4.28:
" The retrieved values by SEVIRI  "

15. *Page 4, L 27: $T_{low}$ to $T_{high}$ 0.5 – 3 km should be adapted to a realistic range. I suggest an lower limit of 50 – 100 m.*

Response: This issue is answered in the Response to Main Concern 1.

16. *Page 4, L 5-11: One reference to Marenco et al., 2011 is sufficient; no need to cite it three times in five lines.*

Response: Changed.

17. *Page 4, L 5-11: regarding the thickness estimation, see comments above (Page 3, L7)*

    Response: It is answered in the Response to Main Concern 1.

18. *Page 4, L 15: "The observed values ... ranged between 0.2 and 5.0 g $m^{-2}$": specify the period in time you are talking about.*

    Response: "ranged between 0.2 and 5.0 g $m^{-2}$" is not relevant to this paper and is deleted in the revision. Thus, in the revision the time period is not specified in this place.

19. *Page 4, L 19: Vicente et al, 2002, is dealing with parallax corrections. You are talking about simple geometry. Did you made any parallax correction?*

    Response: The reference is not suitable. it is changed in line(s) 4.32–5.1:
    " The VZA for each pixel is computed according to the satellite VZA algorithms (Gieske et al., 2005) by using general parameters (such as longitude, latitude of each pixel).   "

20. *Page 4, L 25: I didn't find any statement in Prata and Prata (2012) that ash plumes can be considered as box-car distribution functions. In contrast, Prata and Prata (2012) states that "because of the spatial heterogeneity of the ash, revealed in the SEVIRI retrievals, sole reliance on lidar measurements for monitoring ash clouds and for validating dispersion models may provide misleading conclusions on the concentration and homogeneity of ash in the atmosphere." Please clearify.*

    Response: The consideration of the "box-car" distribution here is assumed only vertically for a specified location of the SEVIRI retrieved 2D mass loadings. This consideration is based on the layering property of ash cloud (i.e., ashes are concentrated in one or several layers). While, "the spatial heterogeneity of the ash, revealed in the SEVIRI retrievals" is meant mostly in a horizontal sense. These are different concepts and there are no conflict between both. The sentense of "because of the spatial heterogeneity of the ash $\cdots$ homogeneity of ash in the atmosphere" is to provide some recommendations and state the significance of sparse measurements (e.g., measurements from ground-based lidars or from airborne platforms or from measurement sites).

    The full related paragragh in Prata and Prata (2012) is:
    " The full spatial extent of the ash from Eyjafjallajökull can be clearly discerned in the SEVIRI retrievals. Sparse measurements of ash from ground-based lidars, from infrequent airborne platforms and from fixed measurement sites must be used with caution, and in conjunction with other data or model outputs in order to properly assess the extent and magnitude of ash clouds. Because of the spatial heterogeneity of the ash, revealed in the SEVIRI retrievals, sole reliance on lidar measurements for monitoring ash clouds and for validating dispersion models may provide misleading conclusions on the concentration and homogeneity of ash in the atmosphere. However, these other types of measurements are vital for independent validation of satellite retrievals. "

    To avoid misunderstanding, we have removed the sentense (including "box-car" function) which is not very relevant to the paper.

21. *Page 4, L 29: As discussed above, there is not at all a 100 % certainty that the blue layer is containing VA; $T_{low} = 500$ m is not the lowest possible thickness.*

   Response: 100 % is not suitable and is deleted in the revision.

**Technical comments**:

1. *Page 3, L 14: check spelling of Eyjafjalla (several times, check throughout the text)*

   Response: Corrected.

2. *Page 3, L 18: "N" (North) is missing*

   Response: Corrected.

3. *Page 3, L 30: funding information should be in the acknowledgement Page 3, L 32: "(The registration is needed)" sounds strange; I suggest (Registration required)*

   Response: Changed.

The revised manuscript starts from next page.

[revised manuscript text omitted]

$$\mathrm{ML}_{\mathrm{vert}} = \mathrm{ML} \times \cos(\mathrm{VZA}). \tag{1}$$

To extract ash concentrations from SEVIRI retrievals, $\mathrm{ML}_{\mathrm{vert}}$ only is not sufficient and knowledge about the vertical distri-
5   bution of ash cloud must be included. The cloud vertical profile can be described with the height of the top and the thickness of the cloud. As introduced in Section 2, the cloud top height ($\mathrm{H}_{\mathrm{top}}$) is available from satellite remote sensing and the thickness of the plume is investigated ($\mathrm{T}_{\mathrm{low}}$ to $\mathrm{T}_{\mathrm{high}}$, i.e., 0.5 to 3 km). Fig. 2 illustrates how the 3D ash concentrations are extracted from the obtained mass loadings in the vertical direction ($\mathrm{ML}_{\mathrm{vert}}$). The blue layer in Fig. 2 is determined by the lowest possible thickness ($\mathrm{T}_{\mathrm{low}}$) and the extraction layer used in this study only refers to the blue layer.

10   When the top height and the thickness range of ash cloud are known, the ash concentration (C) in the extraction layer can be calculated by using the ash mass loadings ($\mathrm{ML}_{\mathrm{vert}}$) at the corresponding horizontal location. The details are formulated as follows. First we define

$$N_s = \left\lceil \frac{\mathrm{T}_{\mathrm{high}} - \mathrm{T}_{\mathrm{low}}}{\mathrm{T}} \right\rceil \quad , \tag{2}$$

$$\mathrm{T}_i = \mathrm{T}_{\mathrm{low}} + (i-1) \times \mathrm{T}, \quad \mathrm{C}_i = \frac{\mathrm{ML}_{\mathrm{vert}}}{\mathrm{T}_i}, \qquad i = 1, 2, \cdots, N_s \quad , \tag{3}$$

15   where T is a step length and $N_s$ is the number of the possible thickness. $\mathrm{T}_{\mathrm{low}}$ represents the blue layer (see Fig. 2) with the fixed thickness of 0.5 km and $\mathrm{T}_{\mathrm{high}} - \mathrm{T}_{\mathrm{low}}$ represents the yellow layer with the fixed thickness of 2.5 km. T is chosen at a small value compared to $\mathrm{T}_{\mathrm{low}}$, which guarantees $N_s$ is not too small (e.g., less than 2) to sample sufficient number of thickness $\mathrm{T}_1$, $\mathrm{T}_2$, $\cdots$, $\mathrm{T}_{N_s}$ with equal probability. (e.g., T is chosen as 0.05 km in this case study, thus $N_s$ is calculated as 50.)

Corresponding to the sampled thickness, the ash concentration can be calculated also as a sample from $\mathrm{C}_1$ to $\mathrm{C}_{N_s}$, as shown
20   in Eq. (3). Therefore, the mean ($\mathrm{C}_{\mathrm{mean}}$) and the standard deviation ($\mathrm{C}_{\mathrm{std}}$) of the sampled ash concentrations can be calculated by Eq. (4) and (5),

$$\mathrm{C}_{\mathrm{mean}} = \frac{1}{N_s}(\mathrm{C}_1 + \mathrm{C}_2 + \cdots + \mathrm{C}_{N_s}) \quad , \tag{4}$$

$$\mathrm{C}_{\mathrm{std}} = \sqrt{\frac{1}{N_s - 1}[(\mathrm{C}_1 - \mathrm{C}_{\mathrm{mean}})^2 + (\mathrm{C}_2 - \mathrm{C}_{\mathrm{mean}})^2 + \cdots + (\mathrm{C}_{N_s} - \mathrm{C}_{\mathrm{mean}})^2]} \quad . \tag{5}$$

$\mathrm{C}_{\mathrm{mean}}$ is therefore used in this study as the extracted concentration C between the heights [$\mathrm{H}_{\mathrm{top}}$ - $\mathrm{T}_{\mathrm{low}}$] and $\mathrm{H}_{\mathrm{top}}$ (i.e., the
25   blue layer in Fig. 2). How much of the mass is distributed to the blue layer ($\mathrm{ML}_{\mathrm{blue}}$) can be calculated by Eq. (6),

$$\mathrm{ML}_{\mathrm{blue}} = \mathrm{C}_{\mathrm{mean}} \times \mathrm{T}_{\mathrm{low}}. \tag{6}$$

[revised manuscript text omitted]

---

## Referee Report (RR1)

**1 Recommendation**

The authors have made a very substantial revision which significantly improved the quality of the manuscript. The title and conclusions are now in a better agreement with the content of the paper and almost all points addressed in the review were considered.

Therefore I now recommend to accepted the manuscript subject to revisions as listed below:

**2 Specific comments**

- As already pointed out in my former review, the authors discuss the situation that there is an ash layer in clear-sky atmosphere. However, in general there are water and ice clouds above/below/within the ash layer, and situations where clouds move above the ash layer. In such situations the ash layer can't be detected by the satellite algorithms (e.g., Prata and Prata, 2012), leading to wrong assimilation input. I expect that in such cases the model run without assimilation will perform much better than the assimilated one. I suggest to (1) add a comment on that in the paper, and (2) discuss how these errors are treated in your equation (8).

- P 2, L 19: Reference to Lu et al., 2016a is not an adequate reference dealing with satellite observations and does not bring added value to the manuscript. But my main criticism is that it is not a good practice to insert unrequested references to own papers during the review process, even without highlighting the change in the "track change" document. See also `http://www.atmospheric-chemistry-and-physics.net/about/publication_ethics.html`.

- P 3, L 9 – L19: I really appreciate this nice overview on the vertical extensions of ash layers reported in literature.

- P 3, L 20 – 22: The sentence *"Thin ash clouds, by their nature are of less concern to aviation because such clouds would be traversed rapidly avoiding the possibility of particle build-up that might lead to engine failure."* is problematic as vertically thin layers are not necessarily traversed rapidly in the horizontal direction. Although the statement might be true in many cases it is not difficult to construct a (realistic) scenario where it is not true. Also, *"avoiding"* is a rather strong word here. I suggest to use "reducing" or "minimizing" in this context.

- P 3, L 20 – 22: *"From a modeling perspective lack of vertical resolution in model wind data makes it not useful to make the cloud depth any less than 500 m."*: I don't think that this argument is conclusive here. The transformation from 2D satellite data to 3D fields is done prior any contact with the model wind fields; as you state *"The outcome of SOO can be considered as preprocessing to the satellite data assimilation system."*.

- P 3, L 20 – 22: To me it still remains a shortcoming that the authors do not consider the full range of observed vertical thickness, 100 m – 3 km. Although I now agree that the vast mayority is within the considered range, there is no scientific reason to restrict the analysis on these cases. (In the vast majority of days there is no volcanic ash in the European airspace but nobody would argue that this is a adequate reason to not consider such rare events...)

- P 5, L 8 – 9: *"lowest possible thickness"* should be "lowest expected thickness" or "lowest considered thickness"

- P 6, L 8 – 9: Figure 1c does only report errors of ash load for pixels where volcanic ash is detected. What is about the error of pixels where no ash is detected? There might be ash as well (for example ash concentration below the detection threshold or ash which is hidden by meteorological clouds). I think the latter should also go into the $ML_{error}$?

- P 8, L 22: It is not an objective argumentation to select a small area of the whole region (like Netherlands here), where good agreement is achieved, and to discuss the numbers in a way (observations 3.1 mg/m$^2$ versus EnSR values of 2.9 - 3.2 mg/m$^2$) implicitly suggesting that the assimilation performance is that close to observations in general. This suggestion is even supported by using *"for example"* in L 22. However, the Netherlands are one of very few areas where you achieve that good agreement. There are even regions with similiar size like the Netherlands, namely the fjord area of Norway or the area around Nurfolk/Suffolk in England where the non-assimilated run is much closer to observation than the assimilated run. These findings should be reported in a fair and objective way as well.

- P 9, L 15 – 25: The aircraft measurements represent PM10 concentration at inlet position (x,y,z,t) while the model gives values which are averages of the concentration for a 0.25° x 0.1° x 1 km grid box. Please justify why is it appropriate to compare these two values.

- Figure 1: While there is a continous colour scale in Fig. 6b, there are only five colors in the figure. Is the top height retrieval discrete in a way that it turns out only discrete values of 4, 5, 6, 7, and 8 km? I don't think so, as the KNMI height product documentation shows continous values of top heigts.

- Figure 1: This is true for Fig 1a-c, but also several other figures in the manuscript: While zooming into the pdf file it seems that your plot consists of (intransparent) overlapping colored dots which is not the state of art how to plot a 2D satellite retrieval. I.e., the final figure depends on the sequence of the underlying data. I suggest to revise these figures.

- **Figure 6: The measurement values plotted in Fig. 6b are not consistent with the (same) measurement data reported in Weber et al., 2010**, see `http://dx.doi.org/10.1016/j.atmosenv.2011.10.030`. For example, at 10:00 UTC Weber et al., 2010 reports on PM10 concentration values slightly below/above 200 $\mu$g/m$^3$ whereas Fig 6b indicates values at 10:00 UTC of 130 g/m$^3$ which is a approx. 35% smaller

value. This is important here, as this incostistency pimps the general agreement of aircraft measurements and EnSR result in the figure.

- Figure 6: Same is true for Fig 6c where Weber et al. 2010, Fig. 8 report on values of approx. 300 $\mu$g/m$^3$ at 13:05 UTC while the authors data report values of 200 $\mu$g/m$^3$. Please clearify.

**3   minor comments**

P 4, L 17: "caled" should be "called"
P 7, L 4: broken sentence structure.
P 8, L 8 + L 9: comma seem to be misplaced.

---

## Referee Report (RR2)

**Review of**

**Satellite data assimilation to improve forecasts of volcanic ash concentrations: a case study on the 2010 Eyjafjallajökull volcanic ash plume**

**by Fu et al.**

The paper describes a case study when SEVIRI-derived volcanic ash mass loadings and heights of the top of the volcanic layer are used for model calculations/forecasts of ash concentrations. The SEVIRI output is transformed into concentrations, adapted to the model grid (LOTUS-EUROS) and then used in the framework of an EnSR data assimilation scheme. This approach is applied and discussed for a short period of May 2010 and compared to airborne measurements of 5 hours durations. The authors show that the model results with data assimilation compare considerably better with the measurements than results without assimilation.

The paper provides a contribution to the 'hot topic' of the prediction of volcanic ash dispersion and how airlines (and the responsible authorities) can be advised to avoid critical flight routes. A shortcoming certainly is the limited set of data, so it is not possible to draw conclusions that are generally applicable. Insofar I appreciate that it is now clearly stated (even in the title) that the manuscript covers a case study only. Further studies are definitely necessary.

However, before being published section 3.1 needs to be improved. The description is not as clear as it could/should be and the nomenclature is sort of strange. It must be possible to describe the methodology in an unambiguous way from a set of equations! Furthermore, the conclusions and outlook are too much focussed on passive space borne remote sensing; I am sure that ground based networks (active remote sensing) will contribute significantly to the 'volcanic ash topic' and should be mentioned in this paper when discussing future perspectives.

Comments on Section 3.1

I have not understood the concept of the procedure. Maybe other readers will have similar problems, remember that this topic was also raised by the reviewers of the first submission of the manuscript. I don't state that the approach is

wrong, but the description is confusing and obviously not all required information are given: In particular it is not clear, what is prescribed/fixed and what is variable. So I encourage the authors to rephrase this section in such a way that nothing remains unclear or might be misunderstood, with text/equations/figure being consistent. Expressions as '*extraction layer*' can hardly be associated to an atmospheric feature, so a more intuitive name should be chosen to facilitate the understanding. Clarity of this section is essential as it is one of the most important parts of the paper and covers the novelty of the study.

The following remarks/comments/attempts of interpretation may illustrate my confusion:

From SEVIRI mass loading is derived. This is converted to the vertical column by a simple (but sufficiently accurate) approach using Eq. (1). Thus, the remark on page 2/line 35 ('*the path can be a line or a curve...*') can be omitted, especially as it is more confusing than explaining.

The second parameter derived from SEVIRI is $H_{top}$. Here, a typical accuracy of this value should be given. Then it is stated that '*the thickness of the plume is investigated*'. What does this mean? '*Investigated*' implies that the thickness is varied and a specific/best value is selected by a clearly defined independent criterion. However, neither a definition and an explanation is given. $T_{low}$ and $T_{high}$, and the incremental thickness $T$ are prescribed. Thus, all other values ($T_i$, $C_i$) are also fixed and depend on the measured $ML_{vert}$ in the same way in all cases. $C_i$ does not depend on $H_{top}$; that indeed can to be understood.

What is the reason to use different symbols ($H$ and $T$) for lengths, and not similar symbols for similar quantities; especially $T$ should obviously be something like $\Delta H$. The current nomenclature leads to unnecessary confusion. The same is true for $ML_{vert}$. This should be $M_{vert}$ or $m_{vert}$, two characters for one quantity are not common in physics, especially as part of equations (it cannot be distinguished from a product of $M$ and $L_{vert}$)! By the way: to be consistent with a range of possible thicknesses between $0.5\,\mathrm{km}$ and $3.0\,\mathrm{km}$, $N_s$ should be 51, not 50.

Eq.(4) assumes that each thickness (0.5 km, 0.55 km, 0.6 km, ...,2.95 km) has

the same probability (cf. line 18 on page 5): is this meant by the authors? Is a consequence of this that the vertical distribution (expressed as $C_i$) is unchanged during the dispersion of the ash-cloud? A variable thickness of the ash-layer (i.e. realistic conditions) is only possible, if $T_{high}$ (i.e. $N_s$) is changed. Is this indicated by the authors' remark 'Corresponding to the sampled thickness...'? However, it remains open what 'the sampled thickness' is and how it is derived.

The paragraph starting at line 27 (page 5) is also difficult to understand. How can it be inferred from Eq. (3) that the concentration in the 'yellow region' is zero? Where is an information like 'the cloud thickness is 1 km' derived from? It seems contradictorily if the thickness is stated to be 1 km, but the ash concentration is allocated to a 0.5 km layer ('between 7.5 km and 8.0 km'). Fig. 2 suggests that the ash layer is always $T_{top}$=0.5 km thick.

If parts of the procedure are only used to estimate the 'extraction error', it should be clearly stated here.

I hope that a revision of this section is possible to get a convincing description of the methodology! Maybe it is sufficient to add two 'carefully formulated' sentences, especially on the selection/derivation of the 'thickness of the ash layer', and all problems are solved.

Minor/technical comments

1. page 2, line 8: 'over a long distance': be more precise and give typical ranges (hundreds of kilometers).

2. page 2, line 27: 'there are 3712 ...': This information is not relevant, more interesting is the spatial resolution (as given later in the paper).

3. page 2, line 28: 'Nowadays ... as well as the ash cloud top height': An indication of the accuracy is welcome.

4. page 3, line 10: 'mostly from hundreds...': please rephrase; the thickness is certainly not hundreds of kilometers.

5. page 3, line 20: 'From a modeling perspective...': The model used in this study should be introduced here (at least earlier than in the current version of

the manuscript) and the resolution (and other key parameters) should be mentioned here.

6. page 3, line 32: '*Secondly, using the extracted ...*': What is meant by this statement? What kind of in-situ measurements are used, and where are they from?

7. page 4, line 6: '*...geographic area affected by the Eyjafjallajökull volcanic ash*': The ash affected a larger area than shown, e.g. Spain (Navas-Guzman et al., 2013), Greece (Kokkalis et al., 2013) or Romania (Nemuc et al., 2014). There is also an 'European overview' given by Pappalardo et al. (2013). So, the sentence should be more general.

8. page 4, lines 11 and 14: The statements concerning the error can be combined.

9. page 4, line 21: '*...resolution to the VATDM resolution*': Give numbers for illustration.

10. page 4, line 23: '*However, the correction cannot...*': What is meant with correction; be more precise.

11. page 4, line 24: '*...due to the insufficient vertical resolution in satellite data*': This only holds for passive remote sensing. In case of active remote sensing the contrary is true. A comment on this would be welcome.

12. page 6, line 3: It would be appreciated if another name for the '*extraction error*' is applied. This name does not support the understanding.

13. page 10: In the conclusions two very general comments on future needs/activities are made: '*This effective time period probably...* ' and '*How to determine the reasonable ...*'. This should be improved by more specific comments. To my knowledge a large number of/most(?) national weather services recently have implemented ceilometer networks, mainly for monitoring the dispersion of volcanic ash clouds (Weigner at al., 2014). These data set will be (and in part are already) available in near real time and

will provide excellent information about the (horizontal and) vertical distribution (with some restrictions due to cloud cover; but this is also true for space borne observations). It seems to me that they could be very promising candidates for data assimilation as well, and can complement satellite data.

References:

- Nemuc, A., Stachlewska, I.S., Vasilescu, J. et al. Optical properties of long-range transported volcanic ash over Romania and Poland during Eyjafjallajokull eruption in 2010, Acta Geophys. (2014) 62: 350. doi:10.2478/s11600-013-0180-7

- Navas-Guzman, F., D. Muller, J. A. Bravo-Aranda, J. L. Guerrero-Rascado, M. J. Granados-Munoz, D. Perez-Ramirez, F. J. Olmo, and L. Alados-Arboledas (2013), Eruption of the Eyjafjallajokull Volcano in spring 2010: Multiwavelength Raman lidar measurements of sulphate particles in the lower troposphere, J. Geophys. Res. Atmos., 118, 1804-1813, doi:10.1002/jgrd.50116.

- Kokkalis, P., Papayannis, A., Amiridis, V., Mamouri, R. E., Veselovskii, I., Kolgotin, A., Tsaknakis, G., Kristiansen, N. I., Stohl, A., and Mona, L.: Optical, microphysical, mass and geometrical properties of aged volcanic particles observed over Athens, Greece, during the Eyjafjallajökull eruption in April 2010 through synergy of Raman lidar and sunphotometer measurements, Atmos. Chem. Phys., 13, 9303-9320, doi:10.5194/acp-13-9303-2013, 2013.

- Pappalardo, G., Mona, L., D'Amico, G., Wandinger, U., Adam, M., Amodeo, A., Ansmann, A., Apituley, A., Alados Arboledas, L., Balis, D., Boselli, A., Bravo-Aranda, J. A., Chaikovsky, A., Comeron, A., Cuesta, J., De Tomasi, F., Freudenthaler, V., Gausa, M., Giannakaki, E., Giehl, H., Giunta, A., Grigorov, I., Groß, S., Haeffelin, M., Hiebsch, A., Iarlori, M., Lange, D., Linn, H., Madonna, F., Mattis, I., Mamouri, R.-E., McAuliffe, M. A. P., Mitev, V., Molero, F., Navas-Guzman, F., Nicolae, D., Papayannis, A., Perrone, M. R., Pietras, C., Pietruczuk, A., Pisani, G., Preißler, J., Pujadas, M., Rizi, V., Ruth, A. A., Schmidt, J., Schnell, F., Seifert, P., Serikov, I., Sicard, M., Simeonov, V., Spinelli, N., Stebel, K., Tesche, M., Trickl, T., Wang, X., Wagner, F., Wiegner, M., and Wilson, K. M.: Four-dimensional distribution of the 2010 Eyjafjallajökull volcanic cloud over Europe observed by EARLINET, Atmos. Chem. Phys., 13, 4429-4450, doi:10.5194/acp-13-4429-2013, 2013.

- Wiegner, M., Madonna, F., Binietoglou, I., Forkel, R., Gasteiger, J., Geiß, A., Pappalardo, G., Schäfer, K., and Thomas, W.: What is the benefit of ceilometers for aerosol remote sensing? An answer from EARLINET, Atmos. Meas. Tech., 7, 1979-1997, doi:10.5194/amt-7-1979-2014, 2014.

---

## Author Response (AR2)

Dear Editor,

Herewith we submit the revised manuscript acp-2016-436, "Data assimilation for volcanic ash plumes using a Satellite Observational Operator: a case study on the 2010 Eyjafjallajökull volcanic eruption ". First we would like to thank you and the three anonymous reviewers, and really appreciate the detailed comments and suggestions. We have carefully considered all the concerns and made changes accordingly in the revised paper. We believe the new version has been improved a lot compared to the previous version. In the new version, with the agreement of all authors, we have changed the author's order to Guangliang Fu, Fred Prata, Hai Xiang Lin, Arnold Heemink, Arjo Segers, Sha Lu.

In the following we will give our answers and reactions to the comments. To make the changes easier to identify, we have numbered them.

Best regards,
Guangliang Fu
on behalf all co-authors

(The revised manuscript is in the latter part of this pdf., Changes for Reviewer#1: "blue"; Reviewer#3: "red"; Reviewer#4: "purple")

**Reply to Reviewer #1:**

**Specific comments**:

1. *As already pointed out in my former review, the authors discuss the situation that there is an ash layer in clear-sky atmosphere. However, in general there are water and ice clouds above/below/within the ash layer, and situations where clouds move above the ash layer. In such situations the ash layer can't be detected by the satellite algorithms (e.g., Prata and Prata, 2012), leading to wrong assimilation input. I expect that in such cases the model run without assimilation will perform much better than the assimilated one. I suggest to (1) add a comment on that in the paper, and (2) discuss how these errors are treated in your equation (8).*

   Response:
   Yes, this is a good point. We have added a comment and a discussion on this issue in line(s) 7.6–7.12:
   " Note that in this study we discuss the situation that there is an ash plume in clear-sky atmosphere. However, in general there are water and ice clouds above/below/within the ash plume, and situations where clouds move above the ash layer. In such situations the ash plume cannot be detected by the satellite algorithms (Prata and Prata, 2012), leading to wrong assimilation input. For these reasons the SEVIRI retrieval takes a very conservative approach with lots of different thresholds and conditions to be met before a pixel is deemed to be ash-affected. Thus the error is more likely to be towards under-estimating the amount of ash. There is an error associated with the ash retrieval that reflects the confidence of the detection algorithm, as well as the error estimates on the derived quantities. Therefore under these situations, the SOO errors are also more likely to be an under-estimation.  "

2. *P 2, L 19: Reference to Lu et al., 2016a is not an adequate reference dealing with satellite observations and does not bring added value to the manuscript. But my main criticism is that it is not a good practice to insert unrequested references to own papers during the review process, even without highlighting the change in the "track change" document. See also `http://www.atmospheric-chemistry-and-physics.net/about/publication_ethics.html`.*

   Response:
   Agree. It is removed in line(s) 2.23:
   " satellite observations (Stohl et al., 2011; Prata and Prata, 2012),  "

3. *P 3, L 9 – L19: I really appreciate this nice overview on the vertical extensions of ash layers reported in literature.*

   Response:
   Thanks.

4. *P 3, L 20 – 22: The sentence "Thin ash clouds, by their nature are of less concern to aviation because such clouds would be traversed rapidly avoiding the possibility of particle build-up that might lead to engine failure." is problematic as vertically thin layers are not necessarily traversed rapidly in the horizontal direction. Although the statement might be true in many cases it is not difficult to construct a (realistic) scenario where it is not true. Also, "avoiding" is a rather strong word here. I suggest to use "reducing" or "minimizing" in this context.*

5. *P 3, L 20– 22: "From a modeling perspective lack of vertical resolution in model wind data makes it not useful to make the cloud depth any less than 500 m.": I don't think that this argument is conclusive here. The transformation from 2D satellite data to 3D fields is done prior any con- tact with the model wind fields; as you state "The outcome of SOO can be considered as preprocessing to the satellite data assimilation system."*

6. *P 3, L 20 – 22: To me it still remains a shortcoming that the authors do not consider the full range of observed vertical thickness, 100 m – 3 km. Although I now agree that the vast mayority is within the considered range, there is no scientific reason to restrict the analysis on these cases. (In the vast majority of days there is no volcanic ash in the European airspace but nobody would argue that this is a adequate reason to not consider such rare events...)*

Response:

The comment 4 is a good point. Moreover, "avoiding" has been changed to "reducing". We also agree with both the comment 5 and comment 6. We have re-written this part based on these three very useful comments. We employ 0.2–3.0 km as the most reasonable range (based on literature) for the Eyjafjallajökull volcanic plume, and we re-run experiments based on the new considered range. We also add a note for the 2006 Chaiten eruption, the thickness range of 0.1–0.4 km should be better recommended. All of this part is re-written in line(s) 3.29–4.5:

" For the vertical thickness information of volcanic ash clouds, Schumann et al. (2011) investigated on the 2010 Eyjafjallajökull eruption using airborne data that the volcanic ash clouds spread over large parts of Central Europe, mostly from hundreds of meters to 3 km depth. This is consistent with the results of (Marenco et al., 2011) who observed layer depths between 0.5 and 3.0 km. Dacre et al. (2015) also examined the ground-based lidar data for the Eyjafjallajökull eruption and found a mean layer depth of 1.2±0.9 km and compared this with model based estimates of 1.1±0.8 km. Prata and Prata (2012) found variable thicknesses ranging from 0.2 up to 3 km. The vast majority of data suggest thickness in the range 0.2–3 km for the 2010 Eyjafjallajökull eruption. Based on these investigations, it is not realistic to use a deterministic value to represent the overall ash cloud thickness, but we can reasonably assume that the thickness has a range of 0.2–3.0 km at the corresponding horizontal location of the SEVIRI retrieved measurements. Although this thickness information is not deterministic, its uncertainty spread is suitable in an observational operator for satellite data assimilation. Note that the thickness range can be different for other volcanic eruptions. For example, Prata et al. (2015) reported low cloud thickness with 80% of cases for the 2006 Chaiten eruption less than 400 m, thus a thickness range of 0.1–0.4 km is recommended for that eruption. "

7. *P 5, L 8 – 9: "lowest possible thickness" should be "lowest expected thickness" or "lowest considered thickness"*

Response:
Thanks. It is changed to "lowest considered thickness" in line(s) 6.6–6.7:
" The blue layer in Fig. 2 is determined by the lowest considered thickness "

8. *P 6, L 8 – 9: Figure 1c does only report errors of ash load for pixels where volcanic ash is detected. What is about the error of pixels where no ash is detected? There might be ash as well (for example ash concentration below the detection threshold or ash which is hidden by meteorological clouds). I think the latter should also go into the $ML_{error}$ ?*

Response:
Yes I agree. The SEVIRI product includes a confidence level that indicates how well/badly the "detection" (not the retrieval) has performed. As a confidence value it is difficult to include in a quantifiable manner into the scheme. There is a discussion of this in Kristiansen et al. (2015) but we admit this is a shortcoming.

9. *P 8, L 22: It is not an objective argumentation to select a small area of the whole region (like Netherlands here), where good agreement is achieved, and to discuss the numbers in a way (observations 3.1 mg/m2 versus EnSR values of 2.9 - 3.2 mg/m2 ) implicitely suggesting that the assimilation performance is that close to observations in general. This suggestioneven supported by using "for example" in L 22. However, the Netherlands are one of very few areas where you achieve that good agreement. There are even regions with similiar size like the Netherlands, namely the fjord area of Norway or the area around Nurfolk/Suffolk in England where the non-assimilated run is much closer to observation than the assimilated run. These findings should be reported in a fair and objective way as well*

Response:
Yes, this is a good point. Also combined with another Reviewer's comment, we have rewritten the related parts (note that due to we use the thickness range of 0.2–3 km, thus the assimilation result is not exactly same with previous version and we evaluate the new plot by considering the reviewer's concern) in line(s) 9.22–9.27:
" During two-days continuously assimilating SEVIRI measurements of the extracted $PM_{10}$ concentrations, without loss of generality, the analyzed volcanic ash state at 12:00 UTC 17 May/ 00:00 UTC 18 May 2010 is shown in Fig. 5**c**/ Fig. 6**c**. The conventional simulation without assimilation is also presented (Fig. 5**b**/ Fig. 6**b**), which is currently the commonly used strategy for the simulation of volcanic ash transport (Webley et al., 2012; Fu et al., 2015). It is clear that in almost the entire plume, the mass loadings by EnSR with SOO are in a better agreement with the SEVIRI mass loadings. It can be seen that EnSR with SOO effectively decreases the estimation level compared to the conventional simulation.

and also in line(s) 9.30–9.33:
" It should also be noted that while the assimilation does correct a rather large bias in the pure model output, it doesn't mean that the DA result is better in all locations. For example, in locations around Iceland, the DA doesn't lead the pure model; At a location (-1°W, 58°N) around England (see Fig. 5**a**), the simulation without assimilation seems to match better. "

10. *P 9, L 15 – 25: The aircraft measurements represent $PM_{10}$ concentration at inlet position $(x,y,z,t)$ while the model gives values which are averages of the concentration for a $0.25° \times 0.1° \times 1$ km grid box. Please justify why is it appropriate to compare these two values.*

    Response:
    Thanks. I have to say that is the best we can do given that the measurements are made at different resolutions and by different methods. We have added a note related to it. in line(s) 15.15–15.18:
    " Note that the aircraft measurements represent $PM_{10}$ concentration at inlet position while the model gives values which are averages of the concentration for a $0.25° \times 0.125° \times 1$ km grid box. Assuming the uniformity of the fields, it is appropriate to compare these two values given that the measurements are made at different resolutions and by different methods. However, when the heterogeneity is a serious issue, the model representation error should also be better considered. "

11. *Figure 1: While there is a continous colour scale in Fig. 6b, there are only five colors in the figure. Is the top height retrieval discrete in a way that it turns out only discrete values of 4, 5, 6, 7, and 8 km? I don't think so, as the KNMI height product documentation shows continous values of top heigts.*

12. *Figure 1: This is true for Fig 1a-c, but also several other figures in the manuscript: While zooming into the pdf file it seems that your plot consists of (intransparent) overlapping colored dots which is not the state of art how to plot a 2D satellite retrieval. I.e., the final figure depends on the sequence of the underlying data. I suggest to revise these figures.*

    Response:
    Thanks for the comment 11 and 12 in plotting. The dot plot is because the SEVIRI does not sample contiguously. I agree the previous plotting contains overlapping colored dots. In the new version, as suggested, we have re-plotted all the SEVIRI-related figures.

13. *Figure 6: The measurement values plotted in Fig. 6b are not consistent with the (same) measurement data reported in Weber et al., 2010, see `http: // dx. doi. org/ 10. 1016/ j. atmosenv. 2011. 10. 030`. For example, at 10:00 UTC Weber et al., 2010 reports on PM10 concentration values slightly below/above 200 $\mu g/m^3$ whereas Fig 6b indicates values at 10:00 UTC of 130 $\mu g/m^3$ which is a approx. 35% smaller value. This is important here, as this incostistency pimps the general agreement of aircraft measurements and EnSR result in the figure.*

14. *Figure 6: Same is true for Fig 6c where Weber et al. 2010, Fig. 8 report on values of approx. 300 $\mu g/m^3$ at 13:05 UTC while the authors data report values of 200 $\mu g/m^3$. Please clearify.*

    Response:
    This is a good point. We upload the original aircraft data (see Supplement), from which the exact value at 10:00 is 168.4 $\mu$g m$^{-3}$. The original continuous values are sampled every 6 seconds. In our previous plotting, we plotted the continuous concentrations every 1 minute, thus we averaged 10 values. In that way, the continuous plot would be smoother and thus easier for comparation. The same clearification is for 13:05 that the original value is 220.3

$\mu$g m$^{-3}$. In the new version, to be consistent with the original data, we have re-plotted the continuous values.

**Minor comments**:

1. *P 4, L 17: "caled" should be "called"*

   Response:
   Corrected in line(s) 5.6.

2. *P 7, L 4: broken sentence structure.*

   Response:
   Revised in line(s) 8.10–8.11:
   " The stochastic Plume Height (PH) is assumed to be temporally correlated with exponential decay. The correlation parameter $\tau$ is set to be 1 hour (Fu et al., 2015). "

3. *P 8, L 8 + L 9: comma seem to be misplaced.*

   Response:
   Thanks. Corrected in line(s) 9.10–9.13:
   " Note that, in this study only PM$_{10}$ ash component is considered in the assimilation system. This is consistent with that (during satellite retrievals) only the fine particles (mostly with sizes <10.0 $\mu$m) can be detected in the tropospheric volcanic plume based on the robust and reliable retrieval algorithms (Prata, 1989; Corradini et al., 2008). "

**Reply to Reviewer #3:**

**Comments on Section 3.1**:

1. Comments on Section 3.1:
   *(1) I have not understood the concept of the procedure. Maybe other readers will have similar problems, remember that this topic was also raised by the reviewers of the first submission of the manuscript. I don't state that the approach is wrong, but the description is confusing and obviously not all required information are given: In particular it is not clear, what is prescribed/fixed and what is variable. So I encourage the authors to rephrase this section in such a way that nothing remains unclear or might be misunderstood, with text/equations/figure being consistent. Expressions as "extraction layer" can hardly be associated to an atmospheric feature, so a more intuitive name should be chosen to facilitate the understanding. Clarity of this section is essential as it is one of the most important parts of the paper and covers the novelty of the study.*

   Response:
   Thank you very much for very helpful comments and explain to us the possible confusion in every step. We have fully re-written this part according to the details questions. We first answer your questions one by one. First, I agree that "extraction layer" is not a good name, we have changed it to "target layers" in line(s) 6.6–6.7:
   " the target layers used in this study refer to both blue and yellow layers. "

   *(2) The following remarks/comments/attempts of interpretation may illustrate my confusion:*
   *From SEVIRI mass loading is derived. This is converted to the vertical column by a simple (but sufficiently accurate) approach using Eq. (1). Thus, the remark on page 2/line 35 ('the path can be a line or a curve…') can be omitted, especially as it is more confusing than explaining.*

   Response:
   Agree. Omitted as suggested.

   *(3) The second parameter derived from SEVIRI is $H_{top}$. Here, a typical accuracy of this value should be given. Then it is stated that "the thickness of the plume is investigated". What does this mean? "Investigated" implies that the thickness is varied and a specific/best value is selected by a clearly defined independent criterion. However, neither a definition and an explanation is given. $T_{low}$ and $T_{high}$, and the incremental thickness $T$ are prescribed. Thus, all other values ($T_i$, $C_i$) are also fixed and depend on the measured $ML_{vert}$ in the same way in all cases. $C_i$ does not depend on $H_{top}$; that indeed can to be understood.*

   Response:
   Here, "Investigated" is not a suitable word. We choose a reasonable assumption on the thickness range based on a literature review in introduction. Now we re-written this sentence in line(s) 6.3–6.4:

" As introduced in Section 2, the cloud top height ($H_{top}$) is available from satellite remote sensing and the appropriate thickness range (from $T_{low}$ to $T_{high}$, i.e., 0.2 to 3 km) of the plume has been chosen for this case based on a literature review. "

*(4) What is the reason to use different symbols (H and T ) for lengths, and not similar symbols for similar quantities; especially T should obviously be something like $\delta H$ . The current nomenclature leads to unnecessary confusion. The same is true for $ML_{vert}$. This should be $M_{vert}$ or $m_{vert}$, two characters for one quantity are not common in physics, especially as part of equations (it cannot be distinguished from a product of M and $L_{vert}$)! By the way: to be consistent with a range of possible thicknesses between 0.5 km and 3.0 km, $N_s$ should be 51, not 50.*

Response:
We agree all suggested changes, but would prefer to keep H and T different, since H means Height and T means Thickness for different things. $ML_{vert}$ has changed to $M_v$. In this version, we have re-considered a more suitable thickness range of 0.2–3.0 km, thus now $N_s$ is corrected to 57, not 56 as previous one. This part is in line(s) 6.8–6.16:
" The details are formulated as follows. First we define

$$N_s = \left\lceil \frac{T_{high} - T_{low}}{\Delta T} \right\rceil + 1 \quad , \tag{2}$$

$$T_i = T_{low} + (i - 1) \times \Delta T, \quad C_i = \frac{M_v}{T_i}, \qquad i = 1, 2, \cdots, N_s \quad , \tag{3}$$

where $\Delta T$ is a step length and $N_s$ is the number of the possible thickness. $T_{low}$ represents the blue layer (see Fig. 2) with the fixed thickness of 0.2 km and $T_{high} - T_{low}$ (i.e., the substraction between $T_{high}$ and $T_{low}$) represents the yellow layer with the fixed thickness of 2.8 km. $\Delta T$ is chosen at a small value compared to $T_{low}$, which guarantees $N_s$ is not too small (e.g., less than 2) to have a sufficient number of sample thickness $T_1$, $T_2$, $\cdots$, $T_{N_s}$. (e.g., $\Delta T$ is chosen as 0.05 km in this case study, thus $N_s$ is calculated as 57.) "

*(5) Eq.(4) assumes that each thickness (0.5 km, 0.55 km, 0.6 km, ...,2.95 km) has the same probability (cf. line 18 on page 5): is this meant by the authors? Is a consequence of this that the vertical distribution (expressed as $C_i$ ) is unchanged during the dispersion of the ash-cloud? A variable thickness of the ash-layer (i.e. realistic conditions) is only possible, if $T_{high}$ (i.e. $N_s$) is changed. Is this indicated by the authors' remark "Corresponding to the sampled thickness..."? However, it remains open what "the sampled thickness" is and how it is derived.*

Response:
Yes, what you understood is what we meant. The sampled thickness is $T_i$ and it is derived based on Eq. (3). We have clearly stated it as in line(s) 6.17–6.18:
" Corresponding to the sampled thickness (i.e., $T_1$, $T_2$, $\cdots$), the ash concentration can be calculated also as a sample from $C_1$ to $C_{N_s}$ through Eq. (3). "

*(6) The paragraph starting at line 27 (page 5) is also difficult to understand. How can it be inferred from Eq. (3) that the concentration in the "yellow region" is zero? Where is an information like "the cloud thickness is 1 km" derived from ? It seems contradictorily if*

*the thickness is stated to be 1 km, but the ash concentration is allocated to a 0.5 km layer ("between 7.5 km and 8.0 km"). Fig. 2 suggests that the ash layer is always $T_{top}=0.5$ km thick.*

Response:

This part is redundant and easily confuse the readers. In the new version, our target layers are extended to both blue and yellow layers. We have removed this paragraph to avoid the misleading information.

*(7) If parts of the procedure are only used to estimate the "extraction error", it should be clearly stated here.*

Response:

Thanks. We have moved the information to the following "SOO error" part.

*(8) I hope that a revision of this section is possible to get a convincing description of the methodology! Maybe it is sufficient to add two "carefully formulated" sentences, especially on the selection/derivation of the "thickness of the ash layer", and all problems are solved.*

Response:

Thanks for all the discussions above. Benefitting from them, the full Section 3.1 is re-written in line(s) 5.24–6.23:
" The derivation of the Satellite Observational Operator (SOO) is shown in Fig. 2. The retrieved values by SEVIRI for the ash mass loadings (M) can be taken as an integration of ash concentrations along the retrieval path. In principle, the satellite retrieval path could be complicated but generally it is assumed to be a straight line (along the line-of-sight, ignoring refraction) from the measuring apparatus. The angle between the local zenith and the line of sight to the satellite is called Viewing Zenith Angle (VZA). The VZA (represented with $\alpha$) for each pixel is computed according to the satellite VZA algorithms (Gieske et al., 2005) by using general parameters (such as longitude, latitude of each pixel). With the cosine of this angle and the retrieved ash mass loadings (M), the mass loadings in the vertical direction ($M_v$) can be calculated by Eq. (1),

$$M_v = M \times \cos(\alpha). \tag{1}$$

To extract ash concentrations from SEVIRI retrievals, $M_v$ only is not sufficient and knowledge about the vertical distribution of ash cloud must be included. The cloud vertical profile can be described with the height of the top and the thickness of the cloud. As introduced in Section 2, the cloud top height ($H_{top}$) is available from satellite remote sensing and the appropriate thickness range (from $T_{low}$ to $T_{high}$, i.e., 0.2 to 3 km) of the plume has been chosen for this case based on a literature review. Fig. 2 illustrates how the 3D ash concentrations are extracted from the obtained mass loadings in the vertical direction ($M_v$). The blue layer in Fig. 2 is determined by the lowest considered thickness ($T_{low}$) and the target layers used in this study refer to both blue and yellow layers. When the top height and the thickness range of ash cloud are known, the ash concentration in the target layers can be calculated by using the ash mass loadings ($M_v$) at the corresponding

horizontal location. The details are formulated as follows. First we define

$$N_s = \left\lceil \frac{T_{\text{high}} - T_{\text{low}}}{\Delta T} \right\rceil + 1 \quad , \tag{2}$$

$$T_i = T_{\text{low}} + (i-1) \times \Delta T, \quad C_i = \frac{M_{\text{v}}}{T_i}, \qquad i = 1, 2, \cdots, N_s \quad , \tag{3}$$

where $\Delta T$ is a step length and $N_s$ is the number of the possible thickness. $T_{\text{low}}$ represents the blue layer (see Fig. 2) with the fixed thickness of 0.2 km and $T_{\text{high}} - T_{\text{low}}$ (i.e., the substraction between $T_{\text{high}}$ and $T_{\text{low}}$) represents the yellow layer with the fixed thickness of 2.8 km. $\Delta T$ is chosen at a small value compared to $T_{\text{low}}$, which guarantees $N_s$ is not too small (e.g., less than 2) to sample sufficient number of thickness $T_1$, $T_2$, $\cdots$, $T_{N_s}$. (e.g., $\Delta T$ is chosen as 0.05 km in this case study, thus $N_s$ is calculated as 57.)

Corresponding to the sampled thickness (i.e., $T_1$, $T_2$, $\cdots$), the ash concentration can be calculated also as a sample from $C_1$ to $C_{N_s}$ through Eq. (3). According to Eq. (2) and Eq. (3), $T_i$ ($i = 1, 2, \cdots, N_s$) is unchanged during the dispersion of the ash cloud, while $C_i$ is temporally changed but it does not depend on $H_{\text{top}}$. Therefore, at one measurement time, the mean (C) of the sampled ash concentrations can be calculated by Eq. (4)),

$$C = \frac{1}{N_s}(C_1 + C_2 + \cdots + C_{N_s}) \quad . \tag{4}$$

C is therefore used in this study as the extracted concentration between the heights [$H_{\text{top}}$ - $T_{\text{high}}$] and $H_{\text{top}}$ (i.e., both the blue and yellow layers in Fig. 2).  "
and we believe this part now has been well improved.

**Minor/technical comments**:

1. *page 2, line 8: "over a long distance": be more precise and give typical ranges (hundreds of kilometers).*

   Response:
   Revised as suggested.

2. *page 2, line 27: "there are 3712 ...": This information is not relevant, more interesting is the spatial resolution (as given later in the paper).*

   Response:
   Agree. It is removed now.

3. *page 2, line 28: "Nowadays ... as well as the ash cloud top height": An indication of the accuracy is welcome.*

   Response:
   As suggested, we have added a note related to this issue in line(s) 5.1–5.3:

" Although there is indeed an error in the ash cloud top height, the product of the errors in the SEVIRI-KNMI ash height has not been available as the product of mass loading errors. Thus, for the current study, we use the data of ash cloud top height as deterministic. "

4. *page 3, line 10: "mostly from hundreds...": please rephrase; the thickness is certainly not hundreds of kilometers.*

   Response:
   Thanks. It is corrected in line(s) 2.12:
   " hundreds of meters "

5. *page 3, line 20: "From a modeling perspective...": The model used in this study should be introduced here (at least earlier than in the current version of the manuscript) and the resolution (and other key parameters) should be mentioned here.*

   Response:
   I agree with you. Also combined with another reviewer's comment, the transformation from 2D satellite data to 3D fields is done prior any contact with the model. Thus, this sentence is not suitable here and is removed in the new version. We have re-written this part in line(s) 3.29–4.5:
   " For the vertical thickness information of volcanic ash clouds, Schumann et al. (2011) found for the 2010 Eyjafjallajökull eruption using airborne data that the volcanic ash clouds spread over large parts of Central Europe, mostly from hundreds of meters to 3 km depth. This is consistent with the results of (Marenco et al., 2011) who observed layer depths between 0.5 and 3.0 km. Dacre et al. (2015) also examined the ground-based li­dar data for the Eyjafjallajökull eruption and found a mean layer depth of 1.2±0.9 km and compared this with model based estimates of 1.1±0.8 km. Prata and Prata (2012) found variable thicknesses ranging from 0.2 up to 3 km. The vast majority of data suggest thickness in the range 0.2–3 km for the 2010 Eyjafjallajökull eruption. Based on these investigations, it is not realistic to use a deterministic value to represent the overall ash cloud thickness, but we can reasonably assume that the thickness has a range of 0.2–3.0 km at the corresponding horizontal location of the SEVIRI retrieved measurements. Al­though this thickness information is not deterministic, its uncertainty spread is suitable in an observational operator for satellite data assimilation. Note that the thickness range can be different for other volcanic eruptions. For example, Prata et al. (2015) reported low cloud thickness with 80% of cases for the 2006 Chaiten eruption less than 400 m, thus a thickness range of 0.1–0.4 km is recommended for that eruption. "

6. *page 3, line 32: "Secondly, using the extracted ...": What is meant by this statement? What kind of in-situ measurements are used, and where are they from?*

   Response:
   Here the extracted in situ measurements are meant for the extracted 3D concentrations from 2D mass loadings from SOO. It is re-written in line(s) 4.16:
   " Secondly, using the extracted  3D concentrations,  "

7. *page 4, line 6: "...geographic area affected by the Eyjafjallajökull volcanic ash": The ash affected a larger area than shown, e.g. Spain (Navas-Guzman et al., 2013), Greece (Kokkalis et al., 2013) or Romania (Nemuc et al., 2014). There is also an "European overview" given by Pappalardo et al. (2013). So, the sentence should be more general.*

   Response:
   Thanks. We have re-written this part in line(s) 4.23–4.26:
   " A region covering 30° W to 15° E and 45° N to 70° N is selected here for analysis which includes parts of the geographic areas affected by the Eyjafjallajökull volcanic ash (see Fig. 1). Actually the ash affected a larger area than shown, e.g, Spain (Navas-Guzmán et al., 2013), Greece (Kokkalis et al., 2013), Rominia (Nemuc et al., 2014). There is also an 'European overview' given by Pappalardo et al. (2013). "

8. *page 4, lines 11 and 14: The statements concerning the error can be combi- ned.*

   Response:
   Agree. It is revised in line(s) 4.30–5.1:
   " Besides ash mass loadings, other products including the ash cloud top height (Fig. 1**b**), and the error of ash mass loadings (Fig. 1**c**, which indicates the uncertainty and accuracy of the retrieved mass loadings ) are also available in a near real-time sense (Francis et al., 2012; Prata and Prata, 2012). "

9. *page 4, line 21: "...resolution to the VATDM resolution": Give numbers for illustration.*

   Response:
   in line(s) 5.9–5.11:
   " Additional processing on the retrieved measurements is needed to translate the data from the original SEVIRI resolution (i.e., 0.1° longitude × 0.1° latitude (Prata and Prata, 2012)) to a VATDM horizontal resolution (i.e., 0.25° longitude × 0.125° latitude, as used in (Fu et al., 2016)). "

10. *page 4, line 23: "However, the correction cannot...": What is meant with correction; be more precise.*

    Response:
    It is revised in line(s) 5.14–5.16:
    " the correction on the 3D state cannot be directly implemented by DA in the 3D state space due to the insufficient vertical resolution in satellite data (Bocquet et al., 2015). "

11. *page 4, line 24: "...due to the insufficient vertical resolution in satellite data": This only holds for passive remote sensing. In case of active remote sensing the contrary is true. A comment on this would be welcome.*

    Response:
    Yes. This is a good point. One comment has been added in line(s) 5.16–5.21:

" Note that this statement only holds for passive remote sensing, e.g., SEVIRI retrievals. In case of active remote sensing the contrary is true. For example, the spaceborne CALIOP lidar certainly has good vertical resolution. In the lower atmosphere it is 30-60 m (below 20 km). The same is true for ground-based lidars but usually these are of limited value. Ceilometers cannot distinguish ash from other scatterers in the lower atmosphere and cloudiness is a big problem – worse than from passive satellite because, at least for the aviation hazard, the ash needs to be elevated (above 10,000 ft or more) and above clouds (a problem then for looking upwards but not for looking downwards). "

12. *page 6, line 3: It would be appreciated if another name for the "extraction error" is applied. This name does not support the understanding.*

Response:
Yes. We have changed it to "SOO error" which should be to the point.

13. *page 10: In the conclusions two very general comments on future needs/activities are made: "This effective time period probably... " and "How to determine the reasonable ...". This should be improved by more specific comments. To my knowledge a large number of/most(?) national weather services recently have implemented ceilometer networks, mainly for monitoring the dispersion of volcanic ash clouds (Weigner at al., 2014). These data set will be (and in part are already) available in near real time and will provide excellent information about the (horizontal and) vertical distri- bution (with some restrictions due to cloud cover; but this is also true for space borne observations). It seems to me that they could be very promising candidates for data assimilation as well, and can complement satellite data.*

Response:
I agree. We have improved the comments more specific to our case study, as in line(s) 12.29–13.2:
" Evaluations using both satellite retrieved data and aircraft in situ measurements showed that the effective forecast time duration after satellite data assimilation with SOO is about 6 hours for the distal part of the Eyjafjallajökull volcanic eruption. The results also showed that the effective forecast duration (after DA with SOO) tends to be longer as the region is farther away from the volcano, which is because inaccurate ESPs require longer time to impact on the regions farther from the volcano. Note that for other cases (e.g., other volcano), this duration should be different and should be re-evaluated due to its dependence on the specific weather condition and on the specific model. "
In addition, we add comments on the ceilometer networks in the end of the paper in line(s) 13.15–13.20:
" Recently, a large number of national weather services has implemented ceilometer networks, mainly for monitoring the dispersion of volcanic ash clouds (Wiegner et al., 2014). These data set will be (and in part are already) available in near real time and will provide information about the (horizontal and) vertical distribution (with some restrictions due to cloud cover; but this is also true for space borne observations). Thus, they could be promising candidates for data assimilation as well, but there are indeed some caveats. It will be interesting for future research to make use of and demonstrate the case for using ground-based lidars. "

**Reply to Reviewer #4:**

**Major comments**:

1. *The authors state that the satellite observations "are often two-dimensional (2D), and cannot easily be combined with a three-dimensional (3D) volcanic ash model", which motivates their development of a satellite observational operator (SOO). In fact, this issue is common to many DA applications using satellite measurements. Satellite instruments using nadir viewing geometry measure 2D fields. The standard technique in DA is to apply an observational operator (H) to the model output, to be able to compare model and satellite data in measurement space. Satellite observational operators (SOO) are sometimes used to transform observations closer to model space before assimilation. Using an SOO simplifies some aspects of the DA, but does so at the cost of including some set of assumptions in the SOO. In the present case, I find the motivation for the use of a SOO to be lacking (a mismatch between 2D and 3D fields is not sufficient).*

   *Furthermore, the assumptions that are included into this SOO appear to be questionable. The authors argue that volcanic ash clouds have thicknesses between 0.5 and 3.0 km, although they quote studies having measured even smaller thicknesses. Given the relative large range, it is not clear why even smaller thicknesses are not included in the range. Then they calculate a range of possible concentrations by dividing the satellite retrieved ash loading (in g/km$^2$) by thicknesses between 0.5 and 3.0 km. They take the mean and SD of these concentrations as their best guess (and uncertainty in) ash concentration. This seems like a complicated way of assuming a mean thickness of 1.75 km. Then the authors focus on the ash concentration at cloud top, which appears to be the quantity which is compared to the model output within the DA (although this is not sufficiently explained). There is no real justification for why the focus is restricted to the concentration at cloud top, and this process seems like it discards a significant amount of useful information (e.g., the total mass loading).*

   *Most of these assumptions and complications would be avoided if measurements and model results were compared in measurement space rather than model space, as is the standard in DA. This would simply require the observation operator H to take the vertical integral of the modeled ash profile.*

   Response:
   First of all, thank you for pointing out the lack of sufficient motivations. Actually, the motivation to design SOO was not just based on "are often two-dimensional (2D), and cannot easily be combined with a three-dimensional (3D) volcanic ash model". In the new version, we have carefully and fully introduced the motivations, as in line(s) 3.5–3.23:
   " Thus, the 2D measurements are not directly suited in a 3D ensemble-based DA system. One way to ameliorate this difficulty in the analysis step of DA, is to compare the measurements and the model results in the 2D measurement space. This simply requires an observational operator to take the vertical integral of the modeled ash profile. After the analysis in the measurement space, the corrected 2D mass loadings are distributed to each vertical layer based on the prior modeled ash profile. However, this approach adds

artificial vertical correlations to all the vertical ash layers when the prior modeled ash vertical profile is not accurate (Lu et al., 2016b). This is a common problem with respect to passive data assimilation due to the lack of vertical resolution in data (Blayo et al., 2014; Bocquet et al., 2015). For applications where the vertical profile is not an issue (i.e., the prior profile can be modeled well), the added artificial correlations are of minor importance (Blayo et al., 2014). However, this is not the case for volcanic ash application where the ash plume usually has significant vertical variation and is located in a narrow vertical band (e.g., see Fig. 7 in (Prata and Prata, 2012) and Fig. 15(c) in (Lu et al., 2016a)). In general, the used model-based vertical profile is very inaccurate, thus the integral approach cannot accurately reconstruct states for all/most of the vertical ash layers. The influences of the artificial/spurious vertical correlations (introduced by the integral approach with the standard DA) on the assimilation performance has been extensively studied by Lu et al. (2016b) for the volcanic ash application in a concept of variational DA. (In this paper, we also have Section 5 to discuss on this issue with respect to ensemble-based DA).

In this study we look at an approach to avoid the problem of the artificial vertical correlations. Where the satellite provides 2D ash mass loadings, 3D information is available from the model and from additional observations. A 3D Satellite Observational Operator can be derived to make both types of information directly comparable in the 3D model space. This approach does not involve the integral operator, and thus avoids the artificial vertical correlations. "

In the new version, we have also changed the title to clearly state our approach–
" Data assimilation for volcanic ash plumes using a Satellite Observational Operator: a case study on the 2010 Eyjafjallajökull volcanic eruption "

Second, we have considered the smaller thickness in the new version and extended the thickness range to 0.2–3 km. The details about the thickness are in line(s) 3.29–4.5:
" For the vertical thickness information of volcanic ash clouds, Schumann et al. (2011) investigated on the 2010 Eyjafjallajökull eruption using airborne data that the volcanic ash clouds spread over large parts of Central Europe, mostly from hundreds of meters to 3 km depth. This is consistent with the results of (Marenco et al., 2011) who observed layer depths between 0.5 and 3.0 km. Dacre et al. (2015) also examined the ground-based lidar data for the Eyjafjallajökull eruption and found a mean layer depth of 1.2±0.9 km and compared this with model based estimates of 1.1±0.8 km. Prata and Prata (2012) found variable thicknesses ranging from 0.2 up to 3 km. The vast majority of data suggest thickness in the range 0.2–3 km for the 2010 Eyjafjallajökull eruption. Based on these investigations, it is not realistic to use a deterministic value to represent the overall ash cloud thickness, but we can reasonably assume that the thickness has a range of 0.2–3.0 km at the corresponding horizontal location of the SEVIRI retrieved measurements. Although this thickness information is not deterministic, its uncertainty spread is suitable in an observational operator for satellite data assimilation. Note that the thickness range can be different for other volcanic eruptions. For example, Prata et al. (2015) reported low cloud thickness with 80% of cases for the 2006 Chaiten eruption less than 400 m, thus a thickness range of 0.1–0.4 km is recommended for that eruption. "

Moreover, I agree (and thanks) that the extraction only at the cloud top would discard a significant amount of information. Since we designed SOO based on the thickness range 0.2–3 km, the extraction should not be restricted to the concentration within 0.2 km, but should be for the concentration within 3 km. In the new version, we have re-written this part in line(s) 6.22–6.23:

" C is therefore used in this study as the extracted concentration between the heights [$H_{top}$ - $T_{high}$] and $H_{top}$ (i.e., both the blue and yellow layers in Fig. 2). "

and in line(s) 7.14–7.15:

" The outcome of SOO can be considered as preprocessing to the satellite data assimilation system. The extracted data represent the data at the target layers, which can be taken as the data within the 3.0 km layer thickness. "

Third, I agree that the complications would be avoided if measurements and model results were compared in measurement space. However, as answered before, the integral approach would add artificial vertical correlations, which has been investigated and discussed in literature as severe for volcanic ash application. To clearly state assimilation with SOO to be an effective approach for our application, in the new version, we have also added experments in a new section (Section 5) to compare and discuss the two approaches (EnSR with SOO and without SOO), as in line(s) 10.1–10.30:

" Section 5    Discussion and comparison between EnSR with SOO and without SOO

We have implemented satellite data assimilation combined with SOO. However, this does not mean in terms of methodology, ensemble-based data assimilation without SOO cannot work. Under that circumstance, applying data assimilation on 2D measurements and 3D concentrations, the operator $H_k$ in Eq. (A5) does not select the grid cell in $x(k)$, but integrates all the vertical grid cells to calculate 2D mass loadings. This approach (denoted EnSR without SOO) would simplify the assimilation by only using 2D mass loadings. Although we focused SOO as a new way (as introduced in the introduction) to deal with satellite 3D data assimilation, the assimilation effects between with and without SOO should be better compared/studied in order to determine whether SOO is an effective approach.

Comparing to the EnSR implementations with SOO in Fig. **5c**/ Fig. **6c**, the cases of EnSR without SOO are shown in Fig. **5d**/ Fig. **6d**. Taking the SEVIRI retrieved mass loadings (Fig. **5a**/ Fig. **6a**) as references, it is revealed that EnSR without SOO performs better than the case without assimlation (Fig. **5b**/ Fig. **6b**), but worse than EnSR with SOO. One may also want to check the assimilation effect without SOO at the initial analysis time (i.e., 01:00 UTC, 16 May 2010), which is illustrated in Fig. 7. It can be seen that the assimilation difference between with and without SOO is small (by comparing Fig. **7c** and **7d**). While, at the second analysis time (i.e., 02:00 UTC, 16 May 2010) the results show much bigger differences (between Fig. **7g** and Fig. **7h**). These results verify and examine the influences of the artificial/spurious vertical correlations (caused by the integral-type of $H_k$, see introduction). These influences are examined to be accumulated step by step in our volcanic ash application and must be considered/avoided with respect to an acceptable assimilation result.

Now it is explained for our application why we chose SOO to compare measurements and model results in the model space rather than in the measurement space. Although the complications would be avoided if we do the analysis step in measurement space, however, the assimilation accuracy will then be worse than the case in the 3D model space by using SOO. Note that we perform EnSR without SOO in a standard/general way. We expect the problem of artificial vertical correlations (for volcanic ash plumes) may be partially compensated by employing some diagnostic or correction approaches, as discussed in (Blayo et al., 2014; Houtekamer and Zhang, 2016) for other applications. However, this would add many complications/difficulties to standard EnSR assimilation and whether/how it would work remains unknown (at the moment, no literature has reported on this issue for our

application), and that it would be another research.

In this study, we focus on a way not dealing with the artificial vertical correlations, and we propose SOO by incorporating data and information available. The additional data (e.g., cloud top height and thickness information) are important for SOO, which describes the structure of a volcanic ash plume; we also provide an idea in the sense of incorporating many available measurements. In addition, we expect the SOO can be potentially improved by incorporating more data, but at the moment DA with SOO has shown its advantage than the standard way (without SOO) in dealing with passive satellite data assimilation. "

2. *Fig 1a shows ash mass loadings from SEVERI, with values between 0 and 5 $g/m^2$ (although values could be higher). Taking 3 $g/m^2$ as a typical value, and under the assumption that cloud thicknesses are between 0.5 and 3 km thick (and that concentrations are uniform within the cloud), one would expect typical concentrations between 1-6 $mg/m^3$. The SOO extracted measurements in Fig 3a are around an order of magnitude less than this, with most values falling between 0.1-0.6 $mg/m^3$. The rough concentrations estimated here (1-6 $mg/m^3$) are also more in line with the model results shown in Fig 4. Given the apparent simplicity of the SOO, it should be possible for the reader to gain a quantitative feel for the magnitudes of the concentrations involved. At present, this is not possible. If the SOO is more complicated than I have understood it, then the description needs fair amount more detail.*

   Response:
   I think that the misunderstanding is due to:
   (1) the differences between original mass loadings (M) and the vertical mass loadings ($M_v$) (i.e., mass loadings in the vertical direction). The former is illustrated in Fig. 1**a**, while the latter is calculated by $M_v = M \times \cos(VZA)$ and is the one directly used for the SOO. Thank you for pointing out this question, which reminds us to explicitly plot the vertical mass loadings (i.e., $M_v$ in Eq. (1) in Fig. 1**d**.
   (2) because there are values in Fig. 1**d** as high as 5 and as low as 0 g m$^{-2}$ does not mean that 3 g m$^{-2}$ is typical. We have carefully checked that the typical value in Fig. 1**d** could be taken as 1 g m$^{-2}$.

   Then under the assumption that cloud thicknesses are between 0.2 and 3 km thick, one would expect the typical concentrations between 0.3–5 mg m$^{-3}$. The SOO extracted measurements in Fig. 3 are around this order of magnitude (about 0.5–4 mg m$^{-3}$).

   (In this version, we choose the time 01:00 UTC 16 May 2010 to illustrate the development of SOO, which will be further used/refered for the analysis at this time. ) I agree with you that given the apparent simplicity of SOO, it should be possible for the reader to gain a quantitative feel. We believe the current presentation has improved this aspect.

3. *There is only a small amount of evidence shown to support the conclusion that "satellite data assimilation can force the volcanic ash state to match the satellite observations, and that it improves the forecast of the ash state." Fig 5 shows a snapshot of ash loadings at a single time, comparing the SEVERI measurements with results from the model with and without assimilation. While the assimilation does seem to correct a rather large bias in the pure model output, it's not clear that the DA result is better in all locations: while the values over the Netherlands may be more realistic, it appears that the DA results over the west coast of Norway may be underestimated by the DA system. Whether the authors have chosen this single comparison randomly, or if this is a best- or worse-case is not stated.*

*One would expect some overall measure of skill with respect to time and space for such a comparison, in order to gauge the impact of DA.*

Response:

Yes, I agree and understand your concerns. In the new version, without loss of generality, we have shown results at both 12:00 UTC 17 May and 00:00 UTC 18 May. (And in Section 5 we also show the results at 01:00 UTC 16 May, 02:00 UTC 16 May.) The related part is re-written in line(s) 9.19–9.27:

" Based on this consideration, SEVIRI ash mass loadings need to be employed for a further validation. Note that the mass loadings used for this comparison are not the original mass loadings, but the mass loadings in the vertical direction, i.e., $M_v$ in Eq. (1) as plotted in Fig. 5a.

During two-days continuously assimilating SEVIRI measurements of the extracted $PM_{10}$ concentrations, without loss of generality, the analyzed volcanic ash state at 12:00 UTC 17 May/ 00:00 UTC 18 May 2010 is shown in Fig. 5c/ Fig. 6c. The conventional simulation without assimilation is also presented (Fig. 5b/ Fig. 6b), which is currently the commonly used strategy for the simulation of volcanic ash transport (Webley et al., 2012; Fu et al., 2015). It is clear that in almost the entire plume, the mass loadings by EnSR with SOO are in a better agreement with the SEVIRI mass loadings. It can be seen that EnSR with SOO effectively decreases the estimation level compared to the conventional simulation. "

Moreover, I agree that the DA result is not better in all locations. We have added a note to clearly state that in line(s) 9.30–9.33:

" It should also be noted that while the assimilation does correct a rather large bias in the pure model output, it doesn't mean that the DA result is better in all locations. For example, in locations around Iceland, the DA doesn't lead the pure model; At a location (-1°W, 58°N) around England (see Fig. 5a), the simulation without assimilation seems to match better. "

In addition, we think the word "force" is a very strong word, while what we aim to state is "improvement". Thus to avoid misleading to readers, we have changed "force" to "improve" and re-written the abstract in line(s) 1.10–1.11:

" The results show that satellite data assimilation with SOO can improve the estimate of volcanic ash state, and that it improves the forecast. "

and the conclusion in line(s) 12.23–12.24:

" The results showed the assimilation with SOO significantly reduces the estimation level of the conventional simulation. The accuracy of the volcanic ash state was shown to be significantly improved by the assimilation of satellite mass loadings. "

4. *Apparently, DA has already been shown to improve the forecasting of volcanic ash transport (Fu et al., 2015, 2016). The present study claims to quantify the duration of such improvement. Fig 6 shows a comparison of DA forecast results with in situ airplane measurements. Again, the DA initiation helps to correct for a major over estimation by the model without DA. The authors conclude from this comparison that with initial conditions taken from DA, the "effective duration of the improved regional volcanic ash forecasts is about a half day", although this is a "conservative" estimate, which is based only on this single case, and is in fact limited by the 15 h duration of the airplane measurements. It's not clear why the continuous satellite measurements were not used to quantify the duration of the improved forecast. The duration of the skill improvement also likely has much to do with the assumptions regarding the noise added to the plume height estimates. A convincing*

*assessment of the skill duration would require a much more thorough and detailed analysis.*

Response:

Thank you for the comments. I agree that satellite data should be the best to evaluate the effective forecast time duration after DA. We have added experiments and more detailed analysis. The major part of Section 6 has been re-written in line(s) 11.1–12.10:

" Section 6   Evaluation of the effective forecast duration after satellite data assimilation for the distal part of the Eyjafjallajökull volcano

According to Section 4.3, the accuracy of volcanic ash state is significantly improved by ensemble-based data assimilation after a continuous assimilation period (e.g., two days). Apparently, with the improved state as initialization, an improved forecast can be obtained (Fu et al., 2015). However, it remains unknown how long the improvement on forecasts will last.

To investigate the effective duration of the improved ash forecasts after assimilation, a one-day forecast is performed by initializing EnSR analyzed state (Fig. 6c) at 00:00 UTC 18 May 2010. Fig. 8 shows the forecast results after assimilation (first column in Fig. 8), the SEVIRI retrieved measurements (second column in Fig. 8) and the forecast results without assimilation (third column in Fig. 8). Without loss of generality, 06:00, 12:00 and 18:00 UTC are chosen to evaluate the comparisons over three divided distal regions of interest (denoted R1–R3). Note that we do not evaluate the near-volcano region (i.e., the left part of R1 in Fig. 8a), where the improved forecasts can be quickly influenced by the continuously noisy emissions. Thus it is safe to take the effective forecast duration for the near-volcano region as "zero hour".

For the region R1, the improved forecast after assimilation can last for 6 hours. This is viewed by that at 06:00 UTC, the forecast after assimilation (see R1 in Fig. 8a) is closer to the measurements (R1 in Fig. 8b) than the forecast without assimilation (R1 in Fig. 8c). However, the improvement diminishes after 6 hours (see the second and third rows in Fig. 8).

For the region R2, the forecast after DA at 06:00/12:00 UTC ( R2 in Fig. 8a/ 8d) has a good match with the retrieved mass loadings (R2 in Fig. 8b/ 8e). However, we are not sure whether the good match remains at the time 18:00 UTC, because there seems to be lack of data at that time (see R2 in Fig. 8h). Therefore, it is better to evaluate the effective forecast duration after DA with SOO for R2 is 12 hours.

For the region R3, at the three times 06:00/12:00/18:00 UTC, the forecasts after DA are generally better than forecasts without DA by comparing with the measurements. However it should be noted that in Fig. 8e, some higher values (over 3.5 g m$^{-2}$) are indeed retrieved in R3, but not in the forecasts after DA.

Our initial experimental tests show that, the effective time durations of the forecasts after satellite data assimilation with SOO can be considered as 6 hours for the region R1, 12 hours for the region R2, and 18 hours for the region R3. It also shows to us that the effective forecast duration tends to be longer as the region is farther away from the volcano. This is because longer time is needed for the influences of inaccurate ESPs to reach the areas farther from the volcano.

These results are evaluated using satellite retrieved data. For a further test, we employ another type of independent data. For this purpose, we use aircraft-based measurements (see details in Appendix B). Fig. 9a shows the flight route of the aircraft (in the region R3).

Fig. 9**b** and 9**c** are the comparison of aircraft $PM_{10}$ measurements against the forecasted concentrations after assimilation and without assimilation. Note that the low magnitude of aircraft measurements may result in little importance to verify high ash concentrations, but comparisons using them can indicate how good the assimilation is at reconstructing the outskirts/boundaries of the ash plume, which is important to describe the plume's structure.

For the period from 09:30 to 11:00 UTC (Fig. 9**b**), although the forecasting time has been over 9 hours (i.e., the last assimilation is 9 hours ago), the forecasted concentrations still have a good match with the accurate aircraft measurements, while the conventional forecast (i.e., forecast without assimilation) doesn't. This result shows the forecast over 11 hours after assimilation has also kept a high accuracy compared to the measurements. The result can be extended to 15 hours comparing with the other period from 12:30 to 15:00 UTC (Fig. 9**c**). This test result using independent aircraft data confirms our evaluation result using satellite data that for the region R3, the effective forecast duration after DA is about 18 hours.

Based on all of the above evaluations and in the view of the whole distal regions of interest (R1+R2+R3), 6 hours can be taken as the a reasonable effective time duration (after assimilation with SOO) for the case study. This time duration can be taken as an indication about how long a valid regional aviation advice based on the forecast after assimilation can last. It should be noted that for other cases (e.g., another volcanic eruption), the effective duration is different because it depends on the specific weather condition and the specific model used for forecasting. Thus, the effective duration for other case studies should be re-evaluated, and what we presented in this section can be useful for the readers to test it. "

According to the tests, 6 hours can be taken as a reasonable effective time duration for this case. We have re-written this in abstract in line(s) 1.11–1.14:
" Comparison with both satellite retrieved data and aircraft in situ measurements shows that the effective duration of the improved volcanic ash forecasts for the distal part of the Eyjafjallajökull volcano is about 6 hours. "
and in conclusion in line(s) 12.29–13.2:
" Evaluations using both satellite retrieved data and aircraft in situ measurements showed that the effective forecast time duration after satellite data assimilation with SOO is about 6 hours for the distal part of the Eyjafjallajökull volcanic eruption. The results also showed that the effective forecast duration (after DA with SOO) tends to be longer as the region is farther away from the volcano, which is because inaccurate ESPs require longer time to impact on the regions farther from the volcano. Note that for other cases (e.g., other volcano), this duration should be different and should be re-evaluated due to its dependence on the specific weather condition and on the specific model. "
We believe this part has been improved compared to the previous presentation.

**Specific comments**:

1. *P1, L1: "Data assimilation" is very general, the focus on ash forecasting should be clearer.*

**Response:**

I agree. We have clearly mentioned the focus and re-written the sentence in line(s) 1.1–1.2:
" Using data assimilation (DA) to improve model forecast accuracy is a powerful approach that requires available observations. "

2. *P1, L5: It doesn't appear that cloud thickness \*data\* is being included in the DA. Data-based assumptions, maybe.*

**Response:**

Agree. It is corrected in line(s) 1.6–1.7:
" By integrating available data of ash mass loadings and cloud top heights, and data-based assumptions on thickness, "

3. *P2, l4: A number of eruption source parameters are mentioned here, but only plume height is included in the DA ensemble generation. Some discussion would be nice at the end of the paper on the potential impact of uncertainties in the other ESPs.*

**Response:**

This is a good suggestion. The Mass Eruption Rate (MER) is actually uncertain in this study as PH, which is clearly mentioned in line(s) 8.13–8.16:
" Note that in this study the Mass Eruption Rate (MER) is calculated based on each uncertain PH, by using an empirical relationship (Mastin et al., 2009) between PH(km) and MER (kg s$^{-1}$)

$$\text{PH} = 2.00\text{V}^{0.241}, \quad and \quad \frac{\text{V}}{\text{MER}} = \frac{1.5e^3}{4.0e^6}. \tag{7}$$

Therefore, although we only add uncertainties in PH, MER is also not deterministic. "
According to suggestions, we add some discussion on the potential impact of uncertainties in other ESPs, e.g., Vertical Mass Distribution (VMD) and Particle Size Distribution (PSD) in line(s) 13.5–13.11:
" In this study, two of the Eruption Source Parameters (ESPs)–Plume Height (PH) and Mass Eruption Rate (MER) are made uncertain to generate DA ensembles. Actually the uncertainties can also be added in other ESPs, e.g., Vertical Mass Distribution (VMD) and Particle Size Distribution (PSD). However, adding noise in VMD and PSD should be very careful to keep their empirical/realistic distribution, e.g., "umbrella" shaped VMD (Sparks et al., 1997) and PSD for different types of eruption (Durant and Rose, 2009). Otherwise, the noisy VMD or PSD could provide unphysically biased prior ensemble plumes, resulting in DA algorithm impossible to reconstruct physical plume estimates. "

4. *P2, l10: Actually, you would hope that the VATDM was fairly accurate, and the DA was used because of unknowns in the ESPs.*

**Response:**

Agree. It is corrected in line(s) 2.13–2.14:
" For the purpose of improving the forecast accuracy of volcanic ash concentrations, efficient solutions must be employed to compensate the ESPs' inaccuracies. "

5. *P2, l11: DA does more than provide the initial conditions.*

   Response:
   I agree. It is re-written in line(s) 2.14–2.16:
   " Data assimilation (DA) can be used to create accurate initial conditions for model runs by using available measurements, which is one of the most commonly used approaches for real-time forecasting problems (Evensen, 2003; Bocquet et al., 2015; Fu et al., 2015). "

6. *P2, l17: It's not clear if these measurements are specific to Eyjafjallajökull, or more general.*

   Response:
   Here, we mention these measurements specific for the Eyjafjallajökull eruption. It is now clearly mentioned in line(s) 2.20–2.22:
   " For the 2010 Eyjafjallajökull volcanic eruption, during volcanic ash transport, different types of scientific measurement campaigns were performed to collect information of the ash plume. "

7. *P2, l25: should mention that the satellite is geostationary here.*

   Response:
   Thanks. It is mentioned in line(s) 2.28–2.29:
   " Geostationary satellite measurements are of special interest, because the detection domain is large and the output data is is at high temporal frequency (typically 15 – 30 minutes). "

8. *P2, l26: Earth's*

   Response:
   Corrected.

9. *P3, l6: Following from general comment, sparseness of data is not a real impediment to its use in DA, in fact, one could argue that sparse data is exactly what DA is useful for!*

   Response:
   I agree. It is re-written in line(s) 3.26–3.27:
   " Cloud-Aerosol Lidar with Orthogonal Polarization (CALIOP) (Winker et al., 2012) lidar measurements can provide detailed vertical information on plumes, but the measurements have low temporal resolution (polar-orbit) and the data processing and delivery are not designed for near real-time applications. "

10. *P3, l9: This discussion seems to assume that ash clouds have very well defined edges, but could there not be cases where the ash concentration decays smoothly over some vertical range? How is the thickness defined in such cases?*

Response:
Thanks for the question. We have answered it in line(s) 4.7–4.13:
" Layering is a typical property for distal volcanic ash cloud, thus the ash cloud is often called an ash plume. This property results in that usually there are clear edges around the plume, but it could indeed happen that the ash concentration decays smoothly over some vertical range, resulting in unclear ash cloud edges. In these cases, there would be long tails of very low concentrations, but the reported thickness ranges from literature can also fit, because (1) the reported thickness ranges are actually based on the visible/detectable ranges of the ash clouds, which means the observed plume edges do not exactly represent the "zero" concentration edges, but the "detectable" edges; (2) very low concentration is not of interest with respect to air-safety in volcanic ash application. "

11. *P3,l10: "hundreds of meters"*

12. *P3, l16: "Kasatochi"*

13. *P3, l16: Personal communication should probably list from whom the information is from.*

14. *P3, l21: Whether thin ash clouds are less of a concern to aviation or not does not necessarily make them less of a concern for a data assimilation system.*

Responses to 11, 12, 13, 14:
Thanks for the corrections and comments. We agree. In the new version, we have considered the thin ash clouds by updating the thickness range of 0.5–3.0 to 0.2–3.0 km, based on the literatures for the volcano Eyjafjallajökull. Since we focus on the case study of the volcano Eyjafjallajökull, some irrelevant literature have been removed. We have re-written this part in line(s) 3.29–4.5:
" For the vertical thickness information of volcanic ash clouds, Schumann et al. (2011) investigated on the 2010 Eyjafjallajökull eruption using airborne data that the volcanic ash clouds spread over large parts of Central Europe, mostly from hundreds of meters to 3 km depth. This is consistent with the results of (Marenco et al., 2011) who observed layer depths between 0.5 and 3.0 km. Dacre et al. (2015) also examined the ground-based lidar data for the Eyjafjallajökull eruption and found a mean layer depth of $1.2\pm0.9$ km and compared this with model based estimates of $1.1\pm0.8$ km. Prata and Prata (2012) found variable thicknesses ranging from 0.2 up to 3 km. The vast majority of data suggest thickness in the range 0.2–3 km for the 2010 Eyjafjallajökull eruption. Based on these investigations, it is not realistic to use a deterministic value to represent the overall ash cloud thickness, but we can reasonably assume that the thickness has a range of 0.2–3.0 km at the corresponding horizontal location of the SEVIRI retrieved measurements. Although this thickness information is not deterministic, its uncertainty spread is suitable in an observational operator for satellite data assimilation. Note that the thickness range can be different for other volcanic eruptions. For example, Prata et al. (2015) reported low cloud thickness with 80% of cases for the 2006 Chaiten eruption less than 400 m, thus a thickness range of 0.1–0.4 km is recommended for that eruption. "

15. *P4, l12: "As a parameter..." The meaning of this sentence is hard to understand.*

Response:
Yes. It is re-written in line(s) 4.32–5.1:

" The ash cloud top height is adopted with the SEVIRI-KNMI product of ash height, which has been evaluated with a reasonable accuracy, as reported by de Laat and van der A (2012). "

16. *P4, l15: Also unclear, in the previous paragraph a number of data are described as SEVIRI products.*

    Response:
    Yes, it is re-written in line(s) 5.4–5.5:
    " We acquire the data (described above) from the European Space Agency (ESA) funded project – Volcanic Ash Strategic Initiative Team (VAST). The data are illustrated in Fig. 1. "

17. *P5, l25: "ML_blue" is not an intuitive terminology, something physically motivated would be much better.*

    Response:
    Agree. In the new version, we don't restrict the extraction at only the blue layer, but extend it to both the blue and yellow layers. Thus, there is no need to specifically represent "ML_blue".

18. *P6, Eq. 7: Where does this formula come from? Usually, uncorrelated errors add in quadrature. On line 13, a "conditional probability relation" is mentioned, but this does not make Eq. 7 easier to understand.*

    Response:
    Based on the comments of another Reviewer, we have re-written this part. The previous formula was developed by us, but not a very convincing formula. We have re-written this formula and this part in line(s) 6.29–7.5:
    " Eq. (4) can be written as

    $$C = \frac{\cos(\alpha)}{N_s T_1}M + \frac{\cos(\alpha)}{N_s T_2}M + \cdots + \frac{\cos(\alpha)}{N_s T_{N_s}}M \quad . \tag{6}$$

    Thus, given the standard uncertainty in mass loadings ($u_M$, i.e., the data of retrieval error, as shown in Fig. 1**c**) and according to the theory of combined standard uncertainty (Kirkup and Frenkel, 2006) for uncorrelated errors, $u_C$ can be calculated by

    $$u_C = \sqrt{\left(\frac{\cos(\alpha)}{N_s T_1}\right)^2 + \left(\frac{\cos(\alpha)}{N_s T_2}\right)^2 + \cdots + \left(\frac{\cos(\alpha)}{N_s T_{N_s}}\right)^2} \times u_M \quad . \tag{7}$$

    Now $u_C$ is quantified, which together with C describes the 3D measurements (mean, uncertainties) for ensemble-based data assimilation. "

19. *P7, l12: This description of the plume height needs much more detail to be understandable.*

Response:

Yes. We have added more details and re-written this part in line(s) 8.6–8.10:

" The PH here represents the eruption height above the vent of the volcano, which was monitored with the weather radar at Keflavík (155 km west of the volcano) by Icelandic Meteorological Office (IMO), sampling every 5 minutes (Gudmundsson et al., 2010). In this study PH is taken based on the maximum detection data of this weather radar (see the Fig. 2a in (Gudmundsson et al., 2010)), and usually the uncertainty of PH is taken as 20 % (Bonadonna and Costa, 2013). "

20. *P8, l6: "… which is mainly due to lack of sedimentation processes" scared me quite a bit, but Fu et al. (2016) only state that the model contains sedimentation processes, but not coagulation, evaporation and resuspension. I guess assumptions about the size distribution are also important here.*

Response:

Yes, it was not an accurate statement. We have revised it in line(s) 9.8–9.9:

" In reality, a potential overestimation is usually elusive and hard to avoid, which is mainly due to the difficulty in getting an accurate estimation of the particle size distribution and modeling the physical processes (Fu et al., 2016). "

21. *P8, l7: Using DA to correct for model overestimation is a rather unphysical "solution". It would be more satisfying to improve the physics of the model if in fact this is the source of the over-estimation.*

Response:

Agree. But for our case we actually don't know what is the exact source for over-estimation. We have added this statement in the new version in line(s) 9.27–9.30:

" Note that using DA to correct for model over-estimation is an unphysical "solution". It would be more satisfying to improve the physics of the model if in fact this is the source of the over-estimation, but for this case we don't know what is the exact source (which could be e.g., meteorology, ESPs, model processes). "

22. *P8, l26: This is not a very convincing validation of the SOO.*

Response:

I agree. It is not a suitable statement and has been removed.

23. *P8, l27: Is there a reason why SEVIRI mass loading retrieval error and the standard deviation of the mass loadings should be similar? The reason is not apparent.*

Response:

I agree with you that the reason is not apparent, and actually there is no specifical reason for it. This is also why we checked both to see the differences. Since only the retrieval error is (while the standard deviation of the mass loadings is not) relevant to the study, thus the statement on this check is not very relevant to the study. To avoid unclear information, in

the new version, we have removed this sentence.

24. *P9, l3: I don't see why satellite measurements wouldn't be better to test the duration of improvements to forecasts from DA. That satellite measurements have "big uncertainties" seems a strange argument, given that the preceding portions of the paper have used satellite measurements for the DA. While I don't mean to play down the importance of in situ measurements, in this case, the airplane flight track sampled only the very edges of the old ash plume, with concentrations orders of magnitude less than the peak values and much lower than the threshold for airplane safety. Therefore, the accuracy of the forecasts at the times and places of the in situ data used here seem to be of little importance to the question of how good the model is at predicting ash concentrations dangerous to airplanes.*

Response:
Thank you for the comments and explanations, which helps improve the presentation of this part. We have answered this question in the Major concern 4. We agree that satellite measurements should be better to test the duration than the aircraft measurements. In the new version, we use both for the test. Although the low magnitude of aircraft measurements may result in little importance to verify high ash concentrations, but comparisons using them can indicate how good the assimilation is at reconstructing the outskirts/boundaries of the ash plume, which is important to describe the plume's structure. Thus, we kept the usage of them as a second test after the main test with satellite data, and also mentioned this in line(s) 11.30–11.32:

[revised manuscript text omitted]

---

## Author Response (AR3)

Dear Editor,

Herewith we submit the revised manuscript acp-2016-436, "Data assimilation for volcanic ash plumes using a Satellite Observational Operator: a case study on the 2010 Eyjafjallajökull volcanic eruption ".

We would like to thank you and the anonymous reviewer#4, and really appreciate the detailed comments and suggestions. We have carefully considered all the concerns and made changes accordingly in the revised paper.

In the following we will give our answers and reactions to the comments.

Kind regards,
Guangliang Fu
on behalf all co-authors

(The revised manuscript is in the latter part of this pdf.

**Reply to Reviewer #4:**

1. *While the description of the SOO in Sec 3 is improved, I still don't follow completely. In equations 2 and 3, you assume that a range of concentrations are possible, given the observed mass loading (M_v) and a set of possible thicknesses. So far, so good. Then, in equation 4, you calculate an average concentration (C) from the assumed distribution of possible concentrations (C_i). Is this the only concentration that is used in the DA? If so, why not just calculate C as M_v/T_m, where T_m is an assumed thickness? It works out to be the same thing. In your method, using a range of T from 0.2-3 km, with step sizes of 0.05 km, works out to a T_m of 1.0 km (where T_m=1/N * sum(1/T_i). This value seems consistent with the values shown in Figure 1d and Figure 3 (although with different color- scales it's hard to be sure). But then, pg 6, l22-23, you say C is used as the concentration between the altitudes H_top-T_high and H_top. The thickness of this layer is T_high, i.e., 3 km. If the concentration C is calculated implicitly assuming a thickness of 1 km, and then a vertical profile is constructed with a concentration extending over 3 km, won't the vertical integral of your constructed profile have a mass loading 3 times larger than the actual measurement?*

   *Maybe I'm missing something here. The construction of an ensemble of concentrations (C_i) hints at a more innovative methodology: is this ensemble of concentrations used in the assimilation? It's not clear that it is, and the introduction of the mean concentration C makes me think that only a single value is used. I would suggest that if this is the case, that a simpler description be used here? if in effect you are assuming a constant, static plume thickness, it is important that that be known by the reader. And of course, it is very important that the vertical profile of ash concentration used in the DA be consistent with the satellite mass loading!*

   Response:
   Thank you very much to help improve the presentation of SOO. We agree that some details related to the formulation were not clear. Yes, you are right that a single value of the mean concentration is used (corresponding to each mass loading), by using a harmonic-mean plume thickness. We agree that a simpler description is necessary for the reader to follow/understand the method. Thanks for your detailed suggestion. We have re-written this part in line(s) 6.18–6.27:

   " Therefore, at one measurement time, the mean (C) of the sampled ash concentrations can be calculated by Eq. (4)),

   $$C = \frac{1}{N_s} \sum_{i=1}^{N_s} C_i = \frac{M_v}{\frac{N_s}{\sum_{i=1}^{N_s} \frac{1}{T_i}}} \quad . \tag{4}$$

   Here, we note that $\frac{N_s}{\sum_{i=1}^{N_s} \frac{1}{T_i}}$ is the harmonic mean of $T_i$ $(i = 1, 2, \cdots, N_s)$. Thus the

representation of Eq. (4)) can be simplified as

$$C = \frac{M_v}{T_m} \quad , \tag{5}$$

where given a harmonic-mean thickness $T_m$ (which equals to the harmonic mean of $T_i$), C as calculated by Eq. (5) is used in this study as the extracted concentration.

Note that, if $\Delta T$ is fixed (i.e., 0.05 km in this case study), the harmonic-mean thickness $T_m$ is a constant/static plume thickness (see Eq. (2)–(5)). "

In the description, we first present the mean concentration as in Eq. (4), and later simplify the calculation to Eq. (5). The reason we don't directly go to Eq. (5) is that we think it should show the reader why we choose the harmonic mean of the sampled thickness to represent $T_m$.

We totally agree that it is very important that the vertical profile of ash concentration used in DA should be consistent with the satellite mass loadings. Actually we have indeed considered this issue that (in the previous version) after we extracted the concentrations, we calculated a thickness (within which the extracted concentrations are meant for) by dividing the vertical mass loadings and the extracted concentration. We admit it was indeed a misleading in the previous presentation to mention the target layers for the extraction is within 3 km. In the new version, we have clearly stated this issue in line(s) 6.27–6.32:

" Another note is that the extracted concentration C only represents the ash concentrations between the heights [$H_{top}$ - $T_m$] and $H_{top}$. This is very important to guarantee the vertical profile of extracted ash concentration to be consistent with the satellite mass loadings. Therefore, the target layers for the extraction is actually between the heights [$H_{top}$ - $T_m$] and $H_{top}$, see Fig. 2.

The outcome of SOO can be considered as preprocessing to the satellite data assimilation system. The extracted data represent the data at the target layers, which can be taken as the data within the assumed layer thickness ($T_m$). "

2. *Equations 5 and 6 are unnecessarily complicated. C=M_v/T_m (with T_m a constant defined above), so sigma_C=sigma_M/T_m.*

Response:
Agree. As suggested, we have changed it in line(s) 7.5–7.9:
" Now we quantify the standard uncertainty of C ($u_C$, i.e., the SOO error), which is important for a data assimilation system. Eq. (4) can be written as $C = \frac{\cos(\alpha)}{T_m} \times M$. Thus, given the standard uncertainty in mass loadings ($u_M$, i.e., the data of retrieval error, as shown in Fig. 1c), $u_C$ can be calculated as

$$u_C = \frac{\cos(\alpha)}{T_m} \times u_M \quad , \tag{6}$$

which together with C describes the 3D measurements (mean, uncertainties) for ensemble-based data assimilation. "

3. *The way that ensemble members are created is still not quite clear. I guess that different ensemble members have different assumed plume heights of the ash injection, which is based on observed plume heights and uncertainties. The discussion about the relationship between plume heights and mass injection should probably be brought up to the method description.*

Response:

Yes. The different ensemble members have different PH of the ash injection (which is based on the observed PH and its uncertainties) and the different MER (which is based on the relationship between PH and MER).

I agree how ensemble members are created remained unclear to readers, and the discussion on the relationship between PH and MER should be moved to the method description. Thus, in the new version, we have added a new sub-section (4.3 Creation of ensemble plumes) to explicitly describe this, as in line(s) 8.22–9.6:

" 4.3 Creation of ensemble plumes

The specification of uncertainties is essential for a successful data assimilation. Here we use uncertainties in the Plume Height (PH) in the process of creating ensemble members.

The PH represents the eruption height above the vent of the volcano, which was monitored with the weather radar at Keflavík (155 km west of the volcano) by Icelandic Meteorological Office (IMO), sampling every 5 minutes (Gudmundsson et al., 2010). In this study PH is taken based on the detection data of this weather radar (see the Fig. 2a in (Gudmundsson et al., 2010)), and usually the uncertainty of PH is taken as 20% (Bonadonna and Costa, 2013). The stochastic Plume Height (PH) is assumed to be temporally correlated with exponential decay. The correlation parameter $\tau$ is set to be 1 hour (Fu et al., 2015). Thus, the PH noise $(N_{ph})$ at two times $(t_1$ and $t_2)$ has the relation (Evensen, 2004) of $\mathbb{E}[N_{ph}(t_1) \cdot N_{ph}(t_2)] = e^{\frac{-|t_1-t_2|}{\tau}}$, where $\mathbb{E}$ represents the mathematical expectation.

The Mass Eruption Rate (MER) is calculated based on each uncertain PH, by using an empirical relationship (Mastin et al., 2009) between PH(km) and MER (kg s$^{-1}$)

$$PH = 2.00V^{0.241}, \quad and \quad \frac{V}{MER} = \frac{1.5e3}{4.0e6}. \qquad (7)$$

Thus, although we only add uncertainties in PH, MER is also not deterministic.

Therefore, the different ensemble members have different PH of the ash injection (which is based on the observed PH and its uncertainties) and the different MER (which is based on Eq. (7)). "

4. *It is great that the authors have repeated their DA without the SOO to illustrate the impact of the SOO. But, I have to admit I just don't understand how any DA, which ingests observations which have values less than half that of the ensemble mean forecast can produce an analysis which barely differs in magnitude from the forecast (Fig 6, b vs. d). I therefore worry that the apparent different between "with SOO" and "without SOO" might be due to other considerations, e.g., perhaps differences in the weighting of the observations in each case based on their estimated uncertainties. For example, if in the SOO case, the estimated concentration above the retrieved cloud height is zero, and has a small estimated uncertainty, this could be a very strong constraint in the DA which the "without SOO" analysis*

*wouldn't have. I get the impression that the "without SOO" analysis only includes the mass loadings, not the cloud height, so if this is true, the comparison is not quite fair. At least some discussion of these issues should be included.*

Response:

Thanks for this valuable comment and suggestion. We understand your concern, and we agree that usually DA (which ingests observations which have values less than half that of the ensemble mean forecast) should produce the analysis different in magnitude from the forecast.

However, for our application (vertically narrow) the artificial vertical correlations (caused by the integral-type of $H_k$, see introduction) strongly degrade the performance of standard DA (e.g., EnSR without SOO). These influences are accumulated one by one assimilation step, finally resulting in the assimilation result in Fig. 6**d** which barely differs in magnitude from the forecast (Fig. 6**b**). We have discussed this issue in line(s) 10.17–10.24:

" One may also want to check the assimilation effect without SOO at the initial analysis time (i.e., 01:00 UTC, 16 May 2010), which is illustrated in Fig. 7. It can be seen that the assimilation difference between with and without SOO is small (by comparing Fig. 7**c** and 7**d**). While, at the second analysis time (i.e., 02:00 UTC, 16 May 2010) the results show much bigger differences (between Fig. 7**g** and Fig. 7**h**). These results verify and examine the influences of the artificial/spurious vertical correlations (caused by the integral-type of $H_k$, see introduction). These influences are examined to be accumulated step by step in our volcanic ash application, finally resulting in the assimilation result in Fig. 6**d** which barely differs in magnitude from the forecast (Fig. 6**b**). Therefore, these influences must be considered/avoided in order to obtain an acceptable assimilation result. "

We also understand your comment that the comparison (between EnSR with and without SOO) could be understood as "not quite fair", thus some discussions have been included in line(s) 11.1–11.8:

[revised manuscript text omitted]

---

## Author Response (AR4)

Dear Editor,

Herewith we submit the revised manuscript. We sincerely thank you and the fourth reviewer for the detailed suggestions. We have carefully considered all of them in the revised paper.

Kind regards for 2017,
Guangliang Fu
on behalf all co-authors

(The revised manuscript is in the latter part of this pdf.)

**Reply to comments:**

1. *2.14: I know what you are trying to say here, but Data assimilation, as a general technique, doesn't really create accurate initial conditions, it it produces a best estimate of of a physical state. Only when you use an ensemble technique, and build your ensemble by varying the initial conditions, does DA effectively act to choose the best initial conditions. Anyhow, you might be a little more careful here to avoid confusing the reader.*

   Response:
   Agree. We have re-written the sentence in line(s) 2.14–2.16:

   " Data assimilation (DA) is one of the most commonly used approaches for real-time forecasting problems (Evensen, 2003; Bocquet et al., 2015; Fu et al., 2015). It can be used to provide an estimate of the state of the atmosphere which can then be used to initialize forecasts. "

2. *3.35: You say here it is not realistic to use a static value for the cloud thickness... but this is what you then do. Although you are now clearer about the technique used (in Sec 3.1), statements like this can be confusing to the reader, making it seem like your method incorporates a range of thicknesses into the assimilation method somehow.*

   Response:
   I agree with you that the previous sentence was confusing to the reader. To avoid the confusion, we have re-written the sentence in line(s) 3.35–4.1:

   " Based on these investigations, we can reasonably assume that the thickness has a range of 0.2–3.0 km at the corresponding horizontal location of the SEVIRI retrieved measurements. "

3. *4.1: Looks like an error with the citation here.*

Response:
Agree. The citation was not so relevant to the paper. We have removed it in the new version, in line(s) 4.2–4.3.

4. *6.26: Even if Delta T was not "fixed", i.e., if the thicknesses $T_i$ were not a uniform sequence, the mean (or harmonic mean) of $T_i$ is still going to be a scalar, so I don't see the logic of "if Delta T is fixed".*

Response:
Thanks for the explanation. As suggested, we have re-written the sentence in line(s) 6.24:

[revised manuscript text omitted]
_{\text{high}} - T_{\text{low}}$ (i.e., the substraction between $T_{\text{high}}$ and $T_{\text{low}}$) represents the yellow layer with the fixed thickness of 2.8 km. $\Delta T$ is chosen at a small value compared to $T_{\text{low}}$, which guarantees $N_s$ is not too small (e.g., less than 2) to have a sufficient number of sample thickness $T_1$, $T_2$, $\cdots$, $T_{N_s}$. (e.g., $\Delta T$ is chosen as 0.05 km in this case study, thus $N_s$ is calculated as 57.)

Corresponding to the sampled thickness (i.e., $T_1$, $T_2$, $\cdots$), the ash concentration can be calculated also as a sample from $C_1$ to $C_{N_s}$ through Eq. (3). According to Eq. (2) and Eq. (3), $T_i$ ($i = 1, 2, \cdots, N_s$) is unchanged during the dispersion of the ash cloud, while $C_i$ is temporally changed but it does not depend on $H_{\text{top}}$. Therefore, at one measurement time, the mean (C) of the sampled ash concentrations can be calculated by Eq. (4)),

$$C = \frac{1}{N_s} \sum_{i=1}^{N_s} C_i = \frac{M_{\text{v}}}{\frac{N_s}{\sum_{i=1}^{N_s} \frac{1}{T_i}}} \quad . \tag{4}$$

Here, we note that $\frac{N_s}{\sum_{i=1}^{N_s} \frac{1}{T_i}}$ is the harmonic mean of $T_i$ ($i = 1, 2, \cdots, N_s$). Thus the representation of Eq. (4)) can be simplified as

$$C = \frac{M_{\text{v}}}{T_{\text{m}}} \quad , \tag{5}$$

where given a harmonic-mean thickness $T_m$ (which equals to the harmonic mean of $T_i$), C as calculated by Eq. (5) is used in this study as the extracted concentration.

Note that, the harmonic-mean thickness $T_m$ is a constant/static plume thickness (see Eq. (2)–(5)). Another note is that the extracted concentration C only represents the ash concentrations between the heights [$H_{\text{top}}$ - $T_{\text{m}}$] and $H_{\text{top}}$. This is very important to guarantee the vertical profile of extracted ash concentration to be consistent with the satellite mass loadings. Therefore, the target layers for the extraction is actually between the heights [$H_{\text{top}}$ - $T_{\text{m}}$] and $H_{\text{top}}$, see Fig. 2.

The outcome of SOO can be considered as preprocessing to the satellite data assimilation system. The extracted data represent the data at the target layers, which can be taken as the data within the assumed layer thickness ($T_{\text{m}}$).

**3.2 SOO error**

Fig. 2 and Eq. (2) to (4) describe the details of the SOO. The operator transforms the 2D ash mass loadings (M) to 3D ash concentrations (C). Fig. 3**a** shows the extracted ash concentrations (C) at the target layers. It can be seen that the extracted ash concentrations in the ash plume are mostly between 0.5 and 3.0 mg m$^{-3}$.

5   Now we quantify the standard uncertainty of C ($u_\mathrm{C}$, i.e., the SOO error), which is important for a data assimilation system. Eq. (4) can be written as C $= \frac{\cos(\alpha)}{\mathrm{T_m}} \times$ M. Thus, given the standard uncertainty in mass loadings ($u_\mathrm{M}$, i.e., the data of retrieval error, as shown in Fig. 1**c**), $u_\mathrm{C}$ can be calculated as

[revised manuscript text omitted]